# LORA-MaOO: Learning Ordinal Relations and Angles for Expensive Many-Objective Optimization

## Abstract

Many-objective optimization (MaOO) simultaneously optimizes many conflicting objectives to identify the Pareto front - a set of diverse solutions that represent different optimal balances between conflicting objectives. For expensive MaOO problems, due to their costly function evaluations, computationally cheap surrogates have been widely used in MaOO to save evaluation budget. However, as the number of objectives $M$ increases, the cost of using surrogates increases rapidly as many optimization algorithms need maintain $M$ surrogates. In addition, a large $M$ indicates a high-dimensional objective space, increasing the difficulty of maintaining solution diversity. It is a challenge to reach diverse optimal solutions with a relatively low cost of using surrogates for MaOO problems. To handle this challenge, we propose LORA-MaOO, a surrogate-assisted MaOO algorithm that learns $M$ surrogates from spherical coordinates, including an ordinal-regression-based surrogate that learns the ordinal relations between solutions (denoted as radial surrogate) and $M$-1 regression-based surrogates that trained on angular coordinates (denoted as angular surrogates). In each optimization iteration, model-based search is completed with a single radial surrogate, while $M$-1 angular surrogates are used only once for selecting diverse candidates. Therefore, the frequency of using angular surrogates is largely reduced, lowering the cost of using surrogates. In addition, we design a clustering method to quantify artificial ordinal relations for non-dominated solutions and improve the quantification of dominance-based ordinal relations. These ordinal relations are used to train the radial surrogate which predicts how desirable the candidates are in terms of convergence. The solution diversity is maintained via angles between solutions instead of pre-defined auxiliary reference vectors, which is parameter-free. Experimental results show that LORA-MaOO significantly outperforms other surrogate-assisted MaOO methods on most MaOO benchmark problems and real-world applications.

## 1 Introduction

Multi-objective optimization problems (MOOPs) and many-objective optimization problems (MaOOPs) [1] widely exist in many real-world applications, such as production scheduling Lin & Gen (2018), traffic signal control Shaikh et al. (2020), and water resource engineering Janga Reddy & Nagesh Kumar (2021). These MOOPs and MaOOPs have many conflicting objectives to optimize, and thus all objectives cannot reach their optimum simultaneously. As a result, the optimum of MOOPs and MaOOPs is the *Pareto front (PF)*: A set of non-dominated solutions in the objective space that represent different optimal balance between conflicting objectives. These optimization problems aim to find non-dominated solutions that are close to the PF and also well distributed along the PF, indicating that MOOPs and MaOOPs should consider both convergence and diversity.

Various evolutionary optimization algorithms have been proposed to solve MOOPs Deb et al. (2002) and MaOOPs Deb & Jain (2013). These optimization algorithms usually require plenty of solution samplings and evaluations to find converged and diverse non-dominated solutions. However, in many real-world MOOPs and MaOOPs, the evaluation of solution performance could be costly Yu

---

[1]Multi-objective optimization has 2 or 3 objectives, many-objective optimization has 4 or more objectives.

et al. (2022). Therefore, the evaluation budget only allows a limited number of solutions to be evaluated on the expensive objective functions. To address expensive optimization problems, evolutionary optimization is combined with computationally cheap surrogates to enhance sampling efficiency and save evaluations, which are known as surrogate-assisted evolutionary algorithms (SAEAs).

Yet, compared with well-studied MOOPs, MaOOPs are more challenging for SAEAs since the cost of using surrogates and the difficulty of maintaining solution diversity could increase rapidly as the number of objectives $M$ increases. For example, conventional SAEAs usually use regression-based surrogates to approximate each objective function separately Chugh et al. (2016); Song et al. (2021). For MaOOPs, many objectives indicate maintaining many surrogates for surrogate-assisted search and selection, which results in a low efficiency of SAEAs. In addition, it is difficult to maintain solution diversity in high-dimensional objective space. Some SAEAs Knowles (2006); Zhang et al. (2010); Chugh et al. (2016) need to investigate proper parametric strategies to generate reference vectors or divide objective space into subspaces. Recently, a family of classification-based SAEAs Pan et al. (2018); Hao et al. (2022) attempted to use a single surrogate to learn pairwise dominance relations, which hugely reduces the cost of using surrogates. However, their single surrogate can provide very limited information about solution diversity, making these algorithms more efficient but less effective than the SAEAs with many surrogates. Additionally, many Bayesian optimization (BO) algorithms Tu et al. (2022); Zhang & Golovin (2020); Paria et al. (2020); Abdolshah et al. (2019) were proposed to solve expensive MOOPs. However, they are mainly based on the computation of hypervolume, which would be very time-consuming in MaOOPs.

In this paper, we propose a different framework to implement surrogate-assisted evolutionary optimization for expensive MaOOPs, named LORA-MaOO, where a single surrogate is developed to learn ordinal relations for guiding optimization, and several angular surrogates are generated from spherical coordinates to maintain diversity. LORA-MaOO reaches diverse and as optimal as possible solutions for MaOOPs but with relatively low cost of using surrogates. Our major contributions are summarized as follows:

- We introduce the framework of spherical coordinates approximation into surrogate-assisted evolutionary optimization and proposed LORA-MaOO to solve expensive MaOOPs. Different from existing SAEAs which learn approximation models from Cartesian coordinates and use all surrogates to handle convergence and diversity, we consider convergence and diversity via separate surrogates: An ordinal surrogate is treated as a radial coordinate for convergence purpose, while remaining regression-based surrogates approximate angular coordinates for maintaining diversity. This framework provides a flexibility to reduce the frequency of using surrogates and thus reduce the cost of using surrogates.

- We develop a novel ordinal-regression-based model to learn the ordinal landscape of expensive MaOOPs. A clustering method is designed to generate artificial ordinal relations for improving modeling performance for many objectives. In addition, we also propose an improved way to quantify dominance-based ordinal relations for surrogate modeling.

- A non-parametric approach is developed to select diverse solutions for expensive evaluations via our angular coordinate surrogates.

- Extensive experiments on benchmark and real-world optimization problems are conducted under a range of scales and numbers of objectives. Empirical results show that our LORA-MaOO is effective and outperforms the state-of-the-arts.

## 2 RELATED WORK

### 2.1 MULTI-/MANY-OBJECTIVE SURROGATE-ASSISTED EVOLUTIONARY ALGORITHMS

**Regression-based SAEAs.** Regression-based SAEAs employ regression-based surrogates such as Kriging Stein (1999); Williams & Rasmussen (2006) to approximate either the objective values of solutions or the objective functions of expensive problems Jin (2005). To maintain solution diversity, ParEGO Knowles (2006) employs a Kriging model to iteratively approximate an scalarized objective function which aggregates all objectives into one via a set of pre-defined scale vectors. In MOEA/D-EGO Zhang et al. (2010), plenty of scale vectors are generated uniformly to decompose the target MOOP into many single-objective subproblems. K-RVEA Chugh et al. (2016) also designs a set of

scale vectors as reference vectors to maintain solution diversity. Similarity or density estimation is an alternative option for maintaining diversity. For instance, KTA2 Song et al. (2021) estimates the distribution status of non-dominated solutions by defining a similarity or density indicator.

**Classification-based SAEAs.** In model-based optimization, the optimization is guided by the relation between solutions rather than accurate objective values. Therefore, there is a tendency for recently proposed SAEAs to use classification-based surrogates to learn the relation between solutions directly. CSEA Pan et al. (2018) trains a neural network to justify whether candidate solutions can be dominated by given reference points or not. $\theta$-DEA-DP Yuan & Banzhaf (2022) uses two neural networks to predict the Pareto dominance relation and $\theta$-dominance relation between two solutions, respectively. REMO Hao et al. (2022) employs a neural network to fit a ternary classifier, which is able to learn the dominance relation between pairs of solutions. Compared with regression-based SAEAs, although classification-based SAEAs take advantage of learning solution relations directly, their drawbacks are also clear: The prediction of solution relations lacks the information of how solutions are distributed in the objective space, making it difficult for classification-based SAEAs to maintain solution diversity. In Pan et al. (2018); Hao et al. (2022), a radial projection selection approach is adapted to select diverse reference points. However, its effect on diversity maintenance is limited. In addition, although classification-based SAEAs maintain only one surrogate, the cost of learning pairwise relations from large datasets is inevitably increased.

**SAEAs based on Other Surrogates.** HSMEA Habib et al. (2019) uses an ensemble of multiple surrogates in the optimization. In addition, a new category of surrogates, namely dominance-based ordinal regression surrogate Yu et al. (2019) or level-based classification surrogate Liu et al. (2022), is proposed to combine regression-based surrogates with classification-based surrogates. However, the shortcoming remains the same as these surrogates lack the information of solution distribution, especially when $M$ is large. Moreover, in MaOOPs, dominance-based ordinal relations could be less effective due to the large proportion of non-dominated solutions.

## 2.2 MULTI-OBJECTIVE BAYESIAN OPTIMIZATION

**MOBO.** Bayesian Optimization (BO) Song et al. (2022); Huang et al. (2024) is also a typical model-based optimization method for expensive optimization, while multi-objective BO (MOBO) methods are designed for expensive MOOPs Daulton et al. (2021; 2022); Lin et al. (2022); Ahmadianshalchi et al. (2024). Some MOBO generalizes the acquisition functions such as upper confidence bound (UCB) Zuluaga et al. (2016), expected improvement (EI) Emmerich et al. (2006), Thompson sampling Belakaria et al. (2020), to solve expensive MOOPs. In addition, entropy search methods have also been employed in MOBO Belakaria et al. (2019); Suzuki et al. (2020). To maintain solution diversity, the EI of a multi-objective performance indicator, Hypervolume (HV) Zitzler & Thiele (1998), was used as the acquisition function in recent MOBO Daulton et al. (2020); Lin et al. (2022). Based on the Hypervolume improvement (HVI), PSL Lin et al. (2022) proposes a learning method to approximate the whole Pareto set for MOBO, and PDBO Ahmadianshalchi et al. (2024) automatically selects the best acquisition function for objective functions in each iteration. However, the time complexity of computing HV increases exponentially with the number of objectives, which may limit the application of MOBO methods on MaOOPs.

**Connection to Multi-/Many-Objective SAEAs.** Both multi-/many-objective SAEAs (denoted as SAEAs below) and MOBO are model-based optimization methods. A SAEA is also a MOBO if it uses probability models as surrogates and employs an acquisition function for candidate selection , and a MOBO is also a SAEA if it searches candidate solutions with evolutionary search algorithms. Therefore, some model-based optimization methods belong to both SAEAs and MOBO Knowles (2006); Emmerich et al. (2006); Zhang et al. (2010).

## 3 LORA-MAOO: THE PROPOSED ALGORITHM

This section first introduces the LORA-MaOO framework, followed by detailed algorithm descriptions.

## 3.1 LORA-MaOO FRAMEWORK

The pseudocode of LORA-MaOO is depicted in Alg. 1, it consists of four phases:

---

**Algorithm 1** LORA-MaOO framework

---

**Input:** $M$ objective functions of the optimization problem $f(\boldsymbol{x}) = (f_1(\boldsymbol{x}), \ldots, f_M(\boldsymbol{x}))$;
  Evaluation budget: The number of allowed function evaluations $FE_{max}$.
**Procedure:**
 1: Sample a set of solutions $\{\boldsymbol{x}_1, \ldots, \boldsymbol{x}_{11D-1}\}$ and evaluate them on $f$.
 2: Save all evaluated solutions $(\boldsymbol{x}, f(\boldsymbol{x}))$ in an archive $S_A$. Set the number of used function evaluations $FE = |S_A|$.
 3: **while** $FE < FE_{max}$ **do**
 4:   Ordinal training set $S_o \leftarrow$ Quantify ordinal values for all $\boldsymbol{x}_i \in S_A$ (Alg. 2).
 5:   Ordinal surrogate $h_o \leftarrow$ Train Kriging$(S_A, S_o)$.
 6:   Population of candidate solutions $P \leftarrow$ Run an optimizer on $h_o$ (Alg. 3).
 7:   $\boldsymbol{x}_1^* \leftarrow$ Use the ordinal surrogate to select a solution from $P$ by convergence criterion.
 8:   Evaluate $\boldsymbol{x}_1^*$ and update $S_A = S_A \cup \{(\boldsymbol{x}_1^*, f(\boldsymbol{x}_1^*))\}$, $FE = FE + 1$.
 9:   Angular training set $S_a \leftarrow$ Calculate angular coordinates for all $\boldsymbol{x}_i \in S_A$.
10:   $M$-1 angular surrogates $h_{ai} \leftarrow$ Train Kriging $(S_A, S_a)$, $i = 1, \ldots, M-1$.
11:   $\boldsymbol{x}_2^* \leftarrow$ Use angular surrogates to select a solution from $P$ by diversity criterion (Alg. 4).
12:   Evaluate $\boldsymbol{x}_2^*$ and update $S_A = S_A \cup \{(\boldsymbol{x}_2^*, f(\boldsymbol{x}_2^*))\}$, $FE = FE + 1$.
13: **end while**
**Output:** Non-dominated solutions in archive $S_A$.

---

1. Initialization: An initial dataset of size $11D$ - 1 (As suggested in the literature Knowles (2006)) are sampled from the decision space using the Latin hypercube sampling (LHS) McKay et al. (2000) (line 1), where $D$ is the dimensionality of decision variables. The sampled solutions are evaluated on objective functions $f$ and then saved in an archive $S_A$ (line 2).

2. Surrogate modeling: For all solutions $\boldsymbol{x} \in S_A$, quantify their ordinal values (line 4) and calculate their angular coordinates (line 9). The set of ordinal values $S_o$ is used to train the ordinal surrogate $h_o$ (line 5). The angular coordinates are used to fit $M - 1$ angular surrogates $h_{ai}$ separately (line 10).

3. Sampling (Search and Selection): Run an optimizer on surrogate $h_o$ to generate a population of candidate solutions $P$ (line 6). Select optimal candidate solutions $\boldsymbol{x}_1^*$, $\boldsymbol{x}_2^*$ from $P$ based on surrogates $h_o$, $h_{ai}$, respectively (lines 7 and 11).

4. Update: Evaluate new optimal candidate solutions $\boldsymbol{x}_1^*$, $\boldsymbol{x}_2^*$ on expensive objective functions $f$, update archive $S_A$ and the number of used function evaluations $FE$ (lines 8 and 12). The algorithm will go to phase 2 until the evaluation budget $FE_{max}$ has run out.

## 3.2 SURROGATE MODELING

The ordinal surrogate $h_o$ is mainly trained on dominance-based ordinal relations, additional clustering-based artificial ordinal relations will be introduced for training if $M$ is large. Additionally, for an $M$-objective problem, $M$-1 angular surrogates $h_{ai}$ are trained on angular coordinates. These surrogates are used in the selection procedure for diversity but are idle in the search procedure.

### 3.2.1 LEARNING DOMINANCE-BASED ORDINAL RELATIONS.

In LORA-MaOO, the concept of ordinal regression Yu et al. (2019) is adapted to learn dominance-based ordinal relations. Clearly, the dominance-based ordinal relation between a set of reference points $S_{RP}$ and a given solution $\boldsymbol{x}$ is quantified as a relation value. Such a relation value is a numerical value that is used for training the ordinal-regression surrogate $h_o$. The quantification of relation values consists of two steps: The selection of reference points $S_{RP}$ and the computation of relation values.

**Selection of Reference Points.** We propose the definition of $\lambda$-dominance relationship to simplify the selection of reference points.

**Definition 1.** *($\lambda$-Dominance Relationship)*
*A solution $\boldsymbol{x}^1$ is said to $\lambda$-dominate another solution $\boldsymbol{x}^2$ (denoted by $\boldsymbol{x}^1 \prec_\lambda \boldsymbol{x}^2$) if and only if:*

$$g_\lambda(\boldsymbol{x}^1) \prec g_\lambda(\boldsymbol{x}^2), \tag{1}$$

*where $\lambda \geq 0$ is the dominance coefficient and $g_\lambda$ is a smooth objective function defined as:*

$$f_{in}(\boldsymbol{x}) = \frac{f_i(\boldsymbol{x}) - z_i^*}{|z_i^{nad} - z_i^*|}, \tag{2}$$

$$g_{\lambda,i}(\boldsymbol{x}) = f_{in}(\boldsymbol{x}) + \lambda \quad max(f_{jn}(\boldsymbol{x})), j \in \{1, \ldots, M\}, \tag{3}$$

*where $f_{in}$, $f_{jn}$ denotes a normalized objective function, $\boldsymbol{z}^* = \{z_1^*, \ldots, z_M^*\}, \boldsymbol{z}^{nad} = \{z_1^{nad}, \ldots, z_M^{nad}\}$ are ideal point and nadir point for the current non-dominated solutions, respectively.*

More detailed definitions about the background of MOO or MaOO are available in Appendix A. All non-$\lambda$-dominated solutions in $S_A$ are selected as reference points $S_{RP}$. There are two reasons to introduce the definition of $\lambda$-dominance:

- The $\lambda$-dominance can smoothen the original PF by excluding dominance resistant solutions (DRSs) Hanne (1999); Wang et al. (2018). DRSs are solutions that are best or close to best on one or several objectives but extremely poor on at least one of the remaining objectives. Such a solution is apparently not desirable but may be regarded as one of the best solutions since there may not exist any other solutions dominating it in the solution set.

- Second, $\lambda$-dominance can eliminate some similar non-dominated solutions from the Pareto set, which can be used to adjust the size of Pareto set. When $M$ is large, it is possible that a majority of past evaluated samples are non-dominated to each other. To balance the number of reference points and remaining samples, we introduce the dominance coefficient $\lambda$ to sightly reduce the ratio of reference points in $S_A$. This alleviates the situation of extreme imbalance of samples in different ordinal levels (see the division of ordinal levels below).

**Computation of Relation Values.** To quantify ordinal relation values, we first calculate extension coefficients $ec(\boldsymbol{x})$ for each $\boldsymbol{x} \in S_A$. $ec(\boldsymbol{x})$ is defined as the minimal coefficient $ec \geq 1$ to make a solution $\boldsymbol{x}$ non-$\lambda$-dominated to all solutions $\boldsymbol{x}'$ in the extended reference:

$$ec(\boldsymbol{x}) = \arg \min_{ec \geq 1} \nexists \boldsymbol{x}' \in S_{RP} : (\boldsymbol{x}' * ec) \prec_\lambda \boldsymbol{x}. \tag{4}$$

Although extension coefficient $ec(\boldsymbol{x})$ quantifies the distance between a solution $\boldsymbol{x}$ and reference $S_{RP}$, it has not been used to train the ordinal regression-based surrogate directly. To generate a stable ordinal regression-based surrogate, solutions in $S_A$ are divided into $N_o = max(n_o, |S_A|/|S_{RP}|)$ ordinal levels, where $n_o$ is a pre-defined parameter denoting the minimal number of ordinal levels. The solutions in $S_{RP}$ are classified into the non-dominated ordinal level, thus the relation value $v_1 = 1.0$ is assigned to them. Remaining solutions in $S_A$ are sorted by their extension coefficients $ec(\boldsymbol{x})$ and then divided into $N_o$-1 ordinal levels uniformly. The relation value $v_i = 1 - \frac{i-1}{N_o-1}$ will be assigned to the solutions $\boldsymbol{x}$ in the $i^{th}$ ordinal level. Lastly, relation values serve as radial coordinates and a Kriging model is employed to approximate them.

### 3.2.2 ARTIFICIAL CLUSTERING-BASED ORDINAL RELATIONS.

When the number of objectives $M$ is large, most evaluated solutions in archive $S_A$ could be non-dominated solutions, indicating that these solutions will be divided into the same non-dominated ordinal level and thus treated as reference points $S_{RP}$. This is harmful to the ordinal surrogate modeling due to the extreme imbalance between the numbers of training samples in different ordinal levels. To reduce the ratio of $S_{RP}$, we use a clustering method to generate $n\_clusters$ clusters for $S_{RP}$, where $n\_clusters$ is the half of the size of $S_{RP}$. All solutions $\boldsymbol{x} \in S_{RP}$ are mapped to the closest cluster centers. The solutions with the shortest projection on each cluster center will be selected as the new $S_{RP}$, while the remaining solutions will be moved to the next ordinal level. Such artificial ordinal relations greatly reduce the ratio of $S_{RP}$ in $S_A$. In LORA-MaOO, we set a ratio threshold $rp\_ratio$ for $S_{RP}$, once the ratio of $S_{RP}$ is larger than $rp\_ratio$, artificial ordinal relations will be generated for surrogate modeling. Details are available in Appendix C, Alg. 2 and Fig. 5.

### 3.2.3 SURROGATES FOR ANGULAR COORDINATES.

Given a solution $\boldsymbol{x} \in S_A$ with Cartesian coordinates $(f_1(\boldsymbol{x}), \ldots, f_M(\boldsymbol{x}))$, The angular coordinates of solution $\boldsymbol{x}$ are transformed with the following rules:

$$\varphi_i = arccos \frac{f_i(\boldsymbol{x}) - z_i^*}{\sqrt{(f_i(\boldsymbol{x}) - z_i^*)^2 + \cdots + (f_M(\boldsymbol{x}) - z_M^*)^2}}, i = 1, \ldots, M - 1, \qquad (5)$$

where $\boldsymbol{z}^*$ is the ideal point. The resulting angular coordinates $(\varphi_1, \ldots, \varphi_{M-1})$ are used to fit $M-1$ regression-based surrogates separately. In LORA-MaOO, we use the Kriging model to approximate angular coordinates. The introduction and usage of Kriging model is given in Appendix B.

### 3.3 SAMPLING: SEARCH AND SELECTION

In this subsection, we describe how to use surrogate $h_o$ to search for candidate solutions and how to use surrogates $h_o$ and $h_{ai}$ to select optimal ones from candidate solutions for expensive evaluations.

### 3.3.1 SEARCH: GENERATION OF CANDIDATE SOLUTIONS.

An advantage of LORA-MaOO is that it searches for candidate solutions on ordinal surrogate $h_o$ only, leaving all angular surrogates $h_{ai}$ idle in this search procedure. This saves a lot of time from predicting with all surrogates. LORA-MaOO employs an optimizer (e.g. PSO Eberhart & Kennedy (1995)) to generate a population of candidate solutions $P$ (Detailed pseudo-code is available in Appendix C, Alg. 3). The initial population for optimization search consists of two parts. The first half initial solutions are generated randomly from the decision space, while the remaining initial solutions are mutants of current reference points $S_{RP}$. To ensure the diversity of initial candidate solutions, a KNN clustering method is applied to divide $S_{RP}$ into several different clusters, from each cluster, an equal number of mutants are generated as initial candidate solutions. The global optimal population $P$ produced by PSO is the candidate solutions for further environmental selection.

### 3.3.2 SELECTION CRITERIA.

To take both convergence and diversity into consideration, in each iteration, LORA-MaOO selects two optimal candidate solutions $\boldsymbol{x}_1^*, \boldsymbol{x}_2^*$ from $P$ for objective function evaluations. $\boldsymbol{x}_1^*, \boldsymbol{x}_2^*$ are sampled on the basis of convergence and diversity, respectively.

**Convergence Criterion** for environmental selection is the expected improvement (EI) Emmerich et al. (2006) of ordinal values, which is similar to many MOBO methods Knowles (2006); Zhang et al. (2010). Since the output of our ordinal surrogate $h_o(\boldsymbol{x})$ is an 1-D numerical value, the solution with maximal 1-D EI in $P$ is selected as $\boldsymbol{x}_1^*$.
**Diversity Criterion** to sample $\boldsymbol{x}_2^*$ from $P$ is defined as angles $d_{ang}$ between candidate solutions and reference points $S_{RP}$. Firstly, the minimal degree between each candidate solution and $S_{RP}$ is measured. Among these minimal degrees $md_{ang}$, the solution with MAX($md_{ang}$) is selected as $\boldsymbol{x}_2^*$ (Detailed pseudo-code is available in Appendix C, Alg. 4).

## 4 EXPERIMENTS

To evaluate the optimization performance of LORA-MaOO on expensive MaOOPs, we conduct experiments to compare LORA-MaOO with other SAEAs on different MaOOPs, including a series of scalable multi-/many-objective benchmark optimization problems DTLZ Deb et al. (2005), WFG Huband et al. (2006), and a real-world network architecture search (NAS) problem.

### 4.1 EXPERIMENTAL SETUPS

**Optimization Problem Setup.** To ensure a fair comparison, the following optimization problem setup is the same as the setup that has been widely used in the literature Chugh et al. (2016); Pan et al. (2018); Song et al. (2021); Hao et al. (2022). In our experiments, initial datasets of size $FE_{init}$ = 11 $D$ - 1 are used to initialize surrogates, while the maximum number of allowed evaluations $FE_{max}$ is 300. The statistical results are obtained from 30 independent runs. For each run, different

comparison algorithms share the same initial dataset.

**Comparison Algorithms.** We compare LORA-MaOO with 6 state-of-the-art SAEAs, some of them also known as MOBO methods. These comparison algorithms can be classified into three categories:

- Regression-based optimization methods: ParEGO Knowles (2006), K-RVEA Chugh et al. (2016), and KTA2 Song et al. (2021). ParEGO is a classic regression-based SAEA and also a MOBO, which serves as a baseline. K-RVEA is a typical SAEA which uses reference vector to guide the diversity maintenance. KTA2 is a newly proposed algorithm to use an independent archive to keep solution diversity.

- Classification-based optimization methods: CSEA Pan et al. (2018), REMO Hao et al. (2022). CSEA is a classic classification-based SAEA which serves as a baseline. REMO is a newly proposed SAEA which represents the state-of-the-art performance of classification-based SAEAs.

- Ordinal-regression-based optimization method: OREA Yu et al. (2019) is a new category of SAEA that is different from common regression-based and classification-based SAEAs. We compare with it since it is directly related to our radial surrogate.

Note that some classic SAEAs and MOBO methods such as MOEA/D-EGO Zhang et al. (2010) and CPS-MOEA Zhang et al. (2015) are not compared in our experiments as they failed to outperform other comparison algorithms on any DTLZ problem Hao et al. (2022). Some HV-based MOBO methods are not compared as they failed to solve many objectives.

**Parameter Setup.** For the surrogate modeling, the Kriging models used in all comparison algorithms are implemented using DACE Sacks et al. (1989), just as Knowles (2006) suggested. For regression-based Kriging surrogates, the range of hyper-parameter $\theta \in [10^{-5}, 100]$. And for the neural networks in CSEA and REMO, the parameters are the same as suggested in the literature. In the sampling strategy, the mutation operator used to initialize candidate solutions is polynomial mutation Deb & Goyal (1996), the mutation probability $p_m = 1/d$ and mutation index $\eta_m = 20$, as recommended in Song et al. (2021); Hao et al. (2022). The size of offspring population is 100. The settings of the PSO optimizer are the range of hyper-parameter in the ordinal-regression-based surrogate are the same as suggested in Yu et al. (2019).

For the specific parameters exist in LORA-MaOO, such as the dominance coefficient $\lambda$ and the threshold ratio of reference points to introduce clustering-based ordinal relations $rp\_ratio$. As there is no relevant study in the literature for their setups, we conducted ablation studies to investigate the effect of these parameters on the performance of LORA-MaOO. The results are summarized in Section 4.2 and reported in Appendix F. The source code of LORA-MaOO [2] will be available online.

**Performance Indicator.** To have a comprehensive estimation of optimization performance, we use three different performance indicators in our experiments: The inverted generational distance (IGD) Bosman & Thierens (2003), the inverted generational distance plus (IGD+) Ishibuchi et al. (2015), and the Hypervolume (HV) Zitzler & Thiele (1998). IGD and IGD+ use a set of truth Pareto front to measure the quality of a set of non-dominated solutions in terms of convergence and diversity. A smaller IGD or IGD+ value indicates better MaOO performance. HV uses a reference point to calculate the area covered by a set of non-dominated solutions, a large HV value is preferable to MaOO. See Appendix D for details and setups about performance indicators.

### 4.2 ABLATION STUDIES ON PARAMETERS AND ALGORITHM COMPONENTS

We conduct ablation studies on DTLZ and WFG benchmark problems with $D = 10$ variables and $M=\{3, 6, 10\}$ objectives. LHS McKay et al. (2000) is used to sample initial dataset. The effects of four parameters are investigated: The minimal number of ordinal levels $n_o$, the dominance coefficient $\lambda$, the ratio threshold of reference points $rp\_ratio$, and the clustering number for reproduction $n_c$. Meanwhile, the contribution of three algorithm components are demonstrated: The $\lambda$-dominance, the artificial relations, and the clustering-based initialization. Three representative results obtained on the WFG5 problem with 3 and 10 objectives are depicted in Fig. 1. Complete results, statistical analysis of ablation studies, and in-depth analysis of component contributions are reported in Appendix F.

---

[2]The link of code and data will be released here once the paper is accepted.

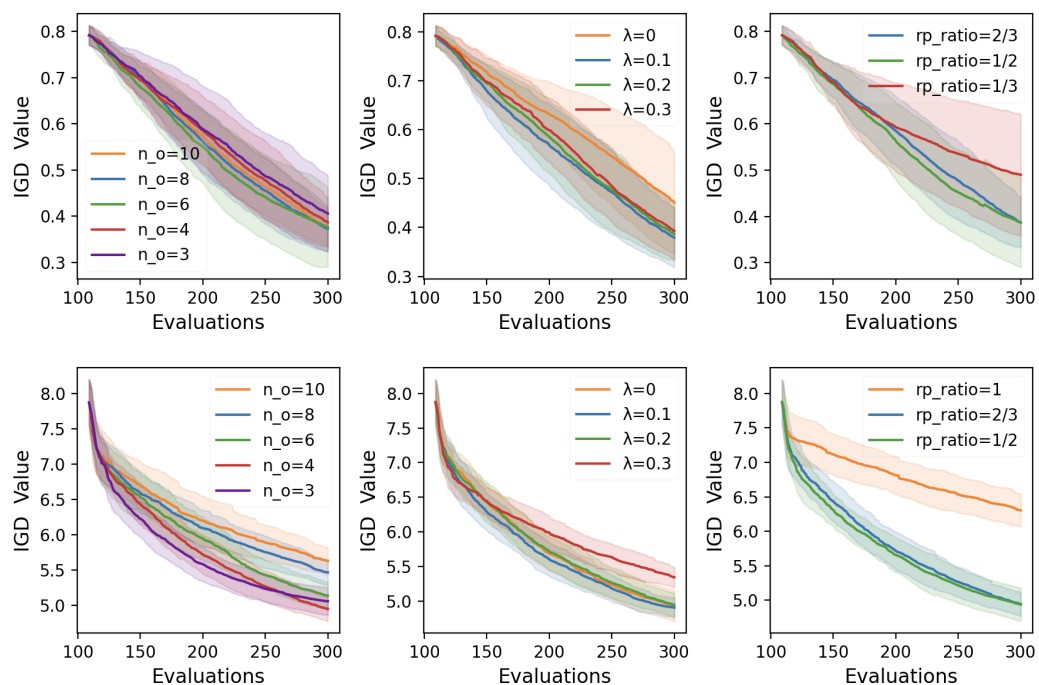

Figure 1: IGD curves averaged over 15 runs on WFG5 problem instances for LORA-MaOO with different parameter setups. Upper: 10 variables, 3 objectives. Lower: 10 variables, 10 objectives. Shaded area is $\pm$ std of the mean.

### 4.3 OPTIMIZATION ON BENCHMARK PROBLEMS

The optimization performance of LORA-MaOO is evaluated on DTLZ and WFG benchmark problems with $D = 10$ variables and $M=\{3, 4, 6, 8, 10\}$ objectives. The IGD values obtained on DTLZ problems with different $M$ are reported in Table 1. It shows that LORA-MaOO achieves the best optimization results among all the comparison algorithms in terms of IGD values, followed by KTA2 and KRVEA. The IGD values obtained on the WFG problems, the IGD+ and HV results, and the results obtained under different scales ($D= 5$ or $20$) are reported in Appendix H. A consistent result can be concluded from the IGD+ and HV values. The results on the 3- and 10-objective problems are plotted in Fig. 2.

### 4.4 REAL-WORLD NETWORK ARCHITECTURE SEARCH PROBLEMS

Further comparison is conducted on two real-world network architecture search (NAS) problems, the best three algorithms listed in Table 1 are compared: LORA-MaOO, KTA2, and KRVEA. The NAS problems tested are two different NASbench201 problems implemented in EvoXBench Lu et al. (2023), the first problem has 6 variables and 5 objectives, the second problem has 6 variables and 8 objectives. Details of two NAS problems are provided in Appendix E. Considering NASbench201 problems are real-world applications and we do not know their exact PF, we use HV to evaluate optimization performance since HV can be calculated without the exact PF. In practice, $log(HV_{\text{diff}})$ is employed to amplify the visual difference of the obtained HV values:

$$log(HV_{\text{diff}}) = log(HV_{\text{max}} - HV)$$

where $HV_{\text{max}}$ is the maximal HV value on the given NAS problem that is provided in EvoXBench.

Fig. 3 plots the results. As can be seen in the figure, LORA-MaOO outperforms KTA2 and KRVEA on two NAS problems. When $M$ is 5, although KTA2 and KRVEA have quicker convergence rate than LORA-MaOO at the beginning of the optimization, both of them slow down their convergence speed as the number of evaluations increases. In comparison, when $M$ is 8, KRVEA and LORA-MaOO have similar convergence rate and both of them are quicker than KTA2's convergence rate.

Table 1: Statistical results of the IGD value obtained by the comparison algorithms on the 35 DTLZ optimization problems over 30 runs. Symbols '+', '≈', '−' denote LORA-MaOO is statistically significantly superior to, equivalent to, and inferior to the compared algorithms in the Wilcoxon rank sum test (significance level is 0.05), respectively. The last three rows are the total win/tie/loss results on DTLZ, WFG, and both of them, respectively.

| Problems | M | ParEGO | KRVEA | KTA2 | CSEA | REMO | OREA | LORA-MaOO (ours) |
|---|---|---|---|---|---|---|---|---|
| DTLZ1 | 3 | 5.98e+1(3.81e+0)+ | 8.88e+1(2.16e+1)+ | 4.75e+1(1.55e+1)≈ | 6.30e+1(1.69e+1)+ | 5.06e+1(1.49e+1)+ | 4.44e+1(1.38e+1)≈ | 4.35e+1(1.80e+1) |
| | 4 | 4.68e+1(3.71e+0)+ | 6.45e+1(1.47e+1)+ | 4.08e+1(1.60e+1)+ | 3.69e+1(1.08e+1)≈ | 3.92e+1(1.11e+1)≈ | 3.80e+1(1.23e+1)≈ | 4.06e+1(1.34e+1) |
| | 6 | 3.04e+1(2.74e+0)+ | 3.22e+1(7.66e+0)+ | 2.03e+1(8.12e+0)+ | 1.56e+1(4.96e+0)≈ | 1.22e+1(4.65e+0)− | 1.74e+1(3.98e+0)≈ | 1.58e+1(6.17e+0) |
| | 8 | 1.23e+1(2.99e+0)+ | 8.52e+0(2.97e+0)+ | 4.54e+0(2.66e+0)≈ | 5.08e+0(2.47e+0)≈ | 3.33e+0(1.93e+0)≈ | 5.87e+0(2.91e+0)+ | 3.83e+0(2.35e+0) |
| | 10 | 4.37e-1(1.63e-1)+ | 3.32e-1(9.91e-2)+ | 3.00e-1(8.76e-2)+ | 2.90e-1(7.13e-2)+ | 2.42e-1(6.97e-2)≈ | 2.58e-1(6.33e-2)≈ | 2.31e-1(3.89e-2) |
| DTLZ2 | 3 | 3.38e-1(2.84e-2)+ | 1.32e-1(2.77e-2)+ | 6.17e-3(3.13e-3)+ | 2.26e-1(2.61e-2)+ | 1.65e-1(2.18e-2)+ | 8.59e-2(8.51e-3)+ | 6.19e-3(3.48e-3) |
| | 4 | 4.23e-1(2.79e-2)+ | 2.06e-1(2.95e-2)+ | 1.41e-1(5.45e-3)+ | 2.92e-1(1.89e-2)+ | 2.43e-1(2.33e-2)+ | 1.83e-1(1.37e-2)+ | 1.38e-1(9.86e-3) |
| | 6 | 5.53e-1(2.17e-2)+ | 2.61e-1(1.20e-2)+ | 3.24e-1(2.63e-2)+ | 4.42e-1(3.37e-2)+ | 3.77e-1(3.16e-2)+ | 3.96e-1(2.57e-2)+ | 2.67e-1(8.78e-3) |
| | 8 | 6.53e-1(1.86e-2)+ | 4.19e-1(2.65e-2)+ | 4.44e-1(1.86e-2)+ | 5.95e-1(2.77e-2)+ | 5.10e-1(3.90e-2)+ | 5.56e-1(2.19e-2)+ | 3.80e-1(1.46e-2) |
| | 10 | 6.95e-1(2.23e-2)+ | 5.92e-1(4.25e-2)+ | 4.50e-1(1.00e-2)≈ | 6.76e-1(2.52e-2)+ | 5.85e-1(3.72e-2)+ | 6.55e-1(2.66e-2)+ | 4.54e-1(1.41e-2) |
| DTLZ3 | 3 | 1.66e+2(1.31e+1)+ | 2.43e+2(4.61e+1)+ | 1.52e+2(4.73e+1)≈ | 1.62e+2(4.84e+1)≈ | 1.49e+2(3.88e+1)≈ | 1.26e+2(3.18e+1)− | 1.57e+2(3.83e+1) |
| | 4 | 1.42e+2(1.57e+1)+ | 1.83e+2(4.00e+1)+ | 1.18e+2(3.49e+1)+ | 1.29e+2(3.58e+1)+ | 1.16e+2(3.00e+1)+ | 1.22e+2(4.13e+1)+ | 1.25e+2(4.20e+1) |
| | 6 | 9.17e+1(1.59e+1)+ | 1.06e+2(2.96e+1)+ | 6.65e+1(2.63e+1)+ | 5.27e+1(1.56e+1)+ | 5.23e+1(1.71e+1)+ | 5.24e+1(1.68e+1)≈ | 5.96e+1(2.05e+1) |
| | 8 | 4.13e+1(9.84e+0)+ | 2.96e+1(1.15e+1)+ | 1.74e+1(1.10e+1)+ | 1.01e+0(2.45e-1)+ | 9.53e-1(2.74e-1)+ | 8.77e-1(1.08e-1)+ | 8.14e-1(1.33e-1) |
| | 10 | 1.36e+0(3.15e-1)+ | 1.23e+0(4.27e-1)+ | 9.95e-1(2.25e-1)+ | 1.01e+0(2.45e-1)+ | 9.53e-1(2.74e-1)+ | 8.77e-1(1.08e-1)+ | 8.14e-1(1.33e-1) |
| DTLZ4 | 3 | 6.70e-1(7.61e-2)+ | 3.32e-1(1.11e-1)+ | 3.49e-1(1.09e-1)+ | 4.62e-1(1.36e-1)+ | 2.31e-1(1.15e-1)+ | 2.39e-1(1.65e-1)+ | 1.89e-1(2.34e-1) |
| | 4 | 7.18e-1(6.40e-2)+ | 4.07e-1(8.73e-2)+ | 4.77e-1(9.70e-2)+ | 4.31e-1(6.36e-2)+ | 3.36e-1(7.02e-2)≈ | 3.45e-1(1.52e-1)≈ | 3.48e-1(1.60e-1) |
| | 6 | 7.06e-1(3.07e-2)+ | 5.04e-1(5.42e-2)+ | 6.05e-1(8.43e-2)+ | 4.94e-1(4.55e-2)+ | 4.97e-1(4.95e-2)+ | 4.47e-1(4.89e-2)≈ | 4.55e-1(6.53e-2) |
| | 8 | 6.81e-1(1.48e-2)+ | 5.49e-1(3.42e-2)+ | 6.24e-1(5.48e-2)+ | 5.85e-1(4.20e-2)+ | 6.16e-1(4.03e-2)+ | 5.29e-1(3.79e-2)≈ | 5.32e-1(2.36e-2) |
| | 10 | 6.77e-1(1.26e-2)+ | 6.07e-1(2.42e-2)+ | 6.36e-1(3.58e-2)+ | 6.38e-1(2.38e-2)+ | 6.71e-1(2.69e-2)+ | 5.90e-1(1.94e-2)≈ | 5.90e-1(2.51e-2) |
| DTLZ5 | 3 | 2.16e-1(4.45e-2)+ | 1.19e-1(3.38e-2)+ | 1.34e-2(2.83e-3)+ | 1.18e-1(2.56e-2)+ | 7.36e-2(2.03e-2)+ | 2.02e-2(4.77e-3)+ | 1.26e-2(2.55e-3) |
| | 4 | 1.89e-1(3.70e-2)+ | 7.05e-2(2.25e-2)+ | 4.24e-2(8.84e-3)+ | 1.16e-1(2.23e-2)+ | 9.02e-2(2.48e-2)+ | 3.48e-2(7.82e-3)+ | 2.85e-2(9.37e-3) |
| | 6 | 1.41e-1(2.32e-2)+ | 3.53e-2(1.02e-2)− | 8.87e-2(1.91e-2)+ | 7.72e-2(2.57e-2)+ | 5.53e-2(1.90e-2)+ | 4.62e-2(1.50e-2)+ | 4.26e-2(1.11e-2) |
| | 8 | 7.72e-2(1.22e-2)+ | 1.99e-2(4.92e-3)− | 6.43e-2(8.60e-3)+ | 3.81e-2(1.03e-2)+ | 3.10e-2(7.33e-3)+ | 2.59e-2(6.96e-3)− | 2.84e-2(4.88e-3) |
| | 10 | 2.25e-2(1.87e-3)+ | 1.25e-2(1.90e-3)+ | 2.04e-2(2.55e-3)+ | 1.27e-2(1.46e-3)+ | 9.35e-3(2.00e-3)− | 1.03e-2(1.62e-3)≈ | 1.06e-2(2.36e-3) |
| DTLZ6 | 3 | 3.15e-1(1.62e-1)+ | 3.06e+0(5.21e-1)+ | 1.83e+0(4.37e-1)+ | 4.86e+0(6.30e-1)+ | 4.27e+0(5.49e-1)+ | 3.09e-1(3.99e-1)+ | 1.18e-1(1.57e-1) |
| | 4 | 3.56e-1(2.12e-1)+ | 2.46e+0(3.84e-1)+ | 1.85e+0(5.06e-1)+ | 5.13e+0(4.23e-1)+ | 4.08e+0(6.16e-1)+ | 1.43e+0(8.89e-1)+ | 3.29e-1(2.21e-1) |
| | 6 | 2.66e-1(1.37e-1)− | 1.36e+0(2.73e-1)+ | 1.51e+0(5.85e-1)+ | 3.15e+0(4.35e-1)+ | 2.33e+0(5.70e-1)+ | 2.05e+0(6.16e-1)+ | 9.89e-1(1.02e+0) |
| | 8 | 1.61e-1(6.17e-2)≈ | 5.28e-1(1.50e-1)+ | 8.64e-1(3.88e-1)+ | 1.56e+0(4.28e-1)+ | 9.64e-1(4.38e-1)+ | 1.06e+0(3.95e-1)+ | 3.56e-1(4.31e-1) |
| | 10 | 1.72e-1(1.45e-1)+ | 7.73e-2(3.13e-2)+ | 1.01e+0(4.97e-2)+ | 2.09e-1(2.28e-1)+ | 7.91e-2(1.11e-1)+ | 1.50e-1(1.37e-2)+ | 7.05e-2(3.25e-2) |
| DTLZ7 | 3 | 2.45e-1(4.80e-2)+ | 1.35e-1(2.37e-2)≈ | 2.19e-1(2.40e-1)− | 1.75e-1(6.32e-1)+ | 1.27e+0(5.65e-1)+ | 2.73e-1(1.58e-1)+ | 2.01e-1(1.93e-1) |
| | 4 | 6.59e-1(1.02e-1)+ | 5.13e-1(7.61e-2)+ | 3.73e-1(1.68e-1)+ | 2.94e+0(6.59e-1)+ | 2.06e+0(7.31e-1)+ | 8.92e-1(4.27e-1)+ | 4.20e-1(2.21e-1) |
| | 6 | 1.21e+0(1.58e-1)− | 6.04e-1(4.57e-2)− | 6.46e-1(1.68e-1)− | 4.92e+0(9.92e-1)+ | 3.09e+0(6.71e-1)+ | 4.03e+0(1.84e+0)+ | 1.71e+0(6.54e-1) |
| | 8 | 1.45e+0(1.24e-1)− | 8.71e-1(7.01e-2)− | 1.02e+0(1.65e-1)− | 6.12e+0(1.85e+0)+ | 3.82e+0(5.39e-1)+ | 4.55e+0(2.63e+0)+ | 2.44e+0(6.78e-1) |
| | 10 | 1.67e+0(1.24e+1)+ | 1.12e+0(4.25e-2)− | 1.30e+0(2.04e-1)≈ | 1.99e+0(3.05e-1)+ | 1.99e+0(3.36e-1)+ | 1.63e+0(2.42e-1)+ | 1.34e+0(9.19e-2) |
| +/≈/− | on DTLZ | 30/2/3 | 27/3/5 | 19/13/3 | 28/7/0 | 23/10/2 | 20/13/2 | |
| +/≈/− | on WFG | 39/4/2 | 21/10/14 | 23/6/16 | 41/1/3 | 38/3/4 | 43/1/1 | |
| +/≈/− | on both | 69/6/5 | 48/13/19 | 42/19/19 | 69/8/3 | 61/13/6 | 63/14/3 | |

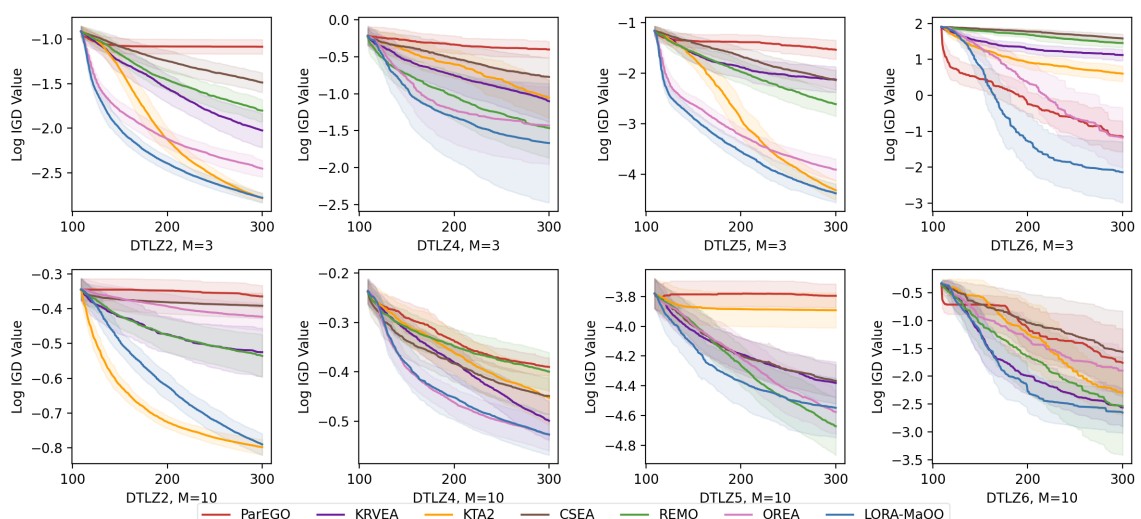

Figure 2: IGD(log) curves averaged over 30 runs on the DTLZ problems for the comparison algorithms (shaded area is ± std of the mean). More figures are displayed in Appendices G and H.

Particularly, KTA2 is trapped on local optima and thus fails to reach better results in two NAS problems. In comparison, LORA-MaOO reaches better NAS results on two problems when the evaluation number is larger than 250 and 150, respectively.

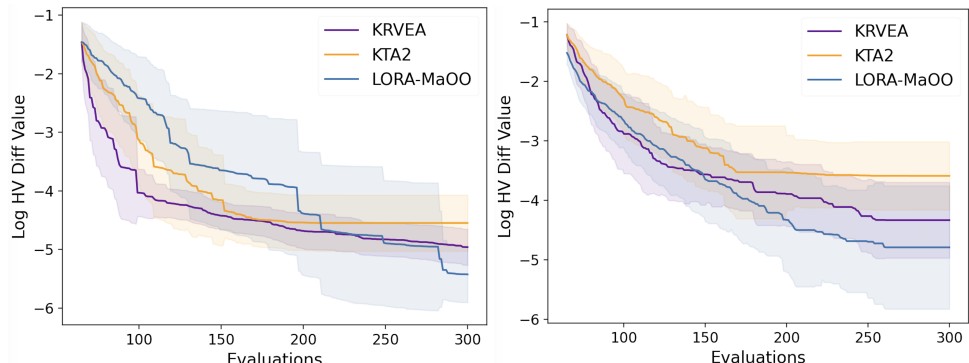

Figure 3: $Log(HV_{\text{diff}})$ curves averaged over 30 runs on the NAS problem for the comparison algorithms. **Left**: $M = 5$ objectives. **Right**: $M = 8$ objectives.

## 4.5 Runtime Comparison

We compare the runtime on benchmark problems for all the comparison algorithms to investigate the relation between their optimization efficiency and the number of objectives $M$.

Fig. 4 illustrates how the runtime of each comparison algorithm varies as the $M$ increases. It can be observed that the runtime of KTA2 increases exactly in the same rate as $M$ increases. In comparison, the runtime of LORA-MaOO increases slightly when $M$ increases. This demonstrates that using angular surrogates only at the end of environmental selection process is beneficial to the optimization efficiency of LORA-MaOO. In addition, the runtimes of ParEGO, CSEA, REMO, and OREA do not increase significantly with $M$ since they do not maintain specific surrogates to manage the diversity of non-dominated solutions. Consequently, their overall performance reported in Table 1 is not desirable. Overall, LORA-MaOO finds a good trade-off between optimization efficiency and optimization results.

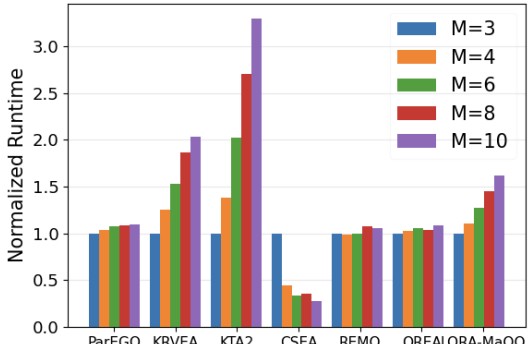

Figure 4: Comparison of runtime averaged over 30 runs on benchmark problems $D = 10$ variables and $M = 3, 4, 6, 8,$ and 10 objectives for the comparison algorithms. For each algorithm, its runtimes are normalized by the runtime it costed on 3-objective problems.

## 5 Conclusion

In this paper, we propose an efficient MaOO method, LORA-MaOO, to solve expensive MaOOPs. Different from existing surrogate modeling approaches, our LORA-MaOO learns surrogate models from ordinal relations and spherical coordinates. LORA-MaOO provides an insight of handling convergence and diversity with different subsets of surrogates, showing a more flexible way to use surrogates during model-based optimization. Particularly, only one ordinal surrogate is used in the model-based search, which hugely improve the efficiency of optimization. Our empirical studies have demonstrated that our LORA-MaOO significantly outperforms other state-of-the-art efficient MaOO methods, including SAEAs and MOBO methods.

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

# A  BACKGROUND OF MANY-OBJECTIVE OPTIMIZATION

We consider minimization problems and many-objective optimization problems (MaOOPs) can be formulated as follows:

**Definition 2.**  *(Expensive Many-Objective Optimization Problem)*
*Given $M$ expensive objective functions $f_1, \ldots, f_M$ and an evaluation budget $FE_{max}$, obtain the Pareto set for the following many-objective optimization problem:*

$$\operatorname*{argmin}_{\boldsymbol{x} \in X} f(\boldsymbol{x}) = (f_1(\boldsymbol{x}), \ldots, f_M(\boldsymbol{x}))$$

*where $X \subseteq \mathbb{R}^D$ is the decision space of the problem.*

The Pareto set is defined through the following definitions: Pareto set and Pareto front are defined as follows:

**Definition 3.**  *Pareto dominance:*
*A solution $\boldsymbol{x}^1$ is said to dominate another solution $\boldsymbol{x}^2$ (denoted by $\boldsymbol{x}^1 \prec \boldsymbol{x}^2$) if and only if:*

$$\forall k \in \{1, 2, \ldots, M\} : f_k(\boldsymbol{x}^1) \leq f_k(\boldsymbol{x}^2) \wedge$$

$$\exists k \in \{1, 2, \ldots, M\} : f_k(\boldsymbol{x}^1) < f_k(\boldsymbol{x}^2)$$

**Definition 4.**  *Non-dominated solution:*
*A non-dominated solution $\boldsymbol{x}^\star$ in the decision space $X$ is a solution that cannot be dominated by any other solutions in $X$:*

$$\nexists \boldsymbol{x} \in X : \boldsymbol{x} \prec \boldsymbol{x}^\star$$

**Definition 5.**  *Pareto set:*
*Pareto set $S_{ps}$ is the set of all non-dominated solutions in the decision space $X$:*

$$S_{ps} = \{\boldsymbol{x}^\star \in X | \nexists \boldsymbol{x} \in X : \boldsymbol{x} \prec \boldsymbol{x}^\star\}$$

**Definition 6.**  *Pareto front:*
*Pareto front $S_{pf}$ is the corresponding unique set of the Pareto set in the objective space:*

$$S_{pf} = \{f(\boldsymbol{x}) | \boldsymbol{x} \in S_{ps}\}$$

# B  KRIGING MODEL

Kriging model, also known as Gaussian process model Jones et al. (1998) or design and analysis of computer experiments (DACE) model Sacks et al. (1989), is a stochastic process model used to approximate an unknown objective function. LORA-MaOO uses Kriging models to implement angular surrogates and the radial surrogate, to avoid potential confusion and help the understanding of our algorithm, the working mechanism of the Kriging model is described below.

A common way to approximate an unknown objective function with $n$ observations is linear regression:

$$y(\boldsymbol{x}^i) = \sum_{k=1}^{N} \beta_k f_k(\boldsymbol{x}^i) + \epsilon^i, \tag{6}$$

where $\boldsymbol{x}^i$ is the $i^{th}$ sample point observed from the objective function. $f_k(\boldsymbol{x}^i)$, $\beta_k$ are a linear or nonlinear function of $\boldsymbol{x}^i$ and its coefficient, respectively. $N$ is the number of functions $f(\boldsymbol{x})$. $\epsilon^i$ is an independent error term, which is normally distributed with mean zero and variance $\sigma^2$.

However, a stochastic process model such as Kriging does not assume that the error terms $\epsilon$ are independent. Hence, an error term $\epsilon^i$ is rewritten as $\epsilon(\boldsymbol{x}^i)$. Moreover, these error terms are assumed to be related or correlated to each other. The correlation between two error terms $\epsilon(\boldsymbol{x}^i)$ and $\epsilon(\boldsymbol{x}^j)$ is inversely proportional to the distance between the corresponding points Jones et al. (1998). The correlation function in the Kriging model is defined as:

$$Corr(\epsilon(\boldsymbol{x}^i), \epsilon(\boldsymbol{x}^j)) = exp[-dis(\boldsymbol{x}^i, \boldsymbol{x}^j)], \tag{7}$$

where the distance between two points $\boldsymbol{x}^i$ and $\boldsymbol{x}^j$ are measured using the special weighted distance formula shown below:

$$dis(\boldsymbol{x}^i, \boldsymbol{x}^j) = \sum_{k=1}^{D} \theta_i |x_k^i - x_k^j|^{p_k}, \tag{8}$$

where $D$ is the number of decision variables, $\boldsymbol{\theta} \in \mathbb{R}_{\geq 0}^D$ and $\mathbf{p} \in [1, 2]^D$ are parameters of the Kriging model. It can be seen from Eq.(7) that the correlation is ranged within $(0, 1]$ and is increasing as the distance between two points decreases. Particularly, in Eq.(8), the parameter $\theta_k$ can be explained as the importance of the decision variable $x_k$, and the parameter $p_k$ can be interpreted as the smoothness of the correlation function in the $k^{th}$ coordinate direction.

Due to the effectiveness of correlation modelling, the regression model in Eq.(6) can be simplified without degrading modelling performance Jones et al. (1998). Clearly, all regression terms are replaced with a constant term, thus the Kriging regression model can be rewritten as follows:

$$y(\boldsymbol{x}^i) = \mu + \epsilon(\boldsymbol{x}^i), \tag{9}$$

where $\mu$ is the mean of this stochastic process, $\epsilon(\boldsymbol{x}^i) \sim \mathcal{N}(0, \sigma^2)$.

### B.1 TRAINING THE KRIGING MODEL

To train the Kriging model and estimate the parameters $\boldsymbol{\theta}, \mathbf{p}$ in Eq.(8), the following likelihood function is maximised:

$$\frac{1}{(2\pi)^{n/2}(\sigma^2)^{n/2}|\mathbf{R}|^{1/2}} exp[-\frac{(\mathbf{y} - \mathbf{1}\mu)^T \mathbf{R}^{-1}(\mathbf{y} - \mathbf{1}\mu)}{2\sigma^2}], \tag{10}$$

where $|\mathbf{R}|$ is the determinant of the correlation matrix, each element in the matrix is obtained using Eq.(7). $\mathbf{y}$ is the $n$-dimensional vector of dependent variables that observed from the objective function. The mean value $\mu$ and variance $\sigma^2$ in Eq.(9) and Eq.(10) can be estimated by:

$$\hat{\mu} = \frac{\mathbf{1}^T \mathbf{R}^{-1} \mathbf{y}}{\mathbf{1}^T \mathbf{R}^{-1} \mathbf{1}}, \tag{11}$$

$$\hat{\sigma} = \frac{1}{n}(\mathbf{y} - \mathbf{1}\hat{\mu})^T \mathbf{R}^{-1}(\mathbf{y} - \mathbf{1}\hat{\mu}). \tag{12}$$

### B.2 PREDICTION WITH THE KRIGING MODEL

For a new solution $\boldsymbol{x}^*$, the Kriging model predicts the approximation of $\hat{y}(\boldsymbol{x}^*)$ and the uncertainty $\hat{s}^2(\boldsymbol{x}^*)$ as follows:

$$\hat{y}(\boldsymbol{x}^*) = \hat{\mu} + \mathbf{r}' \mathbf{R}^{-1}(\mathbf{y} - \mathbf{1}\hat{\mu}), \tag{13}$$

$$\hat{s}^2(\boldsymbol{x}^*) = \hat{\sigma}^2(1 - \mathbf{r}' \mathbf{R}^{-1} \mathbf{r}), \tag{14}$$

where $\mathbf{r}$ is a $n$-dimensional vector of correlations between $\epsilon(\boldsymbol{x}^*)$ and the error terms at the training data, which can be calculated via Eq.(7).

Further details and a comprehensive description of the Kriging model and Gaussian Process can be found in Williams & Rasmussen (2006). In this paper, all regression-based Kriging models have $\boldsymbol{\theta} \in [10^{-5}, 100]^D$, $\mathbf{p} = 2^D$.

## C  ADDITIONAL DESCRIPTION OF LORA-MAOO

This section describes LORA-MaOO with more details.

### C.1  QUANTIFICATION OF ORDINAL RELATIONS

In order to learn the ordinal landscape of MaOOPs, we need to quantify the ordinal relations between solutions into numerical values. Alg. 2 illustrates the pseudocode of quantifying ordinal relations[3], it describes line 4 in Alg. 1 of the main file. It can be seen that Alg. 2 is mainly working on the

---

[3]Symbol '←' indicates the result of a function, Symbol '=' indicates an assignment operation.

---

**Algorithm 2** Quantify Ordinal Relations for LORA-MaOO

---

**Input:**

    $S_A$: Archive of evaluated solutions;

    $rp\_ratio$: Ratio threshold of reference points in $S_A$;

    $n_o$: Minimal number of ordinal levels.

**Procedure:**

1:  $S_{RP} \leftarrow$ Non-dominated solutions in $S_A$ that are non-$\lambda$-dominated to any other solution in $S_A$.

2:  Non-dominated level (The first ordinal level) $L_1 \leftarrow S_{RP}$.

3:  The number of non-dominated ordinal levels $n_{ndl} = 1$.

4:  Ratio of reference points $ratio = \frac{|S_{RP}|}{|S_A|}$.

5:  **if** $ratio > rp_{ratio}$ **then**

6:     $n_{ndl} = n_{ndl} + 1$.

    /* Add Artificial Ordinal Relations. */

7:     Divide $S_{RP}$ into $\frac{|S_{RP}|}{2}$ clusters via KNN clustering.

8:     For $x$ in each cluster, calculate the projection length of $x$ on the corresponding cluster center.

9:     $L_1 \leftarrow$ Solutions $x$ with the shortest projection on each cluster.

10:    $L_2 \leftarrow$ Remaining $\frac{|S_{RP}|}{2}$ solutions in $S_{RP}$.

11: **end if**

12: Calculate extension coefficient $ec(x)$ for all $x \in S_A$.

13: The number of ordinal levels $N_o = \max(n_o, \frac{|S_A|}{|S_{RP}|})$.

14: $L_i \leftarrow$ According to the order of $ec(x)$, uniformly divide solutions $x \in (S_A - S_{RP})$ into $N_o$ - $n_{ndl}$ levels.

15: Ordinal relation value $v_i = 1 - \frac{i-1}{N_o-1}$ for $x \in L_i$.

**Output:** An ordinal training set $S_o$ consisting of ordinal relation values $v_i$.

---

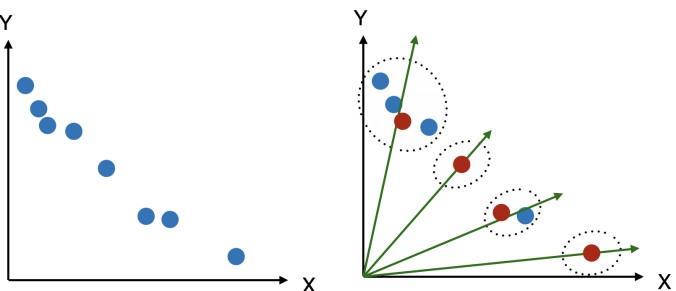

Figure 5: Illustration of artificial clustering-based ordinal relations. **Left**: Non-dominated solutions without artificial ordinal relations. **Right**: Non-dominated solutions with artificial ordinal relations. Red solutions are new non-dominated solutions in $L_1$, remaining blue solutions are moved to next ordinal level $L_2$. Dash circles are clusters, green vectors are cluster centers.

quantification of dominance-based ordinal relations. Artificial ordinal relations will not be added unless the ratio of reference points is larger than ratio threshold $rp_{ratio}$ (line 5).

An illustration of artificial clustering-based ordinal relations is given in Fig. 5. By using clustering methods, artificial ordinal relations are generated for training ordinal regression surrogates. Picking one solution from each cluster ensures the diversity of non-dominated solutions in the first ordinal level $L_1$. Meanwhile, the selection within each cluster is based on the projection length on cluster center, which is beneficial to the convergence of non-dominated solutions.

### C.2   GENERATION OF CANDIDATE SOLUTIONS

Algo. 3 gives the pseudocode of generating candidate solutions, it is the implementation of line 6 in Alg. 1 of the main file. In lines 1-9, a population $P_0$ is generated. Since reference points $S_{RP}$ are the optimal solutions in $S_A$ in terms of convergence, a half initial solutions are generated from $S_{RP}$

---

**Algorithm 3** Generation of candidate solutions in LORA-MaOO

**Input:**

$S_{RP}$: Reference points used in the ordinal regression;

$h_o$: Ordinal regression surrogate;

$n_c$: The number of clusters to initialize population $P$;

$|P|$: The size of population $P$;

$G_{max}$: The number of generations for reproduction.

**Procedure:**

1: $P_r \leftarrow$ Randomly sample $\frac{|P|}{2}$ solutions from the decision space.

2: Divide $S_{RP}$ into $n_c$ clusters via KNN clustering.

3: $P_c = \emptyset$.

4: **for** $i = 1$ to $n_c$ **do**

5:     $P_{ci} \leftarrow$ Randomly sample $\frac{|P|}{2n_c}$ solutions from $i^{th}$ cluster.

6:     $P_{ci} \leftarrow$ Mutation ( $P_{ci}$).

7:     $P_c = P_c \cup P_{ci}$.

8: **end for**

9: Initial population $P_0 = P_r \cup P_c$.

10: $h_o(P_0) \leftarrow$ Evaluate $P_0$ on ordinal surrogate $h_o$.

11: Global Optimal Population $P_{global} = P_0$.

12: **for** $i = 1$ to $G_{max}$ **do**

13:     $P_i \leftarrow$ PSO operation on $P_{i-1}$ and $P_{global}$.

14:     $h_o(P_i) \leftarrow$ Evaluate $P_i$ on ordinal surrogate $h_o$.

15:     Update $P_{global}$ using $h_o(P_i)$ and $h_o(P_{i-1})$.

16: **end for**

**Output:** A generation of candidate solutions $P = P_{global}$.

---

(lines 2-8). To obtain a diverse subset of $S_{RP}$, LORA-MaOO divides $S_{RP}$ into $n_c$ clusters before sampling solutions (line 2). The remaining initial solutions are sampled from the decision space randomly, ensuring the diversity of initial population and thus reducing the risk of being trapped in local optima (line 1). Once population initialization is completed (line 9), a normal PSO is conducted to produce candidate solutions (lines 11-16). Please be noted that, although we are solving expensive MaOOPs, only a single ordinal surrogate $h_o$ is used in the reproduction process (line 14). This is a great advantage of LORA-MaOO since existing regression-based SAEAs involve all $M$ surrogates in the reproduction process. Hence, LORA-MaOO is more efficient than these regression-based SAEAs.

### C.3 ANGLE-BASED DIVERSITY SELECTION

Alg. 4 gives the pseudocode of selecting the second optimal solution $x_2^*$ from $P$ via our angle-based diversity criterion, it is the implementation of line 11 in Alg. 1 of the main file. This angle-based diversity selection does not require extra parameters for generating guidance vectors, it selects the candidate solution that is mostly deviate from solutions in $S_{RP}$. Note that all angular surrogates are only used to evaluate one population $P$ during the whole reproduction and environmental selection procedures. Therefore, although LORA-MaOO fits $M$ surrogates in total (one ordinal surrogate and $M$-1 angular surrogates), its runtime cost is less than other SAEAs which fit $M$ surrogates from Cartesian coordinates.

## D DETAILS OF PERFORMANCE INDICATORS USED IN OUR EXPERIMENTS

In our experiments, we use IGD Bosman & Thierens (2003), IGD+ Ishibuchi et al. (2015), and HV Zitzler & Thiele (1998) to measure the performance of many objective optimization. Both IGD and IGD+ require a subset of Pareto front as reference points. In our experiments, the number of IGD/IGD+ reference points is set to 5000 for 3-, 4-, and 6-objective optimization problems, as widely used in the literature Yu et al. (2019). Considering the large objective space, we set the number of IGD/IGD+ reference points to 10000 for 8- and 10-objective optimization problems to

---

**Algorithm 4** Angle-Based Diversity Selection in LORA-MaOO

---

**Input:**
    $S_{RP}$: Reference points used in the ordinal regression;
    $P$: Population of candidate solutions;
    $h_{a1}, \ldots, h_{a(M-1)}$: $M$-1 angular surrogates;

**Procedure:**
  1: $h_{(ai)}(P) \leftarrow$ Evaluate $P$ on angular surrogates $h_{ai}$, i = 1, ..., $M - 1$.
  2: **for** $j = 2$ to $|P|$ **do**
  3:    $\boldsymbol{x}_j \leftarrow$ The $j^{th}$ solution in $P$. /* Assume the first solution in $P$ is selected as $\boldsymbol{x}_1^*$ already. */
  4:    $d_{ang} \leftarrow$ Calculate the angles between $\boldsymbol{x}_j$ and all reference points in $S_{RP}$.
  5:    $md_{ang} \leftarrow$ The angle between $\boldsymbol{x}_j$ and its nearest reference point.
  6: **end for**
  7: $\boldsymbol{x}_2^* \leftarrow$ The candidate solution in $P$ with maximal $md_{ang}$.

**Output:** The second candidate solution $\boldsymbol{x}_2^*$.

---

Table 2: The HV reference points for all problems in this work.

| Problem | Reference Points |
|---|---|
| DTLZ | $(1,0, \ldots, 1.0) \in \mathbb{R}^M$ |
| WFG | $(1,0, \ldots, 1.0) \in \mathbb{R}^M$ |
| NASBench201 | $(1.0, 1.0, 1.0, 1.0, 1.0)$ |

achieve a more accurate estimation of optimization performance. The method proposed in Li et al. (2014) is employed to generate well-distributed IGD/IGD+ reference points.

In comparison, the calculation of HV values does not require a subset of Pareto front as reference points. For a set of non-dominated solutions, its HV is the volume in the objective space it dominates from the set to a single reference point. Table 2 lists the reference point used for calculating HV values. All HV values are calculated using the reference point and the normalized solutions. A solution $\boldsymbol{x}$ is normalize by the upper bound and lower bound of Pareto front:

$$\frac{\boldsymbol{x} - lb_{pf}}{ub_{pf} - lb_{pf}}, \tag{15}$$

where $ub_{pf}$, $lb_{pf}$ are the upper bound and lower bound of Pareto front, respectively.

## E    DETAILS OF THE NASBENCH201 PROBLEM

NASbench201 Dong & Yang (2020) are discrete optimization problems that aim to identify the optimal architecture for neural networks. The search space is defined by a cell with 4 nodes inside, forming a directed acyclic graph as illustrated in Fig. 6. The decision variables are 6 edges, each

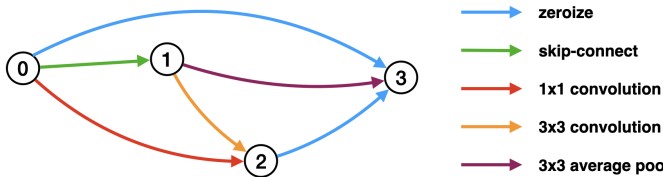

Figure 6: Diagram of a network architecture in NASbench201.

edge is associated with an operation selected from a predefined operation set {zeroize, skip-connect, 1x1 convolution, 3x3 convolution, 3x3 average pool}. Therefore, a network architecture can be encoded into a 6-dimensional decision vector with 5 discrete numbers. In total, there are $5^6$=15,625 different candidates for neural architecture search.

The optimization objectives in NASbench201 varies in different optimization problems. In this paper, our first NASbench201 problem consider 5 objectives, including the accuracy in CI-FAR10

dataset, groundtruth floating point operations (FLOPs), the number of parameters, latency, and energy cost. All these objectives are normalized to [0, 1] in the optimization. The optimization problem can be formulated as

$$F(\boldsymbol{x}) = \{f_{acc}(\boldsymbol{x}), f_{FLOPs}(\boldsymbol{x}), f_{param}(\boldsymbol{x}), f_{latency}(\boldsymbol{x}), f_{energy}(\boldsymbol{x})\}, \tag{16}$$

where decision vector $\boldsymbol{x} \in \{0, 1, 2, 3, 4\}^6$. The second NASbench201 problem consider 3 more objectives, including eyeriss latency, eyeriss energy, and eyeriss arithmetic intensity.

# F   COMPLETE RESULTS OF ABLATION STUDIES

In this section, we report complete results of our ablation studies that are not displayed in the main paper. We conduct four ablation studies to investigate the effect of the following four parameters on the optimization performance of LORA-MaOO.

1. $n_o$: The minimal number of ordinal levels. A parameter in the modeling of our ordinal-regression-based surrogate $h_o$.

2. $\lambda$: The dominance coefficient. A parameter in the modeling of our ordinal-regression-based surrogate $h_o$.

3. $rp_{ratio}$: The ratio threshold of reference points $S_{RP}$. A parameter to determine whether to introduce artificial ordinal relations via clustering.

4. $n_c$: The number of clusters generated from reference points $S_{RP}$ to initialize PSO population. A parameter in the generation of candidate solutions.

Note that setting $\lambda$ to 0 will result in a LORA-MaOO variant without the algorithm component $\lambda$-dominance, so the contribution of this component can be observed and analyzed in the ablation studies on $\lambda$. Similarly, setting $rp_{ratio}$ to 1 or setting $n_c$ to 1 will produce two LORA-MaOO variants without the algorithm components artificial relations or clustering-based initialization. Therefore, we analyze their component contributions in the corresponding ablation studies.

**Setup of Ablation Studies.** Our ablation studies are conducted on 7 DTLZ and 9 WFG benchmark optimization problems. These benchmark problems have different features, such as unimodal, multi-modal, scaled, degenerated, and discontinuous. Therefore, the effect of four parameters can be investigated comprehensively. Considering our paper focuses on many-objective optimization instead of scalable optimization, we are interested in the optimization performance under different numbers of objectives $M$ rather than the performance under different numbers of decision variables $D$. Hence, we set $D = 10$ for all benchmark optimization problems, as suggested in literature Chugh et al. (2016); Pan et al. (2018); Song et al. (2021); Hao et al. (2022). In comparison, we set $M = \{3, 6, 10\}$ to observe the optimization performance with different objectives. Other setups are the same as described in Section 4.1 of the main file.

## F.1   INFLUENCE OF MINIMAL NUMBER OF ORDINAL LEVELS $n_o$.

This subsection investigates the influence of minimal number of ordinal levels $n_o$ on the optimization performance. We set $n_o = \{10, 8, 6, 4, 3\}$ to generate five LORA-MaOO variants. For all variants, in this ablation study, we tentatively set $\lambda = 0.2$, $rp_{ratio} = 2/3$, $n_c = 5$ for a fair comparison. The IGD+ values obtained by five LORA-MaOO variants with different $n_o$ are reported in Table 3.

In the last five rows of Table 3, the summary of statistical test results shows that $n_o = 4$ is the optimal parameter setup for LORA-MaOO, because it is the only variant that is significantly superior to or equivalent to all other variants. In comparison, the LORA-MaOO variant with $n_o = 10, 8, 6, 3$ are significantly inferior to other 4, 1, 1, 2 LORA-MaOO variants, respectively.

## F.2   INFLUENCE OF DOMINANCE COEFFICIENT $\lambda$.

In this subsection, we analyze the influence of $\lambda$-dominance coefficient $\lambda$ on the optimization performance. We set $\lambda = \{0, 0.1, 0.2, 0.3\}$ to generate four LORA-MaOO variants. As determined in the previous ablation study, we set $n_o = 4$ for all variants. The remaining two parameters $rp_{ratio}$

Table 3: Statistical results of the IGD+ value obtained by LORA-MaOO with different $n_o$ on 48 benchmark optimization problems over 15 runs. The last five rows count the total results of Wilcoxon rank sum tests (significance level is 0.05). '+', '≈', and '−' denote the corresponding LORA-MaOO variant is statistically significantly superior to, almost equivalent to, and inferior to the compared variants in Wilcoxon tests, respectively.

| Problems | M | $n_o$=10 | $n_o$=8 | $n_o$=6 | $n_o$=4 | $n_o$=3 |
|---|---|---|---|---|---|---|
| DTLZ1 | 3 | 4.63e+1(1.60e+1) | 4.64e+1(1.23e+1) | 5.61e+1(2.04e+1) | 4.84e+1(1.34e+1) | 4.58e+1(1.85e+1) |
| | 6 | 1.35e+1(7.10e+0) | 1.77e+1(5.08e+0) | 1.87e+1(6.85e+0) | 1.64e+1(3.24e+0) | 1.50e+1(7.84e+0) |
| | 10 | 1.56e-1(3.58e-2) | 1.60e-1(3.60e-2) | 1.63e-1(6.95e-2) | 1.60e-1(2.67e-2) | 1.63e-1(3.51e-2) |
| DTLZ2 | 3 | 4.50e-2(3.90e-3) | 4.54e-2(4.16e-3) | 4.38e-2(2.61e-3) | 4.45e-2(4.72e-3) | 4.39e-2(3.88e-3) |
| | 6 | 2.67e-1(1.47e-2) | 2.73e-1(1.93e-2) | 2.64e-1(1.67e-2) | 2.57e-1(1.91e-2) | 2.51e-1(2.20e-2) |
| | 10 | 3.04e-1(1.55e-2) | 2.97e-1(1.63e-2) | 2.94e-1(1.24e-2) | 3.00e-1(1.31e-2) | 3.11e-1(1.78e-2) |
| DTLZ3 | 3 | 1.50e+2(4.72e+1) | 1.60e+2(4.92e+1) | 1.55e+2(5.03e+1) | 1.48e+2(4.92e+1) | 1.45e+2(4.10e+1) |
| | 6 | 5.43e+1(1.85e+1) | 5.65e+1(1.99e+1) | 6.92e+1(2.39e+1) | 6.68e+1(1.64e+1) | 6.24e+1(2.34e+1) |
| | 10 | 4.51e-1(4.40e-2) | 4.68e-1(6.10e-2) | 4.35e-1(3.71e-2) | 4.72e-1(5.45e-2) | 4.85e-1(7.87e-2) |
| DTLZ4 | 3 | 1.03e-1(1.28e-1) | 8.77e-2(1.30e-1) | 9.16e-2(1.25e-1) | 1.05e-1(1.27e-1) | 1.15e-1(1.33e-1) |
| | 6 | 1.74e-1(3.63e-2) | 1.60e-1(3.35e-2) | 1.84e-1(3.79e-2) | 1.75e-1(3.57e-2) | 1.68e-1(2.11e-2) |
| | 10 | 2.29e-1(1.05e-2) | 2.29e-1(9.43e-3) | 2.36e-1(1.27e-2) | 2.38e-1(1.35e-2) | 2.42e-1(1.71e-2) |
| DTLZ5 | 3 | 8.65e-3(1.39e-3) | 8.76e-3(1.53e-3) | 9.03e-3(1.67e-3) | 9.26e-3(1.22e-3) | 9.26e-3(2.23e-3) |
| | 6 | 3.43e-2(7.07e-3) | 3.28e-2(7.74e-3) | 3.24e-2(7.73e-3) | 3.25e-2(8.25e-3) | 3.33e-2(9.38e-3) |
| | 10 | 4.06e-3(6.52e-4) | 3.99e-3(4.47e-4) | 3.94e-3(4.04e-4) | 3.97e-3(9.34e-4) | 4.02e-3(1.10e-3) |
| DTLZ6 | 3 | 5.09e-2(5.72e-2) | 1.05e-1(2.57e-1) | 2.45e-2(8.80e-3) | 4.67e-2(4.92e-2) | 3.12e-2(1.58e-2) |
| | 6 | 9.45e-1(1.13e+0) | 5.16e-1(6.72e-1) | 5.42e-1(8.28e-1) | 7.52e-1(9.50e-1) | 1.34e+0(1.04e+0) |
| | 10 | 4.48e-2(3.90e-2) | 2.50e-2(7.37e-3) | 5.14e-2(4.26e-2) | 4.18e-2(4.66e-2) | 4.72e-2(4.57e-2) |
| DTLZ7 | 3 | 1.19e-1(1.00e-1) | 9.47e-2(1.15e-1) | 1.16e-1(7.80e-2) | 1.61e-1(2.77e-1) | 1.46e-1(1.27e-1) |
| | 6 | 1.90e+0(9.89e-1) | 1.72e+0(6.52e-1) | 1.77e+0(7.63e-1) | 1.25e+0(4.72e-1) | 1.54e+0(8.80e-1) |
| | 10 | 1.19e+0(9.00e-2) | 1.18e+0(9.13e-2) | 1.17e+0(8.41e-2) | 1.17e+0(8.97e-2) | 1.22e+0(1.13e-1) |
| WFG1 | 3 | 1.65e+0(5.78e-2) | 1.65e+0(6.93e-2) | 1.64e+0(3.86e-2) | 1.67e+0(4.67e-2) | 1.65e+0(5.96e-2) |
| | 6 | 2.24e+0(5.47e-2) | 2.20e+0(6.93e-2) | 2.23e+0(4.37e-2) | 2.22e+0(6.80e-2) | 2.21e+0(5.52e-2) |
| | 10 | 2.62e+0(8.72e-2) | 2.58e+0(7.39e-2) | 2.59e+0(7.81e-2) | 2.62e+0(8.93e-2) | 2.58e+0(1.16e-1) |
| WFG2 | 3 | 2.39e-1(3.16e-2) | 2.49e-1(4.94e-2) | 2.68e-1(4.81e-2) | 2.52e-1(4.94e-2) | 2.66e-1(4.58e-2) |
| | 6 | 5.91e-1(1.79e-1) | 5.85e-1(9.10e-2) | 5.61e-1(1.29e-1) | 5.43e-1(1.51e-1) | 5.67e-1(1.07e-1) |
| | 10 | 1.50e+0(3.53e-1) | 1.41e+0(2.62e-1) | 1.42e+0(3.21e-1) | 1.47e+0(4.49e-1) | 1.39e+0(2.82e-1) |
| WFG3 | 3 | 2.42e-1(4.10e-2) | 2.66e-1(3.75e-2) | 2.57e-1(3.28e-2) | 2.41e-1(3.21e-2) | 2.56e-1(5.04e-2) |
| | 6 | 6.19e-1(8.08e-2) | 6.28e-1(6.58e-2) | 6.15e-1(9.32e-2) | 5.92e-1(7.43e-2) | 6.19e-1(1.22e-1) |
| | 10 | 6.24e-1(9.78e-2) | 6.07e-1(8.67e-2) | 6.18e-1(8.74e-2) | 6.60e-1(8.00e-2) | 6.61e-1(8.80e-2) |
| WFG4 | 3 | 2.62e-1(5.18e-2) | 2.52e-1(1.99e-2) | 2.51e-1(1.27e-2) | 2.48e-1(1.04e-2) | 2.38e-1(8.69e-3) |
| | 6 | 1.41e+0(2.17e-1) | 1.34e+0(1.96e-1) | 1.27e+0(2.31e-1) | 1.30e+0(2.41e-1) | 1.58e+0(4.08e-1) |
| | 10 | 4.12e+0(5.64e-1) | 3.63e+0(6.43e-1) | 3.55e+0(5.77e-1) | 3.99e+0(7.21e-1) | 4.08e+0(7.57e-1) |
| WFG5 | 3 | 2.93e-1(4.46e-2) | 2.89e-1(5.58e-2) | 3.01e-1(9.11e-2) | 3.10e-1(5.46e-2) | 3.19e-1(9.97e-2) |
| | 6 | 1.69e+0(8.33e-2) | 1.72e+0(8.16e-2) | 1.66e+0(9.57e-2) | 1.69e+0(1.53e-1) | 1.83e+0(1.34e-1) |
| | 10 | 4.76e+0(2.87e-1) | 4.57e+0(3.19e-1) | 4.10e+0(3.07e-1) | 3.71e+0(3.87e-1) | 3.71e+0(4.39e-1) |
| WFG6 | 3 | 4.66e-1(4.13e-2) | 4.91e-1(4.44e-2) | 4.51e-1(4.36e-2) | 4.76e-1(6.61e-2) | 4.58e-1(8.29e-2) |
| | 6 | 1.70e+0(1.48e-1) | 1.65e+0(9.89e-2) | 1.61e+0(1.10e-1) | 1.67e+0(1.35e-1) | 1.81e+0(2.71e-1) |
| | 10 | 3.88e+0(6.68e-1) | 3.60e+0(3.51e-1) | 3.64e+0(2.96e-1) | 3.45e+0(4.44e-1) | 3.72e+0(5.21e-1) |
| WFG7 | 3 | 3.12e-1(2.16e-2) | 3.02e-1(2.17e-2) | 3.00e-1(2.68e-2) | 3.02e-1(2.75e-2) | 2.99e-1(2.96e-2) |
| | 6 | 1.78e+0(1.05e-1) | 1.69e+0(1.27e-1) | 1.73e+0(1.38e-1) | 1.67e+0(1.85e-1) | 1.74e+0(2.32e-1) |
| | 10 | 5.15e+0(3.94e-1) | 5.11e+0(2.97e-1) | 4.89e+0(2.62e-1) | 4.97e+0(3.07e-1) | 4.94e+0(4.00e-1) |
| WFG8 | 3 | 5.84e-1(5.34e-2) | 6.09e-1(5.54e-2) | 6.07e-1(4.89e-2) | 5.68e-1(4.78e-2) | 5.70e-1(4.15e-2) |
| | 6 | 2.19e+0(1.08e-1) | 2.11e+0(9.97e-2) | 2.15e+0(1.22e-1) | 2.25e+0(1.12e-1) | 2.37e+0(1.76e-1) |
| | 10 | 5.22e+0(4.43e-1) | 5.31e+0(3.08e-1) | 4.99e+0(3.75e-1) | 5.16e+0(5.37e-1) | 5.37e+0(4.82e-1) |
| WFG9 | 3 | 3.79e-1(7.28e-2) | 3.85e-1(1.20e-1) | 3.73e-1(8.90e-2) | 4.12e-1(1.17e-1) | 4.17e-1(1.11e-1) |
| | 6 | 1.87e+0(1.95e-1) | 1.73e+0(2.02e-1) | 1.78e+0(2.45e-1) | 1.77e+0(2.57e-1) | 1.76e+0(1.35e-1) |
| | 10 | 5.03e+0(2.28e-1) | 4.63e+0(4.11e-1) | 4.44e+0(4.68e-1) | 3.96e+0(3.83e-1) | 3.73e+0(2.50e-1) |
| +/ ≈ /− | $n_o$=10 | -/-/- | 1/41/6 | 2/40/6 | 0/44/4 | 3/41/4 |
| +/ ≈ /− | $n_o$=8 | 6/41/1 | -/-/- | 2/43/3 | 3/42/3 | 4/40/4 |
| +/ ≈ /− | $n_o$=6 | 6/40/2 | 3/43/2 | -/-/- | 3/41/4 | 7/38/3 |
| +/ ≈ /− | $n_o$=4 | 4/44/0 | 3/42/3 | 4/41/3 | -/-/- | 2/45/1 |
| +/ ≈ /− | $n_o$=3 | 4/41/3 | 4/40/4 | 3/38/7 | 1/45/2 | -/-/- |

and $n_c$ are set to 2/3 and 5, respectively. The IGD+ values obtained by four LORA-MaOO variants with different $\lambda$ are reported in Table 4.

The last four rows of Table 4 shows that $\lambda = 0.2$ is the optimal parameter setup for LORA-MaOO. The variant of $\lambda = 0.2$ is significantly superior to both the variants of $\lambda = 0$ and $\lambda = 0.1$, and it is equivalent to the variant of $\lambda = 0.3$. We note that the variant of $\lambda = 0.3$ is also significantly superior to both the variants of $\lambda = 0$ and $\lambda = 0.1$. However, this variant wins/ties/losses 30/105/9 statistical tests in total, while the variant of $\lambda = 0.2$ wins/ties/losses 32/109/3 statistical tests in total. Therefore, setting $\lambda = 0.2$ is preferable to setting $\lambda = 0.3$.

Table 4: Statistical results of the IGD+ value obtained by LORA-MaOO with different $\lambda$ on 48 benchmark optimization problems over 15 runs. The last four rows count the total results of Wilcoxon rank sum tests (significance level is 0.05). '+', '≈', and '−' denote the corresponding LORA-MaOO variant is statistically significantly superior to, almost equivalent to, and inferior to the compared variants in Wilcoxon tests, respectively.

| Problems | M | $\lambda = 0$ | $\lambda = 0.1$ | $\lambda = 0.2$ | $\lambda = 0.3$ |
|---|---|---|---|---|---|
| DTLZ1 | 3 | 7.51e+1(1.74e+1) | 6.88e+1(1.28e+1) | 4.84e+1(1.34e+1) | 4.96e+1(1.56e+1) |
|  | 6 | 2.74e+1(5.30e+0) | 1.73e+1(3.80e+0) | 1.64e+1(3.24e+0) | 1.41e+1(7.02e+0) |
|  | 10 | 1.62e-1(5.15e-2) | 1.43e-1(2.33e-2) | 1.60e-1(2.67e-2) | 1.53e-1(2.28e-2) |
| DTLZ2 | 3 | 4.95e-2(3.32e-3) | 4.89e-2(5.80e-3) | 4.45e-2(4.72e-3) | 4.81e-2(4.10e-3) |
|  | 6 | 2.51e-1(2.91e-2) | 2.56e-1(2.48e-2) | 2.57e-1(1.91e-2) | 2.67e-1(1.34e-2) |
|  | 10 | 2.97e-1(1.72e-2) | 2.94e-1(1.54e-2) | 3.00e-1(1.31e-2) | 2.92e-1(1.35e-2) |
| DTLZ3 | 3 | 1.91e+2(6.02e+1) | 1.80e+2(2.31e+1) | 1.48e+2(4.92e+1) | 1.57e+2(4.54e+1) |
|  | 6 | 9.01e+1(3.13e+1) | 8.06e+1(2.18e+1) | 6.68e+1(1.64e+1) | 6.05e+1(2.03e+1) |
|  | 10 | 5.74e-1(2.57e-1) | 4.60e-1(5.69e-2) | 4.72e-1(5.45e-2) | 4.48e-1(4.14e-2) |
| DTLZ4 | 3 | 9.37e-2(1.30e-1) | 1.16e-1(1.35e-1) | 1.05e-1(1.27e-1) | 1.02e-1(1.28e-1) |
|  | 6 | 1.72e-1(2.91e-2) | 1.63e-1(3.51e-2) | 1.75e-1(3.57e-2) | 1.61e-1(1.96e-2) |
|  | 10 | 2.36e-1(1.29e-2) | 2.37e-1(1.77e-2) | 2.38e-1(1.35e-2) | 2.28e-1(1.05e-2) |
| DTLZ5 | 3 | 1.40e-2(2.50e-3) | 1.13e-2(3.34e-3) | 9.26e-3(1.22e-3) | 7.96e-3(1.58e-3) |
|  | 6 | 5.00e-2(9.20e-3) | 4.52e-2(1.60e-2) | 3.25e-2(8.25e-3) | 3.48e-2(5.12e-3) |
|  | 10 | 5.16e-3(9.20e-4) | 4.44e-3(1.43e-3) | 3.97e-3(9.34e-4) | 4.10e-3(3.97e-4) |
| DTLZ6 | 3 | 1.54e-1(1.65e-1) | 4.14e-2(1.61e-2) | 4.67e-2(4.92e-2) | 4.13e-2(2.30e-2) |
|  | 6 | 1.72e+0(7.66e-1) | 1.52e+0(1.08e+0) | 7.52e-1(9.50e-1) | 2.45e-1(4.79e-1) |
|  | 10 | 9.60e-2(7.76e-2) | 6.08e-2(5.26e-2) | 4.18e-2(4.66e-2) | 2.99e-2(9.13e-3) |
| DTLZ7 | 3 | 6.57e-2(1.85e-2) | 1.25e-1(1.06e-1) | 1.61e-1(2.77e-1) | 1.05e-1(1.80e-1) |
|  | 6 | 2.74e+0(1.22e+0) | 1.53e+0(8.21e-1) | 1.25e+0(4.72e-1) | 1.66e+0(1.06e+0) |
|  | 10 | 1.19e+0(9.70e-2) | 1.18e+0(8.58e-2) | 1.17e+0(8.97e-2) | 1.27e+0(1.61e-1) |
| WFG1 | 3 | 1.74e+0(4.92e-2) | 1.67e+0(4.67e-2) | 1.67e+0(4.67e-2) | 1.64e+0(3.52e-2) |
|  | 6 | 2.30e+0(3.54e-2) | 2.22e+0(8.09e-2) | 2.22e+0(6.80e-2) | 2.23e+0(7.54e-2) |
|  | 10 | 2.71e+0(6.98e-2) | 2.63e+0(7.80e-2) | 2.62e+0(8.93e-2) | 2.63e+0(7.71e-2) |
| WFG2 | 3 | 2.94e-1(5.47e-2) | 2.69e-1(5.46e-2) | 2.52e-1(4.94e-2) | 2.55e-1(3.46e-2) |
|  | 6 | 6.84e-1(1.47e-1) | 5.38e-1(1.05e-1) | 5.43e-1(1.51e-1) | 6.65e-1(2.55e-1) |
|  | 10 | 1.67e+0(5.02e-1) | 1.27e+0(2.80e-1) | 1.47e+0(4.49e-1) | 1.37e+0(3.46e-1) |
| WFG3 | 3 | 4.08e-1(4.84e-2) | 3.25e-1(3.53e-2) | 2.41e-1(3.21e-2) | 2.70e-1(5.19e-2) |
|  | 6 | 8.23e-1(6.96e-2) | 7.51e-1(9.15e-2) | 5.92e-1(7.43e-2) | 4.94e-1(6.55e-2) |
|  | 10 | 7.58e-1(7.71e-2) | 7.71e-1(1.08e-1) | 6.60e-1(8.00e-2) | 6.35e-1(1.04e-1) |
| WFG4 | 3 | 2.55e-1(1.63e-2) | 2.56e-1(1.48e-2) | 2.48e-1(1.04e-2) | 2.57e-1(1.44e-2) |
|  | 6 | 1.28e+0(2.24e-1) | 1.31e+0(2.39e-1) | 1.30e+0(2.41e-1) | 1.37e+0(2.50e-1) |
|  | 10 | 3.85e+0(5.45e-1) | 3.84e+0(5.48e-1) | 3.99e+0(7.21e-1) | 3.79e+0(4.91e-1) |
| WFG5 | 3 | 3.84e-1(1.18e-1) | 2.89e-1(6.47e-2) | 3.10e-1(5.46e-2) | 3.11e-1(6.94e-2) |
|  | 6 | 1.77e+0(1.36e-1) | 1.72e+0(1.43e-1) | 1.69e+0(1.53e-1) | 1.72e+0(1.20e-1) |
|  | 10 | 3.70e+0(4.80e-1) | 3.58e+0(2.79e-1) | 3.71e+0(3.87e-1) | 4.38e+0(2.67e-1) |
| WFG6 | 3 | 4.78e-1(7.23e-2) | 4.63e-1(5.50e-2) | 4.76e-1(6.61e-2) | 4.74e-1(4.87e-2) |
|  | 6 | 1.62e+0(1.67e-1) | 1.59e+0(1.21e-1) | 1.67e+0(1.35e-1) | 1.60e+0(1.52e-1) |
|  | 10 | 3.48e+0(2.80e-1) | 3.43e+0(3.18e-1) | 3.45e+0(4.44e-1) | 3.70e+0(3.85e-1) |
| WFG7 | 3 | 3.16e-1(2.20e-2) | 3.13e-1(3.79e-2) | 3.02e-1(2.75e-2) | 3.17e-1(4.42e-2) |
|  | 6 | 1.62e+0(1.57e-1) | 1.68e+0(1.80e-1) | 1.67e+0(1.85e-1) | 1.69e+0(1.88e-1) |
|  | 10 | 4.88e+0(4.14e-1) | 4.99e+0(3.94e-1) | 4.97e+0(3.07e-1) | 4.98e+0(2.87e-1) |
| WFG8 | 3 | 5.96e-1(4.58e-2) | 6.09e-1(3.63e-2) | 5.68e-1(4.78e-2) | 5.96e-1(3.58e-2) |
|  | 6 | 2.21e+0(1.49e-1) | 2.20e+0(1.18e-1) | 2.25e+0(1.12e-1) | 2.20e+0(7.76e-2) |
|  | 10 | 5.07e+0(4.48e-1) | 4.96e+0(4.84e-1) | 5.16e+0(5.37e-1) | 5.09e+0(3.92e-1) |
| WFG9 | 3 | 3.72e-1(3.91e-2) | 3.82e-1(9.02e-2) | 4.12e-1(1.17e-1) | 3.80e-1(1.00e-1) |
|  | 6 | 1.76e+0(2.07e-1) | 1.67e+0(1.86e-1) | 1.77e+0(2.57e-1) | 1.81e+0(1.69e-1) |
|  | 10 | 3.87e+0(3.66e-1) | 4.13e+0(3.55e-1) | 3.96e+0(3.83e-1) | 4.76e+0(2.31e-1) |
| $+/\approx/-$ | $\lambda=0$ | -/-/- | 0/35/13 | 0/29/19 | 3/27/18 |
| $+/\approx/-$ | $\lambda=0.1$ | 13/35/0 | -/-/- | 0/38/10 | 3/36/9 |
| $+/\approx/-$ | $\lambda=0.2$ | 19/29/0 | 10/38/0 | -/-/- | 3/42/3 |
| $+/\approx/-$ | $\lambda=0.3$ | 18/27/3 | 9/36/3 | 3/42/3 | -/-/- |

Note that all other LORA-MaOO variants outperform the variant of $\lambda = 0$, this implies that excluding some samples from the set of non-dominated solutions is beneficial to the performance of ordinal regression. The effectiveness of using our $\lambda$-dominance approach in LORA-MaOO is demonstrated.

### F.3 INFLUENCE OF RATIO THRESHOLD $rp_{ratio}$.

In this subsection, we investigate the influence of ratio threshold $rp_{ratio}$ on the optimization performance. $rp_{ratio}$ is the threshold to determine when to add artificial ordinal relations for the training of ordinal surrogate $h_o$. We set $rp_{ratio} = \{1, 2/3, 1/2, 1/3\}$ to generate four LORA-MaOO variants. For all variants, we set $n_o$, $\lambda$ to 4, 0.2, respectively, which are consistent with our conclusions in

Table 5: Statistical results of the IGD+ value obtained by LORA-MaOO with different $rp_{ratio}$ on 48 benchmark optimization problems over 15 runs. The last four rows count the total results of Wilcoxon rank sum tests (significance level is 0.05). '+', '≈', and '−' denote the corresponding LORA-MaOO variant is statistically significantly superior to, almost equivalent to, and inferior to the compared variants in Wilcoxon tests, respectively.

| Problems | M | $rp_{ratio}$=1 | $rp_{ratio}$=2/3 | $rp_{ratio}$=1/2 | $rp_{ratio}$=1/3 |
|---|---|---|---|---|---|
| DTLZ1 | 3 | 4.84e+1(1.34e+1) | 4.84e+1(1.34e+1) | 4.75e+1(1.54e+1) | 4.75e+1(1.54e+1) |
| | 6 | 1.83e+1(1.06e+1) | 1.64e+1(3.24e+0) | 1.35e+1(6.23e+0) | 1.35e+1(6.23e+0) |
| | 10 | 1.63e-1(2.74e-2) | 1.60e-1(2.67e-2) | 1.58e-1(2.81e-2) | 1.58e-1(2.81e-2) |
| DTLZ2 | 3 | 4.45e-2(4.72e-3) | 4.45e-2(4.72e-3) | 4.37e-2(3.41e-3) | 3.60e-2(3.69e-3) |
| | 6 | 2.57e-1(1.93e-2) | 2.57e-1(1.91e-2) | 1.80e-1(1.17e-2) | 1.80e-1(7.34e-3) |
| | 10 | 3.74e-1(8.09e-3) | 3.00e-1(1.31e-2) | 2.87e-1(1.71e-2) | 2.87e-1(1.71e-2) |
| DTLZ3 | 3 | 1.48e+2(4.92e+1) | 1.48e+2(4.92e+1) | 1.54e+2(4.89e+1) | 1.54e+2(4.89e+1) |
| | 6 | 6.52e+1(2.87e+1) | 6.68e+1(1.64e+1) | 6.01e+1(2.61e+1) | 6.01e+1(2.61e+1) |
| | 10 | 4.23e-1(5.63e-2) | 4.72e-1(5.45e-2) | 4.84e-1(5.71e-2) | 4.84e-1(5.71e-2) |
| DTLZ4 | 3 | 1.05e-1(1.27e-1) | 1.05e-1(1.27e-1) | 1.06e-1(1.32e-1) | 1.06e-1(1.32e-1) |
| | 6 | 1.70e-1(3.56e-2) | 1.75e-1(3.57e-2) | 1.79e-1(4.06e-2) | 1.79e-1(4.06e-2) |
| | 10 | 2.33e-1(1.26e-2) | 2.38e-1(1.35e-2) | 2.38e-1(1.56e-2) | 2.49e-1(1.46e-2) |
| DTLZ5 | 3 | 9.26e-3(1.22e-3) | 9.26e-3(1.22e-3) | 8.98e-3(1.67e-3) | 8.71e-3(1.89e-3) |
| | 6 | 3.40e-2(9.35e-3) | 3.25e-2(8.25e-3) | 3.31e-2(7.84e-3) | 2.81e-2(1.15e-2) |
| | 10 | 3.83e-3(6.08e-4) | 3.97e-3(9.34e-4) | 4.85e-3(1.78e-3) | 4.92e-3(1.54e-3) |
| DTLZ6 | 3 | 4.67e-2(4.92e-2) | 4.67e-2(4.92e-2) | 6.38e-2(7.62e-2) | 2.56e-2(6.58e-3) |
| | 6 | 4.70e-1(7.64e-1) | 7.52e-1(9.50e-1) | 7.28e-1(1.00e+0) | 1.25e+0(1.13e+0) |
| | 10 | 3.38e-2(1.18e-2) | 4.18e-2(4.66e-2) | 3.92e-2(3.62e-2) | 3.27e-2(2.08e-2) |
| DTLZ7 | 3 | 1.61e-1(2.77e-1) | 1.61e-1(2.77e-1) | 1.36e-1(1.32e-1) | 7.58e-2(2.50e-2) |
| | 6 | 1.41e+0(9.24e-1) | 1.25e+0(4.72e-1) | 1.21e+0(7.32e-1) | 1.28e+0(6.69e-1) |
| | 10 | 1.17e+0(8.28e-2) | 1.17e+0(8.97e-2) | 1.23e+0(1.33e-1) | 1.23e+0(1.33e-1) |
| WFG1 | 3 | 1.67e+0(4.67e-2) | 1.67e+0(4.67e-2) | 1.67e+0(4.86e-2) | 1.67e+0(4.86e-2) |
| | 6 | 2.20e+0(6.03e-2) | 2.22e+0(6.80e-2) | 2.21e+0(5.69e-2) | 2.21e+0(5.69e-2) |
| | 10 | 2.61e+0(1.15e-1) | 2.62e+0(8.93e-2) | 2.55e+0(1.15e-1) | 2.55e+0(1.15e-1) |
| WFG2 | 3 | 2.52e-1(4.94e-2) | 2.52e-1(4.94e-2) | 2.48e-1(5.57e-2) | 2.48e-1(5.57e-2) |
| | 6 | 5.73e-1(1.75e-1) | 5.43e-1(1.51e-1) | 5.35e-1(9.94e-2) | 5.35e-1(9.94e-2) |
| | 10 | 1.37e+0(3.08e-1) | 1.47e+0(4.49e-1) | 1.36e+0(3.13e-1) | 1.25e+0(3.81e-1) |
| WFG3 | 3 | 2.41e-1(3.21e-2) | 2.41e-1(3.21e-2) | 2.51e-1(3.82e-2) | 2.51e-1(3.26e-2) |
| | 6 | 5.82e-1(4.97e-2) | 5.92e-1(7.43e-2) | 5.83e-1(8.20e-2) | 6.05e-1(9.65e-2) |
| | 10 | 6.09e-1(4.65e-2) | 6.60e-1(8.00e-2) | 6.93e-1(1.22e-1) | 6.63e-1(1.05e-1) |
| WFG4 | 3 | 2.48e-1(1.04e-2) | 2.48e-1(1.04e-2) | 2.49e-1(2.61e-2) | 2.96e-1(9.20e-2) |
| | 6 | 2.06e+0(4.21e-1) | 1.30e+0(2.41e-1) | 1.35e+0(3.15e-1) | 1.35e+0(3.15e-1) |
| | 10 | 5.51e+0(6.14e-1) | 3.99e+0(7.21e-1) | 3.86e+0(6.03e-1) | 3.86e+0(6.03e-1) |
| WFG5 | 3 | 3.10e-1(5.46e-2) | 3.10e-1(5.46e-2) | 3.06e-1(1.05e-1) | 4.28e-1(1.46e-1) |
| | 6 | 1.93e+0(1.20e-1) | 1.69e+0(1.53e-1) | 1.72e+0(1.26e-1) | 1.72e+0(1.26e-1) |
| | 10 | 5.50e+0(3.80e-1) | 3.71e+0(3.87e-1) | 3.63e+0(4.80e-1) | 3.63e+0(4.80e-1) |
| WFG6 | 3 | 4.76e-1(6.61e-2) | 4.76e-1(6.61e-2) | 4.87e-1(1.00e-1) | 6.26e-1(1.19e-1) |
| | 6 | 2.21e+0(2.26e-1) | 1.67e+0(1.35e-1) | 1.62e+0(1.85e-1) | 1.62e+0(1.85e-1) |
| | 10 | 5.43e+0(4.78e-1) | 3.45e+0(4.44e-1) | 3.19e+0(2.14e-1) | 3.19e+0(2.14e-1) |
| WFG7 | 3 | 3.02e-1(2.75e-2) | 3.02e-1(2.75e-2) | 2.95e-1(2.76e-2) | 2.98e-1(3.12e-2) |
| | 6 | 2.10e+0(2.12e-1) | 1.67e+0(1.85e-1) | 1.58e+0(1.47e-1) | 1.58e+0(1.47e-1) |
| | 10 | 5.85e+0(5.16e-1) | 4.97e+0(3.07e-1) | 4.76e+0(4.89e-1) | 4.76e+0(4.89e-1) |
| WFG8 | 3 | 5.68e-1(4.78e-2) | 5.68e-1(4.78e-2) | 5.71e-1(4.02e-2) | 5.83e-1(4.65e-2) |
| | 6 | 2.61e+0(2.09e-1) | 2.25e+0(1.12e-1) | 2.21e+0(1.21e-1) | 2.21e+0(1.21e-1) |
| | 10 | 6.41e+0(4.20e-1) | 5.16e+0(5.37e-1) | 5.06e+0(5.80e-1) | 5.06e+0(5.80e-1) |
| WFG9 | 3 | 4.12e-1(1.17e-1) | 4.12e-1(1.17e-1) | 3.81e-1(1.02e-1) | 3.66e-1(8.95e-2) |
| | 6 | 1.86e+0(2.00e-1) | 1.77e+0(2.57e-1) | 1.48e+0(2.27e-1) | 1.45e+0(1.77e-1) |
| | 10 | 5.57e+0(2.73e-1) | 3.96e+0(3.83e-1) | 4.02e+0(4.62e-1) | 4.02e+0(4.62e-1) |
| +/ ≈ /− | $rp_{ratio}$=1 | -/-/- | 2/34/12 | 2/32/14 | 5/28/15 |
| +/ ≈ /− | $rp_{ratio}$=2/3 | 12/34/2 | -/-/- | 0/46/2 | 3/42/3 |
| +/ ≈ /− | $rp_{ratio}$=1/2 | 14/32/2 | 2/46/0 | -/-/- | 2/45/1 |
| +/ ≈ /− | $rp_{ratio}$=1/3 | 15/28/5 | 3/42/3 | 1/45/2 | -/-/- |

previous ablation studies. Parameter $n_c$ is tentatively set to 5. The IGD+ values obtained by four LORA-MaOO variants with different $rp_{ratio}$ are reported in Table 5. It should be noted that, when the number of objectives $M = 3$, the results of $rp_{ratio} = 1$ are the same as the results of $rp_{ratio} = 2/3$, because the ratio of reference points in archive $S_A$ is always lower than 2/3. Consequently, when $M = 3$, setting ratio threshold $rp_{ratio}$ to either 1 or 2/3 makes no difference to the optimization process of LORA-MaOO. Similarly, the results of $rp_{ratio} = 1/3$ on some problems are the same as the results obtained by setting $rp_{ratio}$ to 1/2, because on these problems, the ratio of reference points in $S_A$ is always higher than 1/2.

As shown in Table 5, the variant of $rp_{ratio}$ = 1/2 outperforms other variants and achieves the optimal behavior. Therefore, we set $rp_{ratio}$ = 1/2 for LORA-MaOO. In comparison, the variants of $rp_{ratio}$ = 2/3 and $rp_{ratio}$ = 1/3 have competitive performance, both of them are inferior to the variant of $rp_{ratio}$ = 1/2 but significantly superior to the variant of $rp_{ratio}$ = 1.

Setting $rp_{ratio}$ = 1 indicates this LORA-MaOO variant will never introduce artificial ordinal relations for the learning of the ordinal surrogate. The ordinal surrogate in this variant is trained completely on the basis of dominance ordinal relations. When the number of objectives $M$ is large, a majority of evaluated solutions in archive $S_A$ are non-dominated, leading to a large ratio of reference points $S_{RP}$ in $S_A$. As a result, there would be a significant imbalance between the number of evaluated solutions in each ordinal level, which causes a poor performance on ordinal surrogate and LORA-MaOO. In particular, on most 10-objective WFG problems, the variant of $rp_{ratio}$ = 1 performs worse than all other variants. This observation shows the detrimental effect of imbalance solutions in ordinal levels on the optimization performance, which also demonstrates the effectiveness of using artificial ordinal relations in LORA-MaOO to address many-objective optimization problems.

### F.4 INFLUENCE OF CLUSTERING NUMBER FOR REPRODUCTION $n_c$.

This subsection analyzes the influence of clustering number $n_c$ on the optimization performance. $n_c$ is used in the reproduction process to initialize the PSO population. We set $n_c = \{1, 3, 5, 7, 10\}$ to generate five LORA-MaOO variants. According to the conclusions of previous ablation studies, in this ablation study, we set $n_o = 4$, $\lambda = 0.2$, $rp_{ratio}$ = 1/2 for all variants. The IGD+ values obtained by five LORA-MaOO variants with different $n_c$ are reported in Table 6.

It can be observed that both the variants of $n_c$ = 5 and $n_c$ = 7 outperform three other variants and are inferior to one variant, showing the optimal performance over other variants in this ablation study. In comparison, the variants of $n_c$ = 3 and $n_c$ = 10 are significantly superior to two variants but are also significantly inferior to two other variants. The variant of $n_c$ = 1 reaches the worst optimization results as it is significantly inferior to all other variants. In addition, considering that the variant of $n_c$ = 7 wins/ties/losses 2/45/1 statistical tests when compared with the variant of $n_c$ = 5, we set $n_c$ = 7 for LORA-MaOO.

The result of this ablation study demonstrates the influence of population initialization on the optimization results. By clustering the evaluated solutions into several clusters and sampling the same amount of initial solutions from each cluster, the solutions in the initial population are distributed in a more diverse way than the solutions sampled from the set of reference points $S_{RP}$ directly. Consequently, all variants of $n_c > 1$ have achieved better optimization results than the variant of $n_c$ = 1.

## G SOLUTION DISTRIBUTION

The solution distribution we obtained on some 3-objective DTLZ problems are plotted.

## H COMPLETE RESULTS OF BENCHMARK OPTIMIZATION

In Section 4.3 of the main file, we display the optimization results of comparison algorithms on DTLZ problems in terms of IGD values. In this section, we provide detailed IGD results on WFG problems and more results on IGD+ and HV values. In addition, the optimization results on DTLZ problems with different scales, such as $D$ = 5 and 20, are reported.

### H.1 IGD RESULTS ON WFG OPTIMIZATION PROBLEMS

Table 7 shows the optimization results on WFG problems in terms of IGD values. The last row summarizes the results of statistical tests, which has reported at the end of Table 1 in the main file. It can be seen that LORA-MaOO outperforms all comparison algorithms, followed by KTA2 and KRVEA. This is consistent with the results we observed from Table 1. The results on six 3- and 10-objective WFG problems are plotted in Fig. 11.

Table 6: Statistical results of the IGD+ value obtained by LORA-MaOO with different $n_c$ on 48 benchmark optimization problems over 15 runs. The last five rows count the total results of Wilcoxon rank sum tests (significance level is 0.05). '+', '≈', and '−' denote the corresponding LORA-MaOO variant is statistically significantly superior to, almost equivalent to, and inferior to the compared variants in Wilcoxon tests, respectively.

| Problems | M | $n_c$=1 | $n_c$=3 | $n_c$=5 | $n_c$=7 | $n_c$=10 |
|---|---|---|---|---|---|---|
| DTLZ1 | 3 | 6.45e+1(1.31e+1) | 5.77e+1(2.13e+1) | 4.75e+1(1.54e+1) | 4.02e+1(1.46e+1) | 3.91e+1(1.53e+1) |
|  | 6 | 2.22e+1(5.99e+0) | 1.67e+1(4.35e+0) | 1.35e+1(6.23e+0) | 1.55e+1(5.29e+0) | 1.56e+1(7.51e+0) |
|  | 10 | 1.52e-1(3.01e-2) | 1.67e-1(4.03e-2) | 1.58e-1(3.11e-2) | 1.58e-1(3.11e-2) | 1.64e-1(3.19e-2) |
| DTLZ2 | 3 | 4.40e-2(3.06e-3) | 4.38e-2(4.17e-3) | 4.37e-2(3.41e-3) | 4.48e-2(3.51e-3) | 4.29e-2(4.38e-3) |
|  | 6 | 1.84e-1(1.50e-2) | 1.79e-1(1.02e-2) | 1.80e-1(1.17e-2) | 1.79e-1(9.20e-3) | 1.80e-1(1.49e-2) |
|  | 10 | 2.89e-1(1.00e-2) | 2.97e-1(1.40e-2) | 2.87e-1(1.71e-2) | 2.90e-1(1.22e-2) | 2.85e-1(1.09e-2) |
| DTLZ3 | 3 | 1.89e+2(4.68e+1) | 1.61e+2(3.71e+1) | 1.54e+2(4.89e+1) | 1.58e+2(3.45e+1) | 1.57e+2(3.17e+1) |
|  | 6 | 7.44e+1(2.34e+1) | 6.06e+1(1.32e+1) | 6.01e+1(2.61e+1) | 6.65e+1(2.14e+1) | 6.44e+1(2.63e+1) |
|  | 10 | 4.65e-1(1.12e-1) | 4.70e-1(8.67e-2) | 4.84e-1(5.71e-2) | 4.92e-1(1.38e-1) | 4.61e-1(4.94e-2) |
| DTLZ4 | 3 | 8.66e-2(1.25e-1) | 1.35e-1(1.64e-1) | 1.06e-1(1.32e-1) | 8.82e-2(1.26e-1) | 1.04e-1(1.28e-1) |
|  | 6 | 1.69e-1(2.20e-2) | 1.80e-1(3.27e-2) | 1.79e-1(4.06e-2) | 1.81e-1(4.77e-2) | 1.79e-1(2.78e-2) |
|  | 10 | 2.29e-1(1.15e-2) | 2.30e-1(1.15e-2) | 2.38e-1(1.56e-2) | 2.37e-1(2.00e-2) | 2.37e-1(1.88e-2) |
| DTLZ5 | 3 | 9.75e-3(2.19e-3) | 8.93e-3(1.67e-3) | 8.98e-3(1.67e-3) | 9.15e-3(1.58e-3) | 8.80e-3(1.44e-3) |
|  | 6 | 3.12e-2(9.30e-3) | 2.98e-2(1.02e-2) | 3.31e-2(7.84e-3) | 2.72e-2(7.30e-3) | 3.00e-2(1.05e-2) |
|  | 10 | 5.60e-3(1.76e-3) | 3.92e-3(6.78e-4) | 4.85e-3(1.78e-3) | 5.65e-3(2.12e-3) | 6.02e-3(1.70e-3) |
| DTLZ6 | 3 | 4.87e-2(2.65e-2) | 4.28e-2(2.73e-2) | 6.38e-2(7.62e-2) | 9.93e-2(2.14e-1) | 5.04e-2(3.71e-2) |
|  | 6 | 1.09e+0(1.19e+0) | 1.11e+0(1.07e+0) | 7.28e-1(1.00e+0) | 1.01e+0(1.13e+0) | 8.36e-1(1.16e+0) |
|  | 10 | 2.25e-2(7.14e-3) | 6.20e-2(5.11e-2) | 3.92e-2(3.62e-2) | 3.51e-2(3.23e-2) | 4.42e-2(4.00e-2) |
| DTLZ7 | 3 | 6.96e-2(3.03e-2) | 7.83e-2(5.28e-2) | 1.36e-1(1.32e-1) | 1.28e-1(1.31e-1) | 9.71e-2(5.24e-2) |
|  | 6 | 6.96e-1(2.65e-1) | 1.68e+0(8.29e-1) | 1.21e+0(7.32e-1) | 1.16e+0(6.33e-1) | 1.74e+0(8.02e-1) |
|  | 10 | 1.24e+0(1.54e-1) | 1.20e+0(9.84e-2) | 1.23e+0(1.33e-1) | 1.20e+0(8.92e-2) | 1.25e+0(1.08e-1) |
| WFG1 | 3 | 1.67e+0(4.91e-2) | 1.64e+0(5.90e-2) | 1.67e+0(4.86e-2) | 1.62e+0(3.43e-2) | 1.61e+0(4.98e-2) |
|  | 6 | 2.27e+0(5.70e-2) | 2.24e+0(5.05e-2) | 2.21e+0(5.69e-2) | 2.21e+0(7.43e-2) | 2.20e+0(6.16e-2) |
|  | 10 | 2.67e+0(8.46e-2) | 2.56e+0(1.07e-1) | 2.55e+0(1.15e-1) | 2.64e+0(7.62e-2) | 2.61e+0(8.36e-2) |
| WFG2 | 3 | 2.63e-1(3.41e-2) | 2.63e-1(3.89e-2) | 2.48e-1(5.57e-2) | 2.47e-1(4.40e-2) | 2.44e-1(5.40e-2) |
|  | 6 | 5.17e-1(1.03e-1) | 5.43e-1(1.35e-1) | 5.35e-1(9.94e-2) | 5.24e-1(1.26e-1) | 5.09e-1(1.49e-1) |
|  | 10 | 1.39e+0(4.37e-1) | 1.39e+0(3.77e-1) | 1.36e+0(3.13e-1) | 1.40e+0(2.71e-1) | 1.38e+0(3.83e-1) |
| WFG3 | 3 | 2.57e-1(3.61e-2) | 2.64e-1(7.85e-2) | 2.51e-1(3.82e-2) | 2.78e-1(5.66e-2) | 2.48e-1(2.96e-2) |
|  | 6 | 6.25e-1(1.13e-1) | 5.89e-1(6.72e-2) | 5.83e-1(8.20e-2) | 5.80e-1(7.49e-2) | 6.56e-1(1.04e-1) |
|  | 10 | 6.67e-1(8.95e-2) | 6.93e-1(9.45e-2) | 6.93e-1(1.22e-1) | 7.03e-1(9.06e-2) | 7.47e-1(8.54e-2) |
| WFG4 | 3 | 2.56e-1(3.27e-2) | 2.49e-1(2.04e-2) | 2.49e-1(2.61e-2) | 2.48e-1(1.75e-2) | 2.41e-1(1.77e-2) |
|  | 6 | 1.30e+0(1.91e-1) | 1.34e+0(2.28e-1) | 1.35e+0(3.15e-1) | 1.20e+0(2.23e-1) | 1.38e+0(2.88e-1) |
|  | 10 | 3.68e+0(6.78e-1) | 3.87e+0(7.96e-1) | 3.86e+0(6.03e-1) | 3.83e+0(7.38e-1) | 3.65e+0(3.90e-1) |
| WFG5 | 3 | 3.17e-1(1.22e-1) | 3.50e-1(1.07e-1) | 3.06e-1(1.05e-1) | 3.12e-1(1.25e-1) | 2.92e-1(1.28e-1) |
|  | 6 | 1.78e+0(9.49e-2) | 1.76e+0(1.11e-1) | 1.72e+0(1.26e-1) | 1.73e+0(9.61e-2) | 1.74e+0(1.33e-1) |
|  | 10 | 3.79e+0(2.92e-1) | 3.59e+0(2.81e-1) | 3.63e+0(4.80e-1) | 3.87e+0(3.19e-1) | 3.79e+0(2.71e-1) |
| WFG6 | 3 | 4.48e-1(1.00e-1) | 5.24e-1(1.08e-1) | 4.87e-1(1.00e-1) | 4.86e-1(9.23e-2) | 4.64e-1(9.08e-2) |
|  | 6 | 1.65e+0(1.84e-1) | 1.63e+0(8.15e-2) | 1.62e+0(1.85e-1) | 1.61e+0(1.48e-1) | 1.59e+0(2.47e-1) |
|  | 10 | 3.35e+0(4.95e-1) | 3.51e+0(3.14e-1) | 3.19e+0(2.14e-1) | 3.33e+0(3.76e-1) | 3.14e+0(5.76e-1) |
| WFG7 | 3 | 2.90e-1(3.37e-2) | 3.14e-1(3.26e-2) | 2.95e-1(2.76e-2) | 2.95e-1(2.68e-2) | 2.90e-1(3.27e-2) |
|  | 6 | 1.62e+0(2.02e-1) | 1.72e+0(1.37e-1) | 1.58e+0(1.47e-1) | 1.61e+0(1.63e-1) | 1.64e+0(1.85e-1) |
|  | 10 | 4.55e+0(3.72e-1) | 4.81e+0(3.13e-1) | 4.76e+0(4.89e-1) | 4.82e+0(3.93e-1) | 4.51e+0(2.58e-1) |
| WFG8 | 3 | 5.91e-1(6.73e-2) | 6.06e-1(5.44e-2) | 5.71e-1(4.02e-2) | 5.77e-1(3.92e-2) | 5.61e-1(3.98e-2) |
|  | 6 | 2.20e+0(1.50e-1) | 2.20e+0(1.48e-1) | 2.21e+0(1.21e-1) | 2.24e+0(1.57e-1) | 2.16e+0(1.06e-1) |
|  | 10 | 4.99e+0(4.45e-1) | 5.15e+0(4.48e-1) | 5.06e+0(5.80e-1) | 5.00e+0(3.93e-1) | 4.90e+0(5.04e-1) |
| WFG9 | 3 | 3.68e-1(1.03e-1) | 4.43e-1(1.41e-1) | 3.81e-1(1.02e-1) | 3.85e-1(9.50e-2) | 3.56e-1(6.48e-2) |
|  | 6 | 1.54e+0(1.81e-1) | 1.51e+0(1.73e-1) | 1.48e+0(2.27e-1) | 1.45e+0(1.19e-1) | 1.48e+0(1.75e-1) |
|  | 10 | 4.02e+0(2.34e-1) | 3.97e+0(4.11e-1) | 4.02e+0(4.62e-1) | 3.94e+0(3.94e-1) | 3.96e+0(3.20e-1) |
| +/ ≈ /− | $n_c$=1 | -/-/- | 2/43/3 | 1/41/6 | 1/42/5 | 3/41/4 |
| +/ ≈ /− | $n_c$=3 | 3/43/2 | -/-/- | 0/46/2 | 2/45/1 | 1/41/6 |
| +/ ≈ /− | $n_c$=5 | 6/41/1 | 2/46/0 | -/-/- | 1/45/2 | 2/45/1 |
| +/ ≈ /− | $n_c$=7 | 5/42/1 | 1/45/2 | 2/45/1 | -/-/- | 2/45/1 |
| +/ ≈ /− | $n_c$=10 | 4/41/3 | 6/41/1 | 1/45/2 | 1/45/2 | -/-/- |

## H.2 IGD+ RESULTS ON DTLZ AND WFG OPTIMIZATION PROBLEMS

Tables 8 and 9 display the IGD+ optimization results of comparison algorithms on DTLZ and WFG optimization problems, respectively. Different from IGD results, although LORA-MaOO achieves the smallest IGD+ values on most DTLZ problems, its perform is competitive to KRVEA and KTA2 on WFG problems. However, from the perspective of overall performance, we can still conclude that our LORA-MaOO outperforms all comparison algorithms on benchmark optimization problems in terms of IGD+ values. Such a observation is consistent with the results we observed from IGD values.

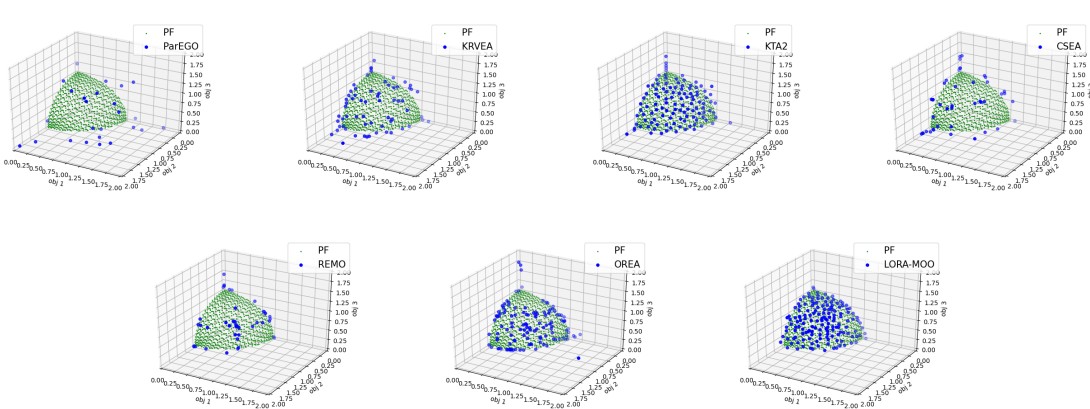

Figure 7: Distribution of obtained non-dominated solutions on DTLZ2 with 10 variables and 3 objectives.

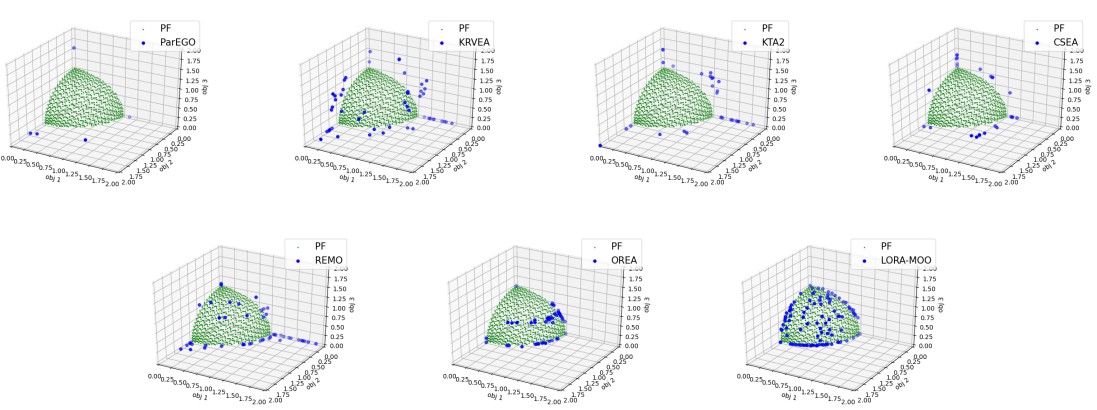

Figure 8: Distribution of obtained non-dominated solutions on DTLZ4 with 10 variables and 3 objectives.

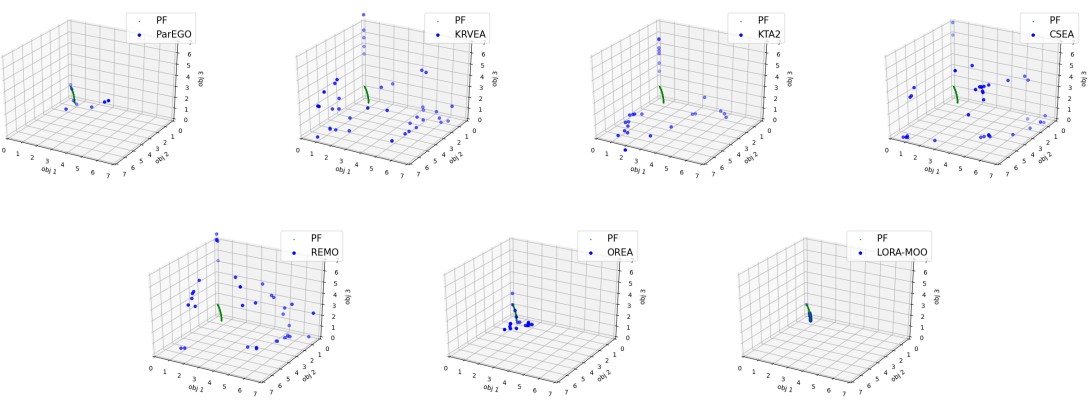

Figure 9: Distribution of obtained non-dominated solutions on DTLZ6 with 10 variables and 3 objectives.

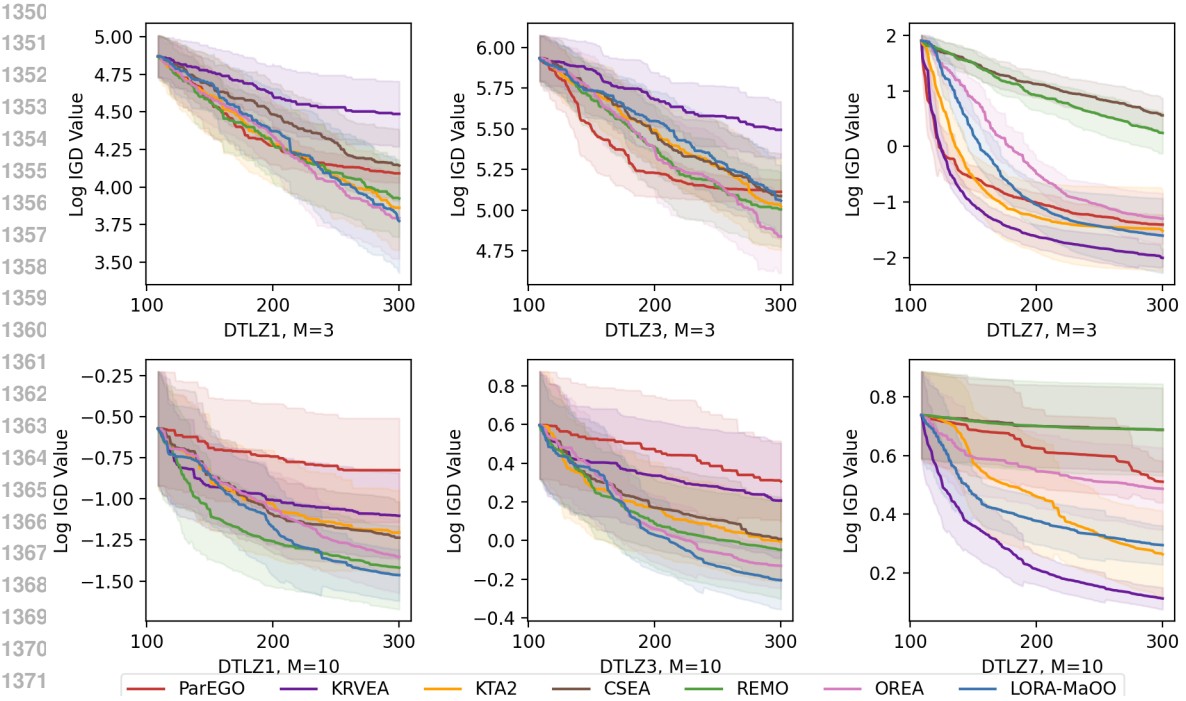

Figure 10: Log (IGD) curves averaged over 30 runs on DTLZ1, DTLZ3, and DTLZ7 for comparison algorithms (shaded area is $\pm$ std of the mean).

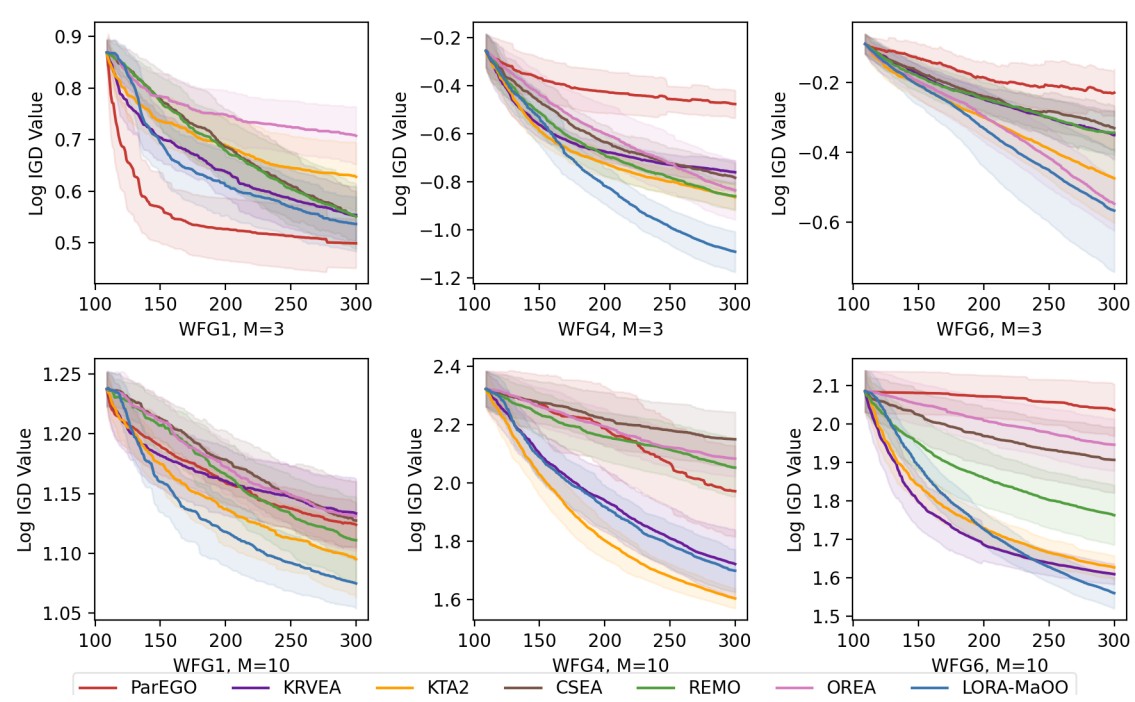

Figure 11: Log (IGD) curves averaged over 30 runs on WFG1, WFG4, and WFG6 for comparison algorithms (shaded area is $\pm$ std of the mean).

### H.3 HV RESULTS ON DTLZ AND WFG OPTIMIZATION PROBLEMS

Tables 10 and 11 report the HV optimization results of comparison algorithms on DTLZ and WFG optimization problems, respectively. Since the calculation of HV values on 8- and 10-obj opti-

Table 7: Statistical results of the IGD value obtained by comparison algorithms on 45 WFG optimization problems over 30 runs. Symbols '+', '≈', '−' denote LORA-MaOO is statistically significantly superior to, almost equivalent to, and inferior to the compared algorithms in the Wilcoxon rank sum test (significance level is 0.05), respectively. The last row counts the total win/tie/loss results.

| Problems | M | ParEGO | KRVEA | KTA2 | CSEA | REMO | OREA | LORA-MaOO |
|---|---|---|---|---|---|---|---|---|
| WFG1 | 3 | 1.65e+0(8.08e-2)− | 1.74e+0(9.91e-2)≈ | 1.87e+0(1.27e-1)+ | 1.74e+0(8.60e-2)≈ | 1.73e+0(1.12e-1)≈ | 2.03e+0(1.16e-1)+ | 1.71e+0(9.26e-2) |
|  | 4 | 1.94e+0(7.04e-2)≈ | 2.07e+0(9.03e-2)+ | 2.18e+0(1.43e-1)+ | 2.05e+0(1.05e-1)+ | 1.96e+0(8.19e-2)≈ | 2.22e+0(9.54e-2)+ | 1.95e+0(7.52e-2) |
|  | 6 | 2.38e+0(5.53e-2)≈ | 2.49e+0(6.57e-2)+ | 2.56e+0(9.95e-2)+ | 2.52e+0(9.89e-2)+ | 2.42e+0(5.34e-2)+ | 2.53e+0(1.04e-1)+ | 2.36e+0(5.07e-2) |
|  | 8 | 2.75e+0(5.21e-2)+ | 2.86e+0(7.05e-2)+ | 2.85e+0(1.06e-1)+ | 2.89e+0(5.19e-2)+ | 2.80e+0(7.44e-2)+ | 2.82e+0(7.56e-2)+ | 2.72e+0(6.21e-2) |
|  | 10 | 3.08e+0(5.70e-2)+ | 3.11e+0(9.16e-2)+ | 2.99e+0(9.77e-2)+ | 3.09e+0(1.03e-1)+ | 3.04e+0(1.12e-1)+ | 3.10e+0(9.11e-2)+ | 2.93e+0(6.20e-2) |
| WFG2 | 3 | 7.66e-1(7.11e-2)+ | 3.61e-1(3.87e-2)≈ | 4.24e-1(6.65e-2)+ | 5.48e-1(3.75e-2)+ | 5.22e-1(7.67e-2)+ | 4.88e-1(6.53e-2)+ | 3.72e-1(4.87e-2) |
|  | 4 | 1.05e+0(1.40e-1)+ | 5.00e-1(3.97e-2)− | 5.66e-1(3.80e-2)+ | 7.61e-1(1.21e-1)+ | 7.48e-1(1.23e-1)+ | 7.45e-1(1.45e-1)+ | 5.46e-1(3.53e-2) |
|  | 6 | 1.90e+0(3.51e-1)+ | 7.77e-1(5.25e-2)− | 9.00e-1(5.39e-2)+ | 1.28e+0(4.02e-1)+ | 1.28e+0(3.75e-1)+ | 1.49e+0(3.76e-1)+ | 8.55e-1(7.00e-2) |
|  | 8 | 2.74e+0(6.68e-1)+ | 1.06e+0(5.98e-2)− | 1.18e+0(1.14e-1)− | 2.10e+0(6.97e-1)+ | 1.90e+0(5.25e-1)+ | 2.06e+0(4.58e-1)+ | 1.24e+0(1.23e-1) |
|  | 10 | 3.73e+0(9.41e-1)+ | 1.18e+0(9.32e-2)− | 1.37e+0(1.03e-1)− | 2.84e+0(8.61e-1)+ | 2.59e+0(9.91e-1)+ | 2.95e+0(7.55e-1)+ | 1.83e+0(2.27e-1) |
| WFG3 | 3 | 5.82e-1(3.86e-2)+ | 5.39e-1(5.81e-2)+ | 3.29e-1(5.99e-2)+ | 5.04e-1(6.26e-2)+ | 4.60e-1(5.94e-2)+ | 3.85e-1(4.76e-2)+ | 2.83e-1(5.99e-2) |
|  | 4 | 7.30e-1(6.25e-2)+ | 6.66e-1(7.02e-2)+ | 5.63e-1(6.47e-2)+ | 6.05e-1(7.26e-2)+ | 5.64e-1(6.43e-2)+ | 5.68e-1(5.92e-2)+ | 4.13e-1(5.98e-2) |
|  | 6 | 7.75e-1(9.36e-2)+ | 6.76e-1(1.32e-1)+ | 7.94e-1(6.73e-2)+ | 7.41e-1(8.33e-2)+ | 6.37e-1(9.55e-2)+ | 7.96e-1(6.68e-2)+ | 6.51e-1(9.20e-2) |
|  | 8 | 8.38e-1(1.63e-1)− | 8.27e-1(9.79e-2)+ | 9.45e-1(7.42e-2)+ | 7.63e-1(1.06e-1)− | 6.25e-1(1.18e-1)− | 8.92e-1(9.90e-2)≈ | 8.54e-1(9.98e-2) |
|  | 10 | 6.85e-1(1.02e-1)− | 6.87e-1(8.79e-2)− | 9.16e-1(8.20e-2)+ | 5.91e-1(9.34e-2)− | 5.19e-1(1.04e-1)− | 7.28e-1(1.10e-1)− | 8.23e-1(1.14e-1) |
| WFG4 | 3 | 6.21e-1(3.68e-2)+ | 4.67e-1(2.33e-2)+ | 4.21e-1(2.21e-2)+ | 4.57e-1(2.88e-2)+ | 4.23e-1(2.53e-2)+ | 4.34e-1(5.63e-2)+ | 3.36e-1(2.95e-2) |
|  | 4 | 1.11e+0(3.45e-2)+ | 7.86e-1(2.45e-2)+ | 7.78e-1(4.50e-2)+ | 9.83e-1(1.22e-1)+ | 8.46e-1(8.32e-2)+ | 1.07e+0(1.18e-1)+ | 6.82e-1(4.97e-2) |
|  | 6 | 2.75e+0(2.36e-1)+ | 1.87e+0(8.92e-2)≈ | 1.78e+0(7.66e-2)− | 3.13e+0(3.86e-1)+ | 2.69e+0(3.61e-1)+ | 2.92e+0(3.04e-1)+ | 1.86e+0(1.30e-1) |
|  | 8 | 5.09e+0(9.78e-1)+ | 3.47e+0(2.96e-1)− | 3.26e+0(1.67e-1)− | 5.81e+0(5.38e-1)+ | 4.99e+0(4.67e-1)+ | 5.76e+0(4.34e-1)+ | 3.62e+0(3.31e-1) |
|  | 10 | 7.18e+0(1.21e+0)+ | 5.60e+0(6.92e-1)≈ | 4.97e+0(1.72e-1)− | 8.58e+0(8.39e-1)+ | 7.78e+0(8.13e-1)+ | 8.03e+0(5.03e-1)+ | 5.47e+0(4.14e-1) |
| WFG5 | 3 | 4.21e-1(3.05e-2)+ | 3.91e-1(4.22e-2)+ | 3.30e-1(9.56e-2)− | 5.50e-1(3.05e-2)+ | 5.30e-1(1.46e-2)+ | 4.51e-1(6.51e-2)+ | 4.21e-1(1.35e-1) |
|  | 4 | 9.98e-1(8.09e-2)+ | 7.65e-1(2.86e-2)− | 7.20e-1(6.23e-2)− | 8.87e-1(3.98e-2)− | 8.61e-1(4.68e-2)− | 1.02e+0(4.57e-2)+ | 9.81e-1(5.76e-2) |
|  | 6 | 2.82e+0(1.65e-1)+ | 1.78e+0(6.23e-2)− | 1.92e+0(1.03e-1)− | 2.35e+0(1.86e-1)+ | 2.04e+0(1.29e-1)− | 2.44e+0(1.08e-1)+ | 2.11e+0(9.10e-2) |
|  | 8 | 5.25e+0(2.55e-1)+ | 3.30e+0(2.61e-1)− | 3.62e+0(2.64e-1)+ | 4.75e+0(3.77e-1)+ | 3.95e+0(2.83e-1)+ | 4.57e+0(1.82e-1)+ | 3.66e+0(9.43e-2) |
|  | 10 | 7.64e+0(3.23e-1)+ | 4.67e+0(4.78e-1)− | 4.76e+0(1.99e-1)− | 6.88e+0(4.23e-1)+ | 6.11e+0(4.62e-1)+ | 6.68e+0(3.49e-1)+ | 4.98e+0(1.57e-1) |
| WFG6 | 3 | 7.96e-1(5.50e-2)+ | 7.05e-1(5.10e-2)+ | 6.22e-1(8.49e-2)+ | 7.19e-1(4.80e-2)+ | 7.09e-1(4.61e-2)+ | 5.79e-1(4.68e-2)+ | 5.67e-1(1.09e-1) |
|  | 4 | 1.14e+0(4.72e-2)+ | 1.02e+0(4.90e-2)+ | 9.62e-1(4.46e-2)+ | 1.08e+0(4.82e-2)+ | 1.04e+0(4.53e-2)+ | 1.17e+0(4.94e-2)+ | 9.51e-1(9.85e-2) |
|  | 6 | 2.81e+0(2.60e-1)+ | 2.18e+0(7.41e-2)+ | 1.96e+0(4.17e-2)− | 2.56e+0(2.16e-1)+ | 2.20e+0(1.61e-1)+ | 2.77e+0(1.81e-1)+ | 2.04e+0(9.86e-2) |
|  | 8 | 4.70e+0(5.78e-1)+ | 3.60e+0(1.17e-1)+ | 3.54e+0(1.85e-1)≈ | 4.70e+0(5.18e-1)+ | 4.13e+0(3.06e-1)+ | 5.06e+0(3.20e-1)+ | 3.52e+0(1.52e-1) |
|  | 10 | 7.66e+0(5.36e-1)+ | 5.00e+0(1.33e-1)+ | 5.09e+0(1.58e-1)+ | 6.73e+0(5.98e-1)+ | 5.83e+0(4.69e-1)+ | 7.00e+0(4.90e-1)+ | 4.76e+0(1.94e-1) |
| WFG7 | 3 | 6.69e-1(2.70e-2)+ | 6.28e-1(2.45e-2)+ | 5.73e-1(2.76e-2)+ | 5.78e-1(3.23e-2)+ | 5.38e-1(3.58e-2)+ | 4.43e-1(4.15e-2)+ | 3.52e-1(2.22e-2) |
|  | 4 | 1.13e+0(4.94e-2)+ | 9.48e-1(2.66e-2)+ | 9.04e-1(2.51e-2)+ | 9.92e-1(8.75e-2)+ | 8.81e-1(3.49e-2)+ | 9.72e-1(7.29e-2)+ | 7.07e-1(4.29e-2) |
|  | 6 | 3.17e+0(2.89e-1)+ | 2.00e+0(5.61e-2)≈ | 1.96e+0(5.97e-2)≈ | 2.71e+0(3.18e-1)+ | 2.18e+0(1.49e-1)+ | 2.71e+0(1.91e-1)+ | 1.96e+0(1.06e-1) |
|  | 8 | 5.93e+0(3.95e-1)+ | 3.64e+0(1.23e-1)− | 3.37e+0(1.16e-1)− | 5.19e+0(5.20e-1)+ | 4.28e+0(4.59e-1)+ | 5.19e+0(3.07e-1)+ | 3.82e+0(1.63e-1) |
|  | 10 | 8.78e+0(4.70e-1)+ | 5.31e+0(3.01e-1)− | 4.88e+0(1.76e-1)− | 8.07e+0(5.93e-1)+ | 6.77e+0(5.93e-1)+ | 7.57e+0(4.12e-1)+ | 5.73e+0(3.07e-1) |
| WFG8 | 3 | 8.45e-1(2.87e-2)+ | 6.42e-1(2.49e-2)+ | 5.09e-1(4.39e-2)− | 7.49e-1(4.33e-2)+ | 7.13e-1(3.87e-2)+ | 7.01e-1(4.35e-2)+ | 6.02e-1(3.64e-2) |
|  | 4 | 1.33e+0(4.61e-2)+ | 1.14e+0(3.89e-2)+ | 1.02e+0(9.39e-2)− | 1.26e+0(6.23e-2)+ | 1.20e+0(5.28e-2)+ | 1.36e+0(6.94e-2)+ | 1.13e+0(7.12e-2) |
|  | 6 | 3.11e+0(2.82e-1)+ | 2.43e+0(7.15e-2)≈ | 2.28e+0(5.05e-2)− | 3.00e+0(1.53e-1)+ | 2.80e+0(1.90e-1)+ | 3.07e+0(1.74e-1)+ | 2.45e+0(9.73e-2) |
|  | 8 | 5.74e+0(3.56e-1)+ | 4.01e+0(2.28e-1)− | 3.92e+0(1.28e-1)− | 5.56e+0(3.24e-1)+ | 5.11e+0(4.10e-1)+ | 5.34e+0(2.72e-1)+ | 4.22e+0(2.75e-1) |
|  | 10 | 8.30e+0(4.83e-1)+ | 5.56e+0(5.40e-1)− | 5.71e+0(3.80e-1)+ | 7.81e+0(4.74e-1)+ | 7.32e+0(3.46e-1)+ | 7.54e+0(4.88e-1)+ | 5.82e+0(2.95e-1) |
| WFG9 | 3 | 7.14e-1(5.09e-2)+ | 6.75e-1(6.73e-2)+ | 6.37e-1(8.35e-2)+ | 6.74e-1(8.53e-2)+ | 6.11e-1(9.76e-2)+ | 5.12e-1(7.74e-2)+ | 4.34e-1(8.18e-2) |
|  | 4 | 1.24e+0(1.41e-1)+ | 1.06e+0(8.72e-2)+ | 1.07e+0(9.28e-2)+ | 1.16e+0(1.18e-1)+ | 1.05e+0(1.61e-1)+ | 1.02e+0(7.89e-2)+ | 8.43e-1(9.25e-2) |
|  | 6 | 3.14e+0(2.96e-1)+ | 2.22e+0(1.94e-1)+ | 2.19e+0(1.52e-1)+ | 2.83e+0(2.36e-1)+ | 2.30e+0(1.82e-1)+ | 2.55e+0(1.21e-1)+ | 1.97e+0(9.18e-2) |
|  | 8 | 5.78e+0(4.51e-1)+ | 3.93e+0(3.00e-1)+ | 3.77e+0(2.23e-1)+ | 5.43e+0(3.68e-1)+ | 4.60e+0(3.92e-1)+ | 4.73e+0(3.07e-1)+ | 3.61e+0(2.05e-1) |
|  | 10 | 8.41e+0(4.80e-1)+ | 5.69e+0(6.42e-1)+ | 5.26e+0(3.13e-1)≈ | 7.77e+0(5.05e-1)+ | 6.48e+0(5.60e-1)+ | 6.74e+0(4.17e-1)+ | 5.16e+0(2.60e-1) |
| +/≈/− | | 39/4/2 | 21/10/14 | 23/6/16 | 41/1/3 | 38/3/4 | 43/1/1 | |

mization problems is very time-consuming, only the results obtained on 3-, 4-, and 6-objective optimization problems are displayed. Consistent with the IGD an IGD+ results obtained on 3-, 4-, and 6-objectives, our LORA-MaOO achieves the best overall performance over all comparison algorithms, showing the effectiveness of LORA-MaOO on addressing expensive many-objective optimization problems.

## H.4 Problems with Different Scales

In this subsection, we investigate the optimization performance of LORA-MaOO when the number of decision variables $D$ is different. The experimental setups for all comparison algorithms are the same as the setups used in previous benchmark optimization problems, but the setup for optimization problems is different:

- The optimization problems have $D = \{5, 10, 20\}$ decision variables and $M = 3$ objectives.
- When $D = 5$ or 10, a dataset of size 11 $D$ - 1 is used for surrogate initialization. When $D = 20$, since 11 $D$ - 1 would be greater than our evaluation budget (300), the size of initial dataset is set to 100.

Tables 12, 13, and 14 report the obtained IGD, IGD+, and HV values on benchmark optimization problems with different numbers of decision variables $D$, respectively. It can be seen from Table 12 that LORA-MaOO outperforms all comparison algorithms on DTLZ optimization problems when $D = 5$, 10, and 20. In addition, KTA2 reaches competitive optimization results on many optimization problems. The observations from Tables 13 and 14 have demonstrated consistent conclusions.

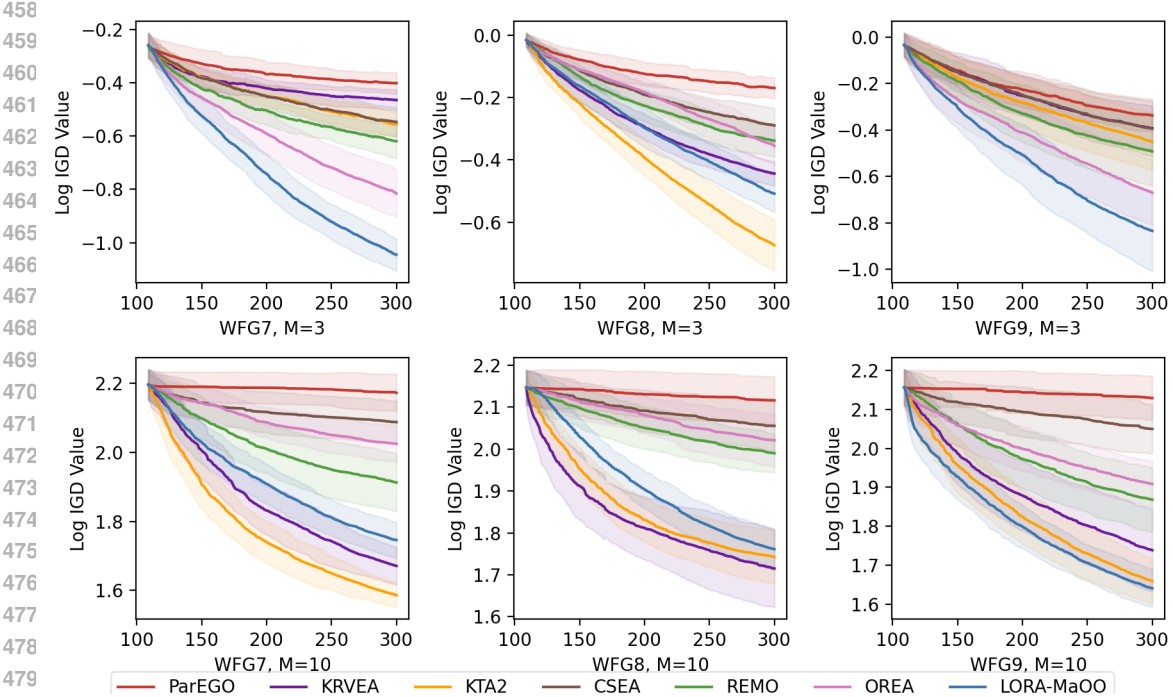

Figure 12: Log (IGD) curves averaged over 30 runs on WFG7, WFG8, and WFG9 for comparison algorithms (shaded area is ± std of the mean).

Table 8: Statistical results of the IGD+ value obtained by comparison algorithms on 35 DTLZ optimization problems over 30 runs. Symbols '+', '≈', '−' denote LORA-MaOO is statistically significantly superior to, almost equivalent to, and inferior to the compared algorithms in the Wilcoxon rank sum test (significance level is 0.05), respectively. The last row counts the total win/tie/loss results.

| Problems | M | ParEGO | KRVEA | KTA2 | CSEA | REMO | OREA | LORA-MaOO |
|---|---|---|---|---|---|---|---|---|
| DTLZ1 | 3 | 5.98e+1(3.81e+0)+ | 8.88e+1(2.16e+1)+ | 4.75e+1(1.55e+1)≈ | 6.30e+1(1.69e+1)+ | 5.06e+1(1.49e+1)+ | 4.44e+1(1.38e+1)≈ | 4.35e+1(1.80e+1) |
| | 4 | 4.68e+1(3.71e+0)+ | 6.45e+1(1.47e+1)+ | 4.08e+1(1.60e+1)+ | 3.69e+1(1.08e+1)≈ | 3.92e+1(1.11e+1)≈ | 3.80e+1(1.23e+1)+ | 4.06e+1(1.34e+1) |
| | 6 | 3.04e+1(2.74e+0)+ | 3.22e+1(7.66e+0)+ | 2.03e+1(8.12e+0)+ | 1.56e+1(4.96e+0)≈ | 1.22e+1(4.65e+0)− | 1.74e+1(3.98e+0)≈ | 1.58e+1(6.17e+0) |
| | 8 | 1.23e+1(2.99e+0)+ | 8.52e+0(2.98e+0)+ | 4.54e+0(2.66e+0)≈ | 5.08e+0(2.47e+0)≈ | 3.33e+0(1.93e+0)≈ | 5.87e+0(2.91e+0)+ | 3.82e+0(2.35e+0) |
| | 10 | 3.82e-1(1.79e-1)+ | 2.76e-1(1.14e-1)+ | 2.33e-1(9.65e-2)+ | 2.22e-1(8.29e-2)+ | 1.75e-1(7.84e-2)≈ | 1.83e-1(6.73e-2)≈ | 1.56e-1(3.41e-2) |
| DTLZ2 | 3 | 2.61e-1(3.63e-2)+ | 9.22e-2(2.57e-2)+ | 3.82e-2(3.29e-3)− | 1.60e-1(2.76e-2)+ | 1.01e-1(1.75e-2)+ | 5.86e-2(8.28e-3)+ | 4.47e-2(2.35e-3) |
| | 4 | 3.55e-1(4.11e-2)+ | 1.30e-1(3.08e-2)+ | 9.05e-2(6.95e-3)+ | 2.05e-1(2.43e-2)+ | 1.60e-1(3.01e-2)+ | 1.37e-1(1.61e-2)+ | 9.74e-2(1.14e-2) |
| | 6 | 4.47e-1(2.32e-2)+ | 1.82e-1(1.49e-2)≈ | 2.36e-1(3.71e-2)+ | 3.15e-1(4.24e-2)+ | 2.64e-1(3.18e-2)+ | 3.21e-1(2.78e-2)+ | 1.82e-1(1.15e-2) |
| | 8 | 4.68e-1(1.49e-2)+ | 2.34e-1(1.91e-2)+ | 3.43e-1(2.37e-2)+ | 3.95e-1(2.66e-2)+ | 3.42e-1(2.91e-2)+ | 4.19e-1(1.86e-2)+ | 2.58e-1(1.88e-2) |
| | 10 | 4.33e-1(2.26e-2)+ | 2.92e-1(3.09e-2)≈ | 3.15e-1(1.47e-2)+ | 4.17e-1(2.03e-2)+ | 3.61e-1(2.70e-2)+ | 4.28e-1(1.61e-2)+ | 2.88e-1(1.27e-2) |
| DTLZ3 | 3 | 1.66e+2(1.31e+1)+ | 2.43e+2(4.61e+1)+ | 1.52e+2(4.73e+1)≈ | 1.62e+2(4.84e+1)+ | 1.49e+2(3.88e+1)+ | 1.26e+2(3.18e+1)− | 1.57e+2(3.83e+1) |
| | 4 | 1.42e+2(1.57e+1)+ | 1.83e+2(4.00e+1)+ | 1.18e+2(3.49e+1)+ | 1.29e+2(3.58e+1)≈ | 1.16e+2(3.00e+1)≈ | 1.22e+2(4.13e+1)≈ | 1.25e+2(4.20e+1) |
| | 6 | 9.17e+1(1.59e+1)+ | 1.06e+2(2.96e+1)+ | 6.65e+1(2.63e+1)+ | 5.27e+1(1.56e+1)+ | 5.23e+1(1.71e+1)+ | 5.24e+1(1.68e+1)+ | 5.96e+1(2.05e+1) |
| | 8 | 4.13e+1(9.84e+0)+ | 2.96e+1(1.15e+1)+ | 1.73e+1(1.10e+1)+ | 1.59e+1(9.77e+0)+ | 1.60e+1(1.71e+0)≈ | 1.49e+1(6.28e+0)+ | 1.26e+1(8.35e+0) |
| | 10 | 1.08e+0(3.73e-1)+ | 9.96e-1(4.96e-1)+ | 7.29e-1(2.75e-1)+ | 6.94e-1(2.89e-1)+ | 6.89e-1(3.18e-1)+ | 5.27e-1(6.34e-2)+ | 4.75e-1(1.13e-1) |
| DTLZ4 | 3 | 4.57e-1(7.52e-2)+ | 2.66e-1(1.02e-1)+ | 2.33e-1(8.36e-2)+ | 2.34e-1(7.76e-2)+ | 1.32e-1(6.41e-2)+ | 1.07e-1(9.68e-2)+ | 8.96e-2(1.25e-1) |
| | 4 | 4.86e-1(5.76e-2)+ | 2.84e-1(7.44e-2)+ | 2.95e-1(6.34e-2)+ | 2.03e-1(3.78e-2)+ | 1.66e-1(3.40e-2)+ | 1.35e-1(9.87e-2)+ | 1.37e-1(9.79e-2) |
| | 6 | 4.24e-1(4.26e-2)+ | 2.94e-1(5.11e-2)+ | 3.61e-1(7.84e-2)+ | 2.41e-1(3.82e-2)+ | 2.27e-1(3.26e-2)+ | 1.67e-1(2.62e-2)+ | 1.78e-1(4.02e-2) |
| | 8 | 3.53e-1(2.66e-2)+ | 2.67e-1(3.51e-2)+ | 3.33e-1(4.56e-2)+ | 2.78e-1(3.65e-2)+ | 2.93e-1(3.63e-2)+ | 2.09e-1(2.55e-2)+ | 2.08e-1(1.89e-2) |
| | 10 | 2.86e-1(1.61e-2)+ | 2.58e-1(2.11e-2)+ | 2.88e-1(3.27e-2)+ | 2.92e-1(2.16e-2)+ | 3.06e-1(2.71e-2)+ | 2.29e-1(1.41e-2)+ | 2.30e-1(1.70e-2) |
| DTLZ5 | 3 | 1.60e-1(4.40e-2)+ | 9.18e-2(2.76e-2)+ | 8.66e-3(1.96e-3)≈ | 9.58e-2(2.60e-2)+ | 5.78e-2(1.81e-2)+ | 1.59e-2(5.12e-3)+ | 9.40e-3(1.93e-3) |
| | 4 | 1.47e-1(3.58e-2)+ | 4.96e-2(1.98e-2)+ | 3.25e-2(9.50e-3)+ | 9.78e-2(2.16e-2)+ | 7.51e-2(2.55e-2)+ | 2.88e-2(7.46e-3)+ | 2.21e-2(7.30e-3) |
| | 6 | 1.08e-1(2.44e-2)+ | 2.24e-2(7.50e-3)− | 8.02e-2(2.16e-2)+ | 6.16e-2(2.49e-2)+ | 4.14e-2(1.76e-2)+ | 3.89e-2(1.47e-2)≈ | 3.20e-2(1.14e-2) |
| | 8 | 5.11e-2(7.70e-3)+ | 1.44e-2(5.17e-3)− | 5.35e-2(1.14e-2)+ | 2.49e-2(6.87e-3)+ | 2.01e-2(5.56e-3)≈ | 1.87e-2(2.32e-3)≈ | 1.87e-2(3.21e-3) |
| | 10 | 1.19e-2(1.01e-3)+ | 6.26e-3(9.09e-4)+ | 1.19e-2(1.80e-3)+ | 7.45e-3(9.85e-4)+ | 4.80e-3(1.09e-3)− | 5.48e-3(9.49e-4)+ | 5.62e-3(1.75e-3) |
| DTLZ6 | 3 | 2.42e-1(1.07e-1)+ | 3.05e+0(5.23e-1)+ | 1.82e+0(4.48e-1)+ | 4.85e+0(6.38e-1)+ | 4.27e+0(5.48e-1)+ | 2.35e-1(4.14e-1)+ | 6.74e-2(1.55e-1) |
| | 4 | 2.64e-1(1.83e-1)+ | 2.44e+0(4.90e-1)+ | 1.84e+0(5.17e-1)+ | 5.12e+0(4.31e-1)+ | 4.07e+0(6.25e-1)+ | 1.35e+0(9.45e-1)+ | 2.07e-1(2.06e-1) |
| | 6 | 1.78e-1(1.07e-1)− | 1.33e+0(2.80e-1)+ | 1.49e+0(5.98e-1)+ | 3.14e+0(4.44e-1)+ | 2.32e+0(5.72e-1)+ | 2.04e+0(6.34e-1)+ | 9.00e-1(1.07e+0) |
| | 8 | 8.31e-2(2.90e-2)≈ | 4.48e-1(1.88e-1)+ | 8.28e-1(4.14e-1)+ | 1.53e+0(4.64e-1)+ | 9.18e-1(4.68e-1)+ | 1.03e+0(4.26e-1)+ | 2.96e-1(4.46e-1) |
| | 10 | 8.21e-2(9.39e-2)+ | 3.08e-2(1.03e-2)≈ | 6.59e-2(5.61e-2)+ | 1.63e-1(2.40e-1)+ | 5.12e-2(1.09e-1)+ | 1.15e-1(7.35e-2)+ | 3.30e-2(2.86e-2) |
| DTLZ7 | 3 | 1.10e-1(3.57e-2)+ | 7.39e-2(1.52e-2)≈ | 1.54e-1(1.97e-1)− | 1.65e+0(6.43e-1)+ | 1.20e+0(5.73e-1)+ | 1.79e-1(1.20e-1)+ | 1.38e-1(1.53e-1) |
| | 4 | 4.98e-1(1.02e-1)+ | 2.20e-1(5.76e-2)≈ | 2.31e-1(1.27e-1)≈ | 2.82e+0(6.75e-1)+ | 1.96e+0(7.49e-1)+ | 7.18e-1(4.34e-1)+ | 2.80e-1(1.73e-1) |
| | 6 | 1.07e+0(1.62e-1)≈ | 4.31e-1(3.82e-2)− | 4.39e-1(1.48e-1)− | 4.80e+0(1.01e+0)+ | 2.93e+0(7.01e-1)+ | 3.96e+0(1.88e+0)+ | 1.46e+0(6.89e-1) |
| | 8 | 1.28e+0(1.27e-1)− | 6.29e-1(7.74e-2)− | 7.72e-1(1.53e-1)− | 6.03e+0(1.87e+0)+ | 3.63e+0(5.55e-1)+ | 4.40e+0(2.74e+0)+ | 2.25e+0(6.88e-1) |
| | 10 | 1.51e+0(1.37e-1)+ | 9.42e-1(4.54e-2)− | 1.11e+0(1.99e-1)− | 1.80e+0(3.39e-1)+ | 1.79e+0(3.78e-1)+ | 1.46e+0(2.55e-1)+ | 1.19e+0(8.31e-2) |
| +/≈/− | | 31/2/2 | 24/5/6 | 20/9/6 | 28/7/0 | 24/9/2 | 20/14/1 | |

Table 9: Statistical results of the IGD+ value obtained by comparison algorithms on 45 WFG optimization problems over 30 runs. Symbols '+', '≈', '−' denote LORA-MaOO is statistically significantly superior to, almost equivalent to, and inferior to the compared algorithms in the Wilcoxon rank sum test (significance level is 0.05), respectively. The last row counts the total win/tie/loss results.

| Problems | M | ParEGO | KRVEA | KTA2 | CSEA | REMO | OREA | LORA-MaOO |
|---|---|---|---|---|---|---|---|---|
| WFG1 | 3 | 1.62e+0(3.90e-2)≈ | 1.68e+0(9.09e-2)+ | 1.78e+0(1.38e-1)+ | 1.68e+0(7.59e-2)+ | 1.69e+0(1.08e-1)+ | 1.92e+0(1.27e-1)+ | 1.63e+0(3.69e-2) |
|  | 4 | 1.90e+0(6.54e-2)+ | 1.99e+0(1.02e-1)+ | 2.07e+0(1.47e-1)+ | 1.98e+0(1.06e-1)+ | 1.90e+0(8.14e-2)+ | 2.12e+0(8.95e-2)+ | 1.85e+0(7.27e-2) |
|  | 6 | 2.30e+0(4.35e-2)+ | 2.36e+0(7.09e-2)+ | 2.41e+0(1.08e-1)+ | 2.37e+0(9.06e-2)+ | 2.29e+0(7.24e-2)+ | 2.39e+0(8.81e-2)+ | 2.22e+0(6.71e-2) |
|  | 8 | 2.64e+0(4.48e-2)+ | 2.66e+0(7.65e-2)+ | 2.60e+0(1.15e-1)+ | 2.62e+0(6.82e-2)+ | 2.55e+0(6.82e-2)+ | 2.59e+0(4.96e-2)+ | 2.49e+0(7.00e-2) |
|  | 10 | 2.88e+0(6.44e-2)+ | 2.78e+0(9.91e-2)+ | 2.65e+0(1.26e-1)≈ | 2.71e+0(1.27e-1)+ | 2.71e+0(1.22e-1)+ | 2.78e+0(1.04e-1)+ | 2.62e+0(7.81e-2) |
| WFG2 | 3 | 6.99e-1(9.48e-2)+ | 2.58e-1(4.09e-2)≈ | 2.39e-1(7.01e-2)≈ | 4.68e-1(5.12e-2)+ | 4.30e-1(9.29e-2)+ | 3.95e-1(7.73e-2)+ | 2.47e-1(4.89e-2) |
|  | 4 | 9.74e-1(1.65e-1)+ | 3.21e-1(4.70e-2)− | 3.52e-1(5.16e-2)+ | 6.27e-1(1.42e-1)+ | 6.22e-1(1.45e-1)+ | 6.23e-1(1.69e-1)+ | 3.52e-1(5.74e-2) |
|  | 6 | 1.77e+0(4.19e-1)+ | 3.84e-1(7.38e-2)− | 5.75e-1(1.00e-1)+ | 1.02e+0(4.94e-1)+ | 1.01e+0(4.70e-1)+ | 1.33e+0(4.17e-1)+ | 5.29e-1(1.26e-1) |
|  | 8 | 2.55e+0(7.48e-1)+ | 4.09e-1(1.34e-1)− | 6.82e-1(1.43e-1)− | 1.77e+0(8.24e-1)+ | 1.52e+0(6.54e-1)+ | 1.84e+0(4.86e-1)+ | 8.28e-1(1.52e-1) |
|  | 10 | 3.49e+0(1.01e+0)+ | 4.18e-1(1.81e-1)− | 8.19e-1(1.39e-1)− | 2.49e+0(9.71e-1)+ | 2.19e+0(1.13e+0)+ | 2.67e+0(8.17e-1)+ | 1.40e+0(2.64e-1) |
| WFG3 | 3 | 5.65e-1(4.14e-2)+ | 5.26e-1(5.99e-2)+ | 3.05e-1(6.02e-2)+ | 4.87e-1(6.70e-2)+ | 4.42e-1(6.58e-2)+ | 3.67e-1(4.79e-2)+ | 2.65e-1(5.63e-2) |
|  | 4 | 7.12e-1(6.70e-2)+ | 6.35e-1(6.90e-2)+ | 5.33e-1(6.42e-2)+ | 5.75e-1(7.97e-2)+ | 5.24e-1(7.33e-2)+ | 5.47e-1(6.00e-2)+ | 3.88e-1(6.09e-2) |
|  | 6 | 7.42e-1(9.98e-2)+ | 6.24e-1(1.35e-1)+ | 7.25e-1(7.13e-2)+ | 6.91e-1(8.44e-2)+ | 5.60e-1(9.53e-2)+ | 7.62e-1(6.68e-2)+ | 6.04e-1(8.95e-2) |
|  | 8 | 7.74e-1(1.66e-1)≈ | 7.26e-1(1.06e-1)+ | 8.46e-1(1.67e-2)+ | 6.83e-1(1.06e-1)− | 5.18e-1(1.13e-1)− | 8.26e-1(1.01e-1)+ | 7.58e-1(9.00e-2) |
|  | 10 | 5.78e-1(9.80e-2)− | 5.54e-1(8.05e-2)− | 7.80e-1(8.72e-2)+ | 4.91e-1(8.69e-2)− | 4.07e-1(9.40e-2)− | 6.44e-1(1.04e-1)≈ | 6.92e-1(1.07e-1) |
| WFG4 | 3 | 4.74e-1(4.21e-2)+ | 3.78e-1(2.17e-2)+ | 3.42e-1(2.35e-2)+ | 3.49e-1(3.80e-2)+ | 3.04e-1(2.99e-2)+ | 3.66e-1(6.70e-2)+ | 2.55e-1(3.20e-2) |
|  | 4 | 8.04e-1(5.34e-2)+ | 5.86e-1(3.17e-2)+ | 6.00e-1(6.42e-2)+ | 7.81e-1(1.78e-1)+ | 6.15e-1(1.13e-1)+ | 9.50e-1(1.50e-1)+ | 4.85e-1(6.14e-2) |
|  | 6 | 1.83e+0(3.74e-1)+ | 1.20e+0(1.52e-1)≈ | 1.12e+0(1.55e-1)+ | 2.78e+0(4.35e-1)+ | 2.26e+0(4.42e-1)+ | 2.56e+0(4.05e-1)+ | 1.21e+0(2.18e-1) |
|  | 8 | 3.39e+0(1.48e+0)≈ | 2.33e+0(5.25e-1)+ | 2.15e+0(3.46e-1)− | 5.15e+0(5.66e-1)+ | 4.22e+0(5.32e-1)+ | 5.19e+0(4.73e-1)+ | 2.55e+0(5.66e-1) |
|  | 10 | 3.27e+0(2.29e+0)− | 4.00e+0(9.92e-1)+ | 3.45e+0(3.75e-1)+ | 7.46e+0(8.64e-1)+ | 6.61e+0(8.48e-1)+ | 7.03e+0(6.17e-1)+ | 3.92e+0(7.04e-1) |
| WFG5 | 3 | 2.07e-1(1.28e-2)− | 3.01e-1(3.82e-2)≈ | 2.38e-1(7.04e-2)≈ | 3.98e-1(3.16e-2)+ | 3.93e-1(5.70e-2)+ | 3.60e-1(7.41e-2)+ | 3.49e-1(1.55e-1) |
|  | 4 | 7.09e-1(1.49e-1)− | 5.32e-1(4.45e-2)− | 4.97e-1(4.53e-2)− | 6.09e-1(6.70e-2)− | 6.13e-1(5.55e-2)− | 9.11e-1(6.00e-2)+ | 8.68e-1(7.81e-2) |
|  | 6 | 2.38e+0(2.47e-1)+ | 1.07e+0(1.36e-1)− | 1.38e+0(1.64e-1)− | 1.89e+0(2.56e-1)+ | 1.52e+0(2.17e-1)− | 2.13e+0(1.77e-1)+ | 1.71e+0(1.09e-1) |
|  | 8 | 4.63e+0(2.89e-1)+ | 2.11e+0(5.15e-1)− | 2.74e+0(4.81e-1)+ | 4.13e+0(4.55e-1)+ | 3.26e+0(4.42e-1)+ | 4.08e+0(2.55e-1)+ | 2.88e+0(2.00e-1) |
|  | 10 | 6.67e+0(3.78e-1)+ | 2.48e+0(9.46e-1)− | 3.13e+0(5.04e-1)− | 5.90e+0(5.30e-1)+ | 5.16e+0(5.38e-1)+ | 5.84e+0(5.37e-1)+ | 3.87e+0(3.50e-1) |
| WFG6 | 3 | 5.52e-1(4.95e-2)+ | 6.19e-1(6.81e-2)+ | 5.70e-1(8.76e-2)+ | 5.71e-1(5.32e-2)+ | 5.65e-1(5.43e-2)+ | 5.09e-1(5.01e-2)≈ | 5.21e-1(1.15e-1) |
|  | 4 | 8.09e-1(7.65e-2)≈ | 7.62e-1(9.60e-2)≈ | 8.14e-1(6.51e-2)≈ | 8.33e-1(7.44e-2)+ | 7.87e-1(7.30e-2)≈ | 1.07e+0(7.09e-2)+ | 8.09e-1(1.12e-1) |
|  | 6 | 2.25e+0(5.29e-1)+ | 1.28e+0(1.52e-1)− | 1.52e+0(9.93e-2)≈ | 2.17e+0(3.22e-1)+ | 1.74e+0(2.70e-1)+ | 2.52e+0(2.20e-1)+ | 1.60e+0(1.59e-1) |
|  | 8 | 3.63e+0(9.69e-1)+ | 1.50e+0(2.46e-1)− | 2.66e+0(3.17e-1)≈ | 3.96e+0(7.85e-1)+ | 3.41e+0(4.65e-1)+ | 4.60e+0(3.93e-1)+ | 2.72e+0(2.95e-1) |
|  | 10 | 6.42e+0(8.39e-1)+ | 1.27e+0(1.06e-1)− | 3.67e+0(3.06e-1)+ | 5.61e+0(7.06e-1)+ | 4.68e+0(6.46e-1)+ | 6.05e+0(7.21e-1)+ | 3.38e+0(4.60e-1) |
| WFG7 | 3 | 5.47e-1(3.21e-2)+ | 5.38e-1(3.52e-2)+ | 4.97e-1(3.13e-2)+ | 4.36e-1(3.98e-2)+ | 3.94e-1(4.46e-2)+ | 3.65e-1(5.17e-2)+ | 2.92e-1(2.42e-2) |
|  | 4 | 9.25e-1(9.05e-2)+ | 7.42e-1(3.50e-2)+ | 7.47e-1(3.15e-2)+ | 7.74e-1(1.39e-1)+ | 6.29e-1(5.40e-2)+ | 8.46e-1(1.05e-1)+ | 5.38e-1(5.32e-2) |
|  | 6 | 2.85e+0(3.54e-1)+ | 1.41e+0(1.08e-1)− | 1.41e+0(1.36e-1)− | 2.29e+0(4.59e-1)+ | 1.74e+0(2.09e-1)+ | 2.45e+0(2.22e-1)+ | 1.61e+0(1.56e-1) |
|  | 8 | 5.37e+0(4.28e-1)+ | 2.59e+0(2.47e-1)− | 2.40e+0(3.16e-1)− | 4.51e+0(6.31e-1)+ | 3.62e+0(5.07e-1)+ | 4.68e+0(3.37e-1)+ | 3.28e+0(2.02e-1) |
|  | 10 | 7.77e+0(5.41e-1)+ | 3.50e+0(4.76e-1)− | 3.47e+0(3.98e-1)− | 6.92e+0(5.79e-1)+ | 5.72e+0(6.38e-1)+ | 6.70e+0(4.31e-1)+ | 4.85e+0(3.42e-1) |
| WFG8 | 3 | 7.23e-1(3.76e-2)+ | 5.89e-1(2.95e-2)+ | 4.72e-1(4.57e-2)+ | 6.59e-1(5.09e-2)+ | 6.21e-1(4.47e-2)+ | 6.77e-1(4.74e-2)+ | 5.79e-1(4.03e-2) |
|  | 4 | 1.19e+0(6.76e-2)+ | 1.01e+0(5.20e-2)− | 9.25e-1(5.15e-2)− | 1.14e+0(8.61e-2)+ | 1.07e+0(7.07e-2)+ | 1.30e+0(7.86e-2)+ | 1.07e+0(7.91e-2) |
|  | 6 | 2.80e+0(3.88e-1)+ | 1.82e+0(1.29e-1)− | 1.96e+0(1.02e-1)− | 2.77e+0(1.80e-1)+ | 2.58e+0(2.23e-1)+ | 2.90e+0(2.21e-1)+ | 2.22e+0(1.47e-1) |
|  | 8 | 5.23e+0(4.86e-1)+ | 2.93e+0(4.96e-1)− | 3.31e+0(2.44e-1)− | 5.13e+0(3.86e-1)+ | 4.69e+0(4.63e-1)+ | 4.98e+0(3.05e-1)+ | 3.78e+0(3.27e-1) |
|  | 10 | 7.43e+0(5.62e-1)+ | 2.74e+0(1.25e+0)− | 4.75e+0(5.95e-1)− | 7.03e+0(5.46e-1)+ | 6.52e+0(3.98e-1)+ | 6.74e+0(5.72e-1)+ | 5.03e+0(9.32e-1) |
| WFG9 | 3 | 5.82e-1(7.28e-2)+ | 5.83e-1(7.77e-2)+ | 5.56e-1(9.06e-2)+ | 6.10e-1(1.00e-1)+ | 5.32e-1(1.12e-1)+ | 4.51e-1(8.67e-2)+ | 3.82e-1(8.04e-2) |
|  | 4 | 1.00e+0(1.88e-1)+ | 8.56e-1(1.30e-1)+ | 8.76e-1(1.43e-1)+ | 1.00e+0(1.56e-1)+ | 8.59e-1(2.01e-1)+ | 8.50e-1(1.15e-1)+ | 6.77e-1(9.61e-2) |
|  | 6 | 2.72e+0(3.83e-1)+ | 1.72e+0(2.90e-1)+ | 1.66e+0(2.48e-1)+ | 2.44e+0(3.25e-1)+ | 1.87e+0(2.59e-1)+ | 2.17e+0(1.80e-1)+ | 1.45e+0(1.42e-1) |
|  | 8 | 5.14e+0(5.22e-1)+ | 3.05e+0(4.65e-1)+ | 2.82e+0(2.91e-1)≈ | 4.80e+0(4.05e-1)+ | 3.95e+0(4.95e-1)+ | 4.17e+0(3.83e-1)+ | 2.76e+0(3.72e-1) |
|  | 10 | 7.30e+0(5.37e-1)+ | 4.30e+0(8.61e-1)≈ | 3.81e+0(4.78e-1)≈ | 6.66e+0(5.44e-1)+ | 5.47e+0(6.11e-1)+ | 5.75e+0(4.84e-1)+ | 3.98e+0(4.51e-1) |
| +/≈/− |  | 37/4/4 | 16/10/19 | 18/11/16 | 41/1/3 | 38/3/4 | 42/3/0 |  |

Table 10: Statistical results of the HV value obtained by comparison algorithms on 21 DTLZ optimization problems over 30 runs. Symbols '+', '≈', '−' denote LORA-MaOO is statistically significantly superior to, almost equivalent to, and inferior to the compared algorithms in the Wilcoxon rank sum test (significance level is 0.05), respectively. The last row counts the total win/tie/loss results.

| Problems | M | ParEGO | KRVEA | KTA2 | CSEA | REMO | OREA | LORA-MaOO |
|---|---|---|---|---|---|---|---|---|
| DTLZ1 | 3 | 0.00e+0(0.00e+0)≈ | 0.00e+0(0.00e+0)≈ | 0.00e+0(0.00e+0)≈ | 0.00e+0(0.00e+0)≈ | 0.00e+0(0.00e+0)≈ | 0.00e+0(0.00e+0)≈ | 0.00e+0(0.00e+0) |
|  | 4 | 0.00e+0(0.00e+0)≈ | 0.00e+0(0.00e+0)≈ | 0.00e+0(0.00e+0)≈ | 0.00e+0(0.00e+0)≈ | 0.00e+0(0.00e+0)≈ | 0.00e+0(0.00e+0)≈ | 0.00e+0(0.00e+0) |
|  | 6 | 0.00e+0(0.00e+0)≈ | 0.00e+0(0.00e+0)≈ | 0.00e+0(0.00e+0)≈ | 0.00e+0(0.00e+0)≈ | 0.00e+0(0.00e+0)≈ | 0.00e+0(0.00e+0)≈ | 0.00e+0(0.00e+0) |
| DTLZ2 | 3 | 4.53e-2(2.22e-2)+ | 2.61e-1(4.46e-2)− | 3.87e-1(6.59e-3)− | 1.55e-1(3.85e-2)+ | 2.49e-1(3.32e-2)+ | 3.49e-1(1.33e-2)+ | 3.77e-1(6.75e-3) |
|  | 4 | 6.06e-2(2.65e-2)+ | 3.71e-1(6.43e-2)+ | 4.80e-1(1.34e-2)+ | 1.95e-1(3.26e-2)+ | 3.09e-1(4.54e-2)+ | 3.87e-1(3.31e-2)+ | 4.75e-1(2.34e-2) |
|  | 6 | 1.26e-1(1.87e-2)+ | 4.85e-1(4.22e-2)+ | 4.48e-1(7.23e-2)+ | 2.86e-1(4.80e-2)+ | 4.00e-1(4.15e-2)+ | 3.66e-1(3.09e-2)+ | 6.09e-1(2.27e-2) |
| DTLZ3 | 3 | 0.00e+0(0.00e+0)≈ | 0.00e+0(0.00e+0)≈ | 0.00e+0(0.00e+0)≈ | 0.00e+0(0.00e+0)≈ | 0.00e+0(0.00e+0)≈ | 0.00e+0(0.00e+0)≈ | 0.00e+0(0.00e+0) |
|  | 4 | 0.00e+0(0.00e+0)≈ | 0.00e+0(0.00e+0)≈ | 0.00e+0(0.00e+0)≈ | 0.00e+0(0.00e+0)≈ | 0.00e+0(0.00e+0)≈ | 0.00e+0(0.00e+0)≈ | 0.00e+0(0.00e+0) |
|  | 6 | 0.00e+0(0.00e+0)≈ | 0.00e+0(0.00e+0)≈ | 0.00e+0(0.00e+0)≈ | 0.00e+0(0.00e+0)≈ | 0.00e+0(0.00e+0)≈ | 0.00e+0(0.00e+0)≈ | 0.00e+0(0.00e+0) |
| DTLZ4 | 3 | 4.20e-4(2.03e-3)+ | 6.42e-2(5.54e-2)+ | 8.85e-2(7.53e-2)+ | 6.53e-2(3.42e-2)+ | 1.99e-1(6.05e-2)+ | 2.52e-1(6.75e-2)+ | 3.24e-1(9.98e-2) |
|  | 4 | 3.27e-3(6.73e-3)+ | 8.79e-2(6.62e-2)+ | 8.14e-2(5.85e-2)+ | 1.46e-1(5.25e-2)+ | 2.52e-1(6.25e-2)+ | 3.66e-1(8.97e-2)+ | 3.93e-1(9.18e-2) |
|  | 6 | 2.14e-2(2.69e-2)+ | 2.05e-1(9.66e-2)+ | 1.44e-1(8.78e-2)+ | 3.16e-1(6.50e-2)+ | 3.53e-1(7.16e-2)+ | 5.12e-1(5.37e-2)≈ | 5.17e-1(4.93e-2) |
| DTLZ5 | 3 | 7.49e-3(1.04e-2)+ | 2.60e-2(1.04e-2)+ | 8.60e-2(1.99e-2)≈ | 2.54e-2(9.46e-3)+ | 4.66e-2(1.02e-2)+ | 8.48e-2(1.78e-3)+ | 8.53e-2(2.03e-3) |
|  | 4 | 4.12e-3(5.91e-3)+ | 2.35e-2(7.10e-3)+ | 3.31e-2(4.30e-3)+ | 1.10e-2(4.90e-3)+ | 1.65e-2(7.08e-3)+ | 3.55e-2(4.96e-3)+ | 3.73e-2(3.97e-3) |
|  | 6 | 1.75e-2(1.88e-3)≈ | 1.28e-2(2.87e-3)− | 8.26e-3(2.88e-3)− | 5.75e-3(3.24e-3)+ | 8.48e-3(3.87e-3)+ | 9.99e-3(3.78e-3)+ | 9.23e-3(3.73e-3) |
| DTLZ6 | 3 | 3.91e-3(7.22e-3)+ | 0.00e+0(0.00e+0)+ | 0.00e+0(0.00e+0)+ | 0.00e+0(0.00e+0)+ | 0.00e+0(0.00e+0)+ | 3.52e-2(2.51e-2)+ | 4.91e-2(2.38e-2) |
|  | 4 | 1.78e-3(2.86e-3)+ | 0.00e+0(0.00e+0)+ | 2.07e-5(1.11e-4)+ | 0.00e+0(0.00e+0)+ | 0.00e+0(0.00e+0)+ | 2.60e-4(9.64e-4)+ | 7.45e-3(9.93e-3) |
|  | 6 | 1.28e-3(2.18e-3)+ | 0.00e+0(0.00e+0)+ | 1.10e-5(5.88e-5)+ | 0.00e+0(0.00e+0)+ | 0.00e+0(0.00e+0)+ | 1.21e-6(6.50e-6)+ | 7.42e-4(2.53e-3) |
| DTLZ7 | 3 | 1.81e-1(4.40e-2)+ | 2.53e-1(9.02e-3)≈ | 2.81e-1(3.28e-2)− | 1.44e-2(2.31e-2)+ | 2.11e-2(2.95e-2)+ | 2.23e-1(3.95e-2)+ | 2.47e-1(3.63e-2) |
|  | 4 | 9.45e-2(3.19e-2)+ | 1.95e-1(1.73e-2)≈ | 2.36e-1(8.48e-3)− | 4.80e-4(2.46e-3)+ | 1.20e-2(2.15e-2)+ | 1.04e-1(1.79e-2)+ | 1.88e-1(3.33e-2) |
|  | 6 | 3.12e-2(1.83e-2)≈ | 1.02e-1(1.04e-2)+ | 1.57e-1(1.62e-2)− | 5.56e-4(2.99e-3)+ | 1.55e-2(1.81e-2)+ | 8.81e-4(1.91e-3)+ | 1.05e-1(2.61e-2) |
| +/≈/− |  | 14/7/0 | 11/9/1 | 8/9/4 | 15/6/0 | 14/7/0 | 10/11/0 |  |

Table 11: Statistical results of the HV value obtained by comparison algorithms on 27 WFG optimization problems over 30 runs. Symbols '+', '≈', '−' denote LORA-MaOO is statistically significantly superior to, almost equivalent to, and inferior to the compared algorithms in the Wilcoxon rank sum test (significance level is 0.05), respectively. The last row counts the total win/tie/loss results.

| Problems | M | ParEGO | KRVEA | KTA2 | CSEA | REMO | OREA | LORA-MaOO |
|---|---|---|---|---|---|---|---|---|
| WFG1 | 3 | 1.92e-1(2.65e-2)− | 1.09e-1(3.15e-2)≈ | 6.25e-2(3.98e-2)+ | 8.61e-2(4.91e-2)≈ | 1.02e-1(4.70e-2)≈ | 1.57e-2(2.69e-2)+ | 1.07e-1(3.15e-2) |
| | 4 | 2.07e-1(2.96e-2)− | 1.14e-1(5.44e-2)+ | 7.27e-2(5.18e-2)+ | 1.17e-1(5.34e-2)+ | 1.66e-1(3.54e-2)≈ | 2.84e-2(3.66e-2)+ | 1.70e-1(4.15e-2) |
| | 6 | 2.16e-1(8.50e-3)≈ | 1.46e-1(2.93e-2)+ | 1.11e-1(4.99e-2)+ | 1.23e-1(5.25e-2)+ | 1.76e-1(2.54e-2)+ | 1.12e-1(5.80e-2)+ | 2.11e-1(2.75e-2) |
| WFG2 | 3 | 5.76e-1(3.88e-2)+ | 7.46e-1(2.87e-2)≈ | 7.11e-1(3.38e-2)+ | 6.57e-1(2.85e-2)+ | 6.65e-1(4.44e-2)+ | 6.92e-1(2.96e-2)+ | 7.42e-1(3.11e-2) |
| | 4 | 6.14e-1(3.28e-2)+ | 8.20e-1(3.33e-2)− | 7.36e-1(3.33e-2)+ | 7.23e-1(4.35e-2)+ | 7.06e-1(4.68e-2)+ | 7.21e-1(3.81e-2)+ | 7.79e-1(3.30e-2) |
| | 6 | 6.46e-1(5.10e-2)+ | 8.51e-1(3.38e-2)≈ | 8.26e-1(3.84e-2)≈ | 7.80e-1(5.00e-2)+ | 7.73e-1(5.46e-2)+ | 7.29e-1(4.17e-2)+ | 8.39e-1(3.76e-2) |
| WFG3 | 3 | 1.04e-1(1.96e-2)+ | 1.13e-1(1.80e-2)+ | 1.90e-1(2.71e-2)≈ | 1.20e-1(1.90e-2)+ | 1.27e-1(2.01e-2)+ | 1.62e-1(2.11e-2)+ | 1.91e-1(2.20e-2) |
| | 4 | 3.10e-2(2.15e-2)+ | 3.48e-2(1.41e-2)+ | 2.73e-2(1.70e-2)+ | 3.65e-2(2.01e-2)+ | 4.07e-2(1.92e-2)+ | 3.10e-2(2.15e-2)+ | 5.57e-2(1.56e-2) |
| | 6 | 1.10e-2(1.26e-2)− | 1.39e-3(2.87e-3)− | 0.00e+0(0.00e+0)≈ | 6.59e-5(2.13e-4)≈ | 2.96e-3(8.32e-3)− | 0.00e+0(0.00e+0)≈ | 0.00e+0(0.00e+0) |
| WFG4 | 3 | 1.74e-1(1.18e-2)+ | 2.18e-1(1.10e-2)+ | 2.44e-1(1.30e-2)+ | 2.37e-1(1.46e-2)+ | 2.55e-1(1.52e-2)+ | 2.66e-1(2.01e-2)+ | 2.98e-1(1.58e-2) |
| | 4 | 2.12e-1(9.87e-3)+ | 2.97e-1(1.52e-2)+ | 3.18e-1(2.01e-2)+ | 2.96e-1(2.19e-2)+ | 3.33e-1(2.24e-2)+ | 2.97e-1(1.89e-2)+ | 3.91e-1(1.96e-2) |
| | 6 | 2.50e-1(1.18e-2)+ | 4.09e-1(3.09e-2)+ | 4.38e-1(2.23e-2)+ | 4.78e-1(2.50e-2)+ | 3.78e-1(2.82e-2)+ | 3.19e-1(2.08e-2)+ | 4.78e-1(2.39e-2) |
| WFG5 | 3 | 2.98e-1(1.33e-2)− | 2.55e-1(2.28e-2)≈ | 2.98e-1(4.75e-2)− | 2.03e-1(1.32e-2)+ | 2.08e-1(2.74e-2)+ | 2.45e-1(3.49e-2)+ | 2.51e-1(6.54e-2) |
| | 4 | 3.19e-1(2.64e-2)− | 3.21e-1(2.50e-2)− | 3.63e-1(3.37e-2)− | 2.83e-1(2.41e-2)− | 2.83e-1(2.44e-2)− | 2.16e-1(1.31e-2)− | 2.05e-1(3.01e-2) |
| | 6 | 3.39e-1(2.37e-2)− | 4.17e-1(3.07e-2)− | 3.72e-1(3.17e-2)− | 3.46e-1(2.51e-2)− | 3.53e-1(2.43e-2)− | 2.78e-1(1.48e-2)− | 2.66e-1(2.60e-2) |
| WFG6 | 3 | 1.15e-1(2.24e-2)+ | 1.20e-1(2.10e-2)+ | 1.59e-1(3.72e-2)+ | 1.29e-1(2.01e-2)+ | 1.31e-1(1.90e-2)+ | 1.87e-1(1.98e-2)≈ | 1.85e-1(4.25e-2) |
| | 4 | 1.83e-1(1.87e-2)+ | 2.18e-1(3.46e-2)≈ | 2.17e-1(2.49e-2)+ | 1.87e-1(2.16e-2)+ | 2.05e-1(2.17e-2)+ | 1.96e-1(1.60e-2)+ | 2.33e-1(5.01e-2) |
| | 6 | 2.30e-1(2.14e-2)+ | 2.75e-1(4.76e-2)+ | 3.15e-1(2.12e-2)≈ | 2.49e-1(1.89e-2)+ | 2.93e-1(3.03e-2)+ | 2.42e-1(1.28e-2)+ | 3.11e-1(2.91e-2) |
| WFG7 | 3 | 1.43e-1(8.60e-3)+ | 1.44e-1(1.11e-2)+ | 1.75e-1(1.26e-2)+ | 1.91e-1(1.74e-2)+ | 2.13e-1(2.05e-2)+ | 2.53e-1(1.32e-2)+ | 2.87e-1(1.30e-2) |
| | 4 | 1.91e-1(1.45e-2)+ | 2.22e-1(1.23e-2)+ | 2.36e-1(1.09e-2)+ | 2.42e-1(1.97e-2)+ | 2.90e-1(2.08e-2)+ | 2.83e-1(1.74e-2)+ | 3.66e-1(2.21e-2) |
| | 6 | 2.25e-1(1.42e-2)+ | 3.24e-1(2.49e-2)+ | 3.38e-1(2.89e-2)+ | 3.16e-1(3.37e-2)+ | 3.77e-1(2.50e-2)+ | 3.07e-1(1.80e-2)+ | 4.06e-1(2.28e-2) |
| WFG8 | 3 | 9.39e-2(1.01e-2)+ | 1.48e-1(9.46e-3)+ | 2.14e-1(1.61e-2)− | 1.24e-1(1.35e-2)+ | 1.32e-1(1.24e-2)+ | 1.60e-1(1.44e-2)+ | 1.84e-1(9.51e-3) |
| | 4 | 1.32e-1(1.22e-2)+ | 2.03e-1(1.81e-2)≈ | 2.17e-1(1.76e-2)− | 1.57e-1(1.81e-2)+ | 1.79e-1(1.75e-2)+ | 1.80e-1(1.38e-2)+ | 1.95e-1(2.50e-2) |
| | 6 | 1.81e-1(1.26e-2)+ | 2.59e-1(2.37e-2)− | 2.58e-1(1.13e-2)− | 2.18e-1(2.14e-2)+ | 2.62e-1(2.31e-2)− | 2.17e-1(1.19e-2)+ | 2.40e-1(2.32e-2) |
| WFG9 | 3 | 1.22e-1(1.94e-2)+ | 1.28e-1(2.33e-2)+ | 1.50e-1(3.21e-2)+ | 1.39e-1(2.58e-2)+ | 1.67e-1(3.64e-2)+ | 2.23e-1(2.82e-2)+ | 2.46e-1(3.68e-2) |
| | 4 | 1.74e-1(3.27e-2)+ | 2.08e-1(3.51e-2)+ | 2.04e-1(2.90e-2)+ | 1.87e-1(3.11e-2)+ | 2.35e-1(4.04e-2)+ | 2.63e-1(2.48e-2)+ | 3.06e-1(4.82e-2) |
| | 6 | 2.14e-1(2.85e-2)+ | 3.31e-1(5.50e-2)+ | 3.65e-1(5.25e-2)≈ | 2.76e-1(3.85e-2)+ | 3.62e-1(3.76e-2)+ | 2.90e-1(2.96e-2)+ | 3.89e-1(3.60e-2) |
| +/≈/− | | 20/1/6 | 16/6/5 | 15/6/6 | 23/2/2 | 20/3/4 | 23/2/2 | |

Table 12: Statistical results of the IGD value obtained by comparison algorithms on $5D$, $10D$, and $20D$ DTLZ optimization problems over 30 runs. Symbols '+', '≈', '−' denote LORA-MaOO is statistically significantly superior to, almost equivalent to, and inferior to the compared algorithms in the Wilcoxon rank sum test (significance level is 0.05), respectively. The last row counts the total win/tie/loss results.

| Problems | D | ParEGO | KRVEA | KTA2 | CSEA | REMO | OREA | LORA-MaOO |
|---|---|---|---|---|---|---|---|---|
| DTLZ1 | 5 | 1.24e+1(4.40e+0)+ | 7.19e+0(3.77e+0)+ | 4.00e+0(2.28e+0)≈ | 5.71e+0(2.66e+0)≈ | 5.97e+0(2.98e+0)≈ | 2.27e+0(1.45e+0)− | 4.78e+0(2.80e+0) |
| | 10 | 5.98e+1(3.81e+0)+ | 8.88e+1(2.16e+1)+ | 4.75e+1(1.55e+1)+ | 6.30e+1(1.69e+1)+ | 5.06e+1(1.49e+1)+ | 4.44e+1(1.38e+1)≈ | 4.35e+1(1.80e+1) |
| | 20 | 1.59e+2(1.56e+1)− | 3.12e+2(3.79e+1)+ | 2.48e+2(3.66e+1)− | 2.35e+2(3.47e+1)+ | 2.01e+2(3.95e+1)− | 2.94e+2(3.78e+1)+ | 2.91e+2(3.98e+1) |
| DTLZ2 | 5 | 1.81e-1(1.26e-2)+ | 6.06e-2(2.40e-3)+ | 4.39e-2(1.11e-3)+ | 1.03e-1(7.78e-3)+ | 7.94e-2(7.71e-3)+ | 6.55e-2(6.87e-3)+ | 4.36e-2(2.15e-3) |
| | 10 | 3.38e-1(2.84e-2)+ | 1.32e-1(2.77e-2)+ | 6.17e-2(3.13e-3)≈ | 2.26e-1(2.01e-2)+ | 1.65e-1(2.18e-2)+ | 8.59e-2(8.51e-3)+ | 6.19e-2(3.48e-3) |
| | 20 | 7.15e-1(1.21e-1)+ | 6.66e-1(7.34e-2)+ | 2.85e-1(5.83e-2)+ | 5.17e-1(6.66e-2)+ | 4.00e-1(7.02e-2)+ | 1.62e-1(1.35e-2)+ | 1.02e-1(1.36e-2) |
| DTLZ3 | 5 | 3.17e+1(1.17e+1)+ | 1.91e+1(7.60e+0)+ | 1.17e+1(6.12e+0)+ | 1.58e+1(7.00e+0)+ | 1.61e+1(9.16e+0)+ | 6.78e+0(4.79e+0)− | 1.51e+1(9.40e+0) |
| | 10 | 1.66e+2(1.31e+1)+ | 2.43e+2(4.61e+1)+ | 1.52e+2(4.73e+1)+ | 1.62e+2(4.84e+1)+ | 1.49e+2(3.88e+1)+ | 1.26e+2(3.18e+1)− | 1.57e+2(3.83e+1) |
| | 20 | 4.32e+2(1.78e+1)− | 9.11e+2(8.72e+1)≈ | 7.23e+2(1.38e+2)− | 7.12e+2(1.10e+2)− | 5.86e+2(1.18e+2)− | 7.81e+2(1.20e+2)− | 8.58e+2(1.31e+2) |
| DTLZ4 | 5 | 4.33e-1(5.55e-2)≈ | 1.35e-1(6.05e-2)≈ | 1.68e-1(1.22e-1)+ | 4.33e-1(1.54e-1)+ | 1.60e-1(6.12e-2)≈ | 2.91e-1(2.44e-1)≈ | 3.96e-1(3.71e-1) |
| | 10 | 6.70e-1(7.61e-2)+ | 3.32e-1(1.11e-1)+ | 3.49e-1(1.09e-1)+ | 4.62e-1(1.36e-1)+ | 2.31e-1(1.15e-1)+ | 2.39e-1(1.65e-1)+ | 1.89e-1(2.34e-1) |
| | 20 | 1.02e+0(1.04e-1)+ | 8.32e-1(1.36e-1)+ | 7.76e-1(1.29e-1)+ | 7.11e-1(1.74e-1)+ | 5.51e-1(1.18e-1)+ | 5.27e-1(2.75e-1)+ | 4.01e-1(3.28e-1) |
| DTLZ5 | 5 | 4.16e-2(9.61e-3)+ | 2.31e-2(3.02e-3)+ | 3.57e-3(2.35e-4)− | 2.18e-2(3.22e-3)+ | 1.49e-2(3.28e-3)+ | 1.12e-2(5.73e-3)+ | 4.20e-3(6.92e-4) |
| | 10 | 2.16e-1(4.45e-2)+ | 1.19e-1(3.38e-2)+ | 1.34e-2(2.83e-3)≈ | 1.18e-1(2.56e-2)+ | 7.36e-2(2.03e-2)+ | 2.02e-2(4.77e-3)+ | 1.26e-2(2.55e-3) |
| | 20 | 6.05e-1(1.43e-1)+ | 6.16e-1(1.14e-1)+ | 2.13e-1(5.07e-2)+ | 4.84e-1(8.14e-2)+ | 3.60e-1(8.07e-2)+ | 8.11e-2(3.39e-2)+ | 4.32e-2(1.45e-2) |
| DTLZ6 | 5 | 4.57e-2(1.11e-2)+ | 4.69e-1(1.54e-1)+ | 2.68e-1(1.01e-1)+ | 7.65e-1(4.09e-1)+ | 4.08e-1(2.59e-1)+ | 2.57e-2(2.92e-2)≈ | 2.98e-2(3.53e-2) |
| | 10 | 3.15e-1(1.62e-1)+ | 3.06e+0(5.21e-1)+ | 1.83e+0(4.37e-1)+ | 4.86e+0(6.30e-1)+ | 4.27e+0(5.49e-1)+ | 3.09e-1(3.99e-1)+ | 1.18e-1(1.57e-1) |
| | 20 | 3.54e+0(1.04e+0)≈ | 1.10e+1(7.15e-1)+ | 8.72e+0(1.01e+0)+ | 1.33e+1(8.48e-1)+ | 1.23e+1(7.84e-1)+ | 7.06e+0(3.05e+0)+ | 6.81e+0(5.11e+0) |
| DTLZ7 | 5 | 1.87e-1(2.40e-2)+ | 1.07e-1(1.50e-2)+ | 6.66e-2(4.28e-2)− | 5.67e-2(2.78e-1)+ | 2.30e-1(1.07e-1)+ | 3.05e-2(2.01e-1)+ | 1.41e-1(1.50e-1) |
| | 10 | 2.45e-1(4.80e-2)+ | 1.35e-1(2.37e-2)≈ | 2.19e-1(2.40e-1)− | 1.75e+0(6.32e-1)+ | 1.27e+0(5.65e-1)+ | 2.73e-1(1.58e-1)+ | 2.01e-1(1.93e-1) |
| | 20 | 2.67e-1(4.98e-2)+ | 4.17e-1(2.04e-1)+ | 4.69e-1(2.56e-1)+ | 3.69e+0(9.09e-1)+ | 2.62e+0(7.33e-1)+ | 4.77e-1(2.53e-1)+ | 2.99e-1(2.51e-1) |
| +/≈/− | | 16/3/2 | 16/5/0 | 7/9/5 | 16/3/2 | 15/4/2 | 12/5/4 | |

## I ANALYSIS ON TIME COMPLEXITY

This section briefly analyze the time complexity of LORA-MaOO and the compared SAEAs. For the convenience of time complexity analysis, we set the following notations:

- $T_n$: the number of training samples.
- $T_N$: the number of test samples.
- $M$: the number of objectives.
- $g$: the number of generations for reproducing candidate solutions.

Table 13: Statistical results of the IGD+ value obtained by comparison algorithms on $5D$, $10D$, and $20D$ DTLZ optimization problems over 30 runs. Symbols '+', '≈', '−' denote LORA-MaOO is statistically significantly superior to, almost equivalent to, and inferior to the compared algorithms in the Wilcoxon rank sum test (significance level is 0.05), respectively. The last row counts the total win/tie/loss results.

| Problems | D | ParEGO | KRVEA | KTA2 | CSEA | REMO | OREA | LORA-MaOO |
|---|---|---|---|---|---|---|---|---|
| DTLZ1 | 5 | 1.24e+1(4.40e+0)+ | 7.19e+0(3.77e+0)+ | 4.00e+0(2.28e+0)≈ | 5.70e+0(2.67e+0)≈ | 5.97e+0(2.98e+0)≈ | 2.27e+0(1.45e+0)− | 4.78e+0(2.81e+0) |
| | 10 | 5.98e+1(3.81e+0)+ | 8.88e+1(2.16e+1)+ | 4.75e+1(1.55e+1)≈ | 6.30e+1(1.69e+1)+ | 5.06e+1(1.49e+1)+ | 4.44e+1(1.38e+1)≈ | 4.35e+1(1.80e+1) |
| | 20 | 1.59e+2(1.56e+1)− | 3.12e+2(3.79e+1)≈ | 2.48e+2(3.66e+1)− | 2.35e+2(3.47e+1)− | 2.01e+2(3.95e+1)− | 2.94e+2(3.78e+1)≈ | 2.91e+2(3.98e+1) |
| DTLZ2 | 5 | 1.01e-1(7.98e-3)+ | 2.86e-2(9.66e-4)+ | 1.94e-2(6.20e-4)− | 5.24e-2(6.84e-3)+ | 3.83e-2(4.18e-3)+ | 3.92e-2(5.96e-3)+ | 2.30e-2(2.07e-3) |
| | 10 | 2.61e-1(3.63e-2)+ | 9.22e-2(2.57e-2)+ | 3.82e-2(3.29e-3)− | 1.60e-1(2.76e-2)+ | 1.01e-1(1.75e-2)+ | 5.86e-2(8.28e-3)+ | 4.47e-2(3.35e-3) |
| | 20 | 6.51e-1(1.39e-1)+ | 6.36e-1(7.19e-2)+ | 2.61e-1(5.87e-2)+ | 4.69e-1(6.69e-2)+ | 3.56e-1(8.04e-2)+ | 1.39e-1(3.02e-2)+ | 8.36e-2(1.22e-2) |
| DTLZ3 | 5 | 3.17e+1(1.17e+1)+ | 1.91e+1(9.13e+0)≈ | 1.17e+1(6.15e+0)≈ | 1.58e+1(7.61e+0)≈ | 1.61e+1(9.16e+0)≈ | 6.77e+0(4.80e+0)− | 1.51e+1(9.41e+0) |
| | 10 | 1.66e+2(1.31e+1)+ | 2.43e+2(4.61e+1)+ | 1.52e+2(4.73e+1)+ | 1.62e+2(4.84e+1)+ | 1.49e+2(3.88e+1)+ | 1.26e+2(3.18e+1)− | 1.57e+2(3.83e+1) |
| | 20 | 4.32e+2(1.78e+1)− | 9.11e+2(8.72e+1)≈ | 7.23e+2(1.38e+2)− | 7.12e+2(1.10e+2)− | 5.86e+2(1.18e+2)− | 7.81e+2(1.20e+2)− | 8.58e+2(1.31e+2) |
| DTLZ4 | 5 | 1.88e-1(3.03e-2)+ | 7.41e-2(4.55e-2)≈ | 7.39e-2(5.63e-2)≈ | 1.80e-1(7.75e-2)+ | 6.02e-2(2.08e-2)≈ | 1.24e-1(1.32e-1)≈ | 1.96e-1(2.08e-1) |
| | 10 | 4.57e-1(7.52e-2)+ | 2.66e-1(1.02e-1)+ | 2.33e-1(8.36e-2)+ | 2.34e-1(7.76e-2)+ | 1.32e-1(6.41e-2)+ | 1.07e-1(9.68e-2)+ | 8.96e-2(1.25e-1) |
| | 20 | 6.79e-1(1.38e-1)+ | 7.74e-1(1.34e-1)+ | 6.65e-1(1.18e-1)+ | 5.50e-1(1.44e-1)+ | 4.63e-1(8.22e-2)+ | 3.16e-1(1.90e-1)+ | 2.27e-1(2.12e-1) |
| DTLZ5 | 5 | 2.37e-2(3.64e-3)+ | 1.30e-2(1.76e-3)+ | 1.65e-3(1.03e-4)− | 1.26e-2(2.08e-3)+ | 7.74e-3(1.49e-3)+ | 6.37e-3(2.67e-3)+ | 2.48e-3(5.73e-4) |
| | 10 | 1.60e-1(4.40e-2)+ | 9.18e-2(2.76e-2)+ | 8.66e-3(1.96e-3)≈ | 9.58e-2(2.60e-2)+ | 5.78e-2(1.81e-2)+ | 1.59e-2(5.12e-3)+ | 9.40e-3(1.93e-3) |
| | 20 | 5.52e-1(1.50e-1)+ | 5.91e-1(7.98e-2)+ | 2.01e-1(5.29e-2)+ | 4.67e-1(8.41e-2)+ | 3.49e-1(8.31e-2)+ | 7.69e-2(3.31e-2)+ | 3.93e-2(1.41e-2) |
| DTLZ6 | 5 | 2.47e-2(6.71e-3)+ | 3.89e-1(1.88e-1)+ | 2.13e-1(1.02e-1)+ | 7.13e-1(4.42e-1)+ | 3.64e-1(2.75e-1)+ | 9.09e-3(9.88e-3)≈ | 1.17e-2(1.30e-2) |
| | 10 | 2.42e-1(1.07e-1)+ | 3.05e+0(5.23e-1)+ | 1.82e+0(4.48e-1)+ | 4.85e+0(6.38e-1)+ | 4.27e+0(5.48e-1)+ | 2.35e-1(4.14e-1)+ | 6.74e-2(1.55e-1) |
| | 20 | 3.49e+0(1.06e+0)≈ | 1.10e+1(7.14e-1)+ | 8.71e+0(1.01e+0)≈ | 1.33e+1(8.47e-1)+ | 1.23e+1(7.85e-1)+ | 7.04e+0(3.06e+0)≈ | 6.77e+0(5.15e+0) |
| DTLZ7 | 5 | 7.68e-2(1.31e-2)+ | 4.68e-2(4.64e-3)+ | 3.52e-2(2.90e-2)≈ | 4.46e-1(2.65e-1)+ | 1.55e-1(8.32e-2)+ | 2.04e-1(1.80e-1)+ | 8.42e-2(1.14e-1) |
| | 10 | 1.10e-1(3.57e-2)+ | 7.39e-2(1.52e-2)≈ | 1.54e-1(1.97e-1)− | 1.65e+0(6.43e-1)+ | 1.20e+0(5.73e-1)+ | 1.79e-1(1.20e-1)+ | 1.38e-1(1.53e-1) |
| | 20 | 1.38e-1(4.67e-2)≈ | 3.30e-1(1.80e-1)+ | 3.60e-1(2.27e-1)+ | 3.65e+0(9.08e-1)+ | 2.61e+0(7.28e-1)+ | 4.15e-1(2.30e-1)+ | 2.28e-1(2.10e-1) |
| $+/≈/−$ | | 16/3/2 | 16/5/0 | 7/8/6 | 16/3/2 | 15/4/2 | 12/5/4 | |

Table 14: Statistical results of the HV value obtained by comparison algorithms on $5D$, $10D$, and $20D$ DTLZ optimization problems over 30 runs. Symbols '+', '≈', '−' denote LORA-MaOO is statistically significantly superior to, almost equivalent to, and inferior to the compared algorithms in the Wilcoxon rank sum test (significance level is 0.05), respectively. The last row counts the total win/tie/loss results.

| Problems | D | ParEGO | KRVEA | KTA2 | CSEA | REMO | OREA | LORA-MaOO |
|---|---|---|---|---|---|---|---|---|
| DTLZ1 | 5 | 0.00e+0(0.00e+0)≈ | 0.00e+0(0.00e+0)≈ | 0.00e+0(0.00e+0)≈ | 0.00e+0(0.00e+0)≈ | 0.00e+0(0.00e+0)≈ | 6.38e-4(3.44e-3)≈ | 1.10e-2(5.92e-2) |
| | 10 | 0.00e+0(0.00e+0)≈ | 0.00e+0(0.00e+0)≈ | 0.00e+0(0.00e+0)≈ | 0.00e+0(0.00e+0)≈ | 0.00e+0(0.00e+0)≈ | 0.00e+0(0.00e+0)≈ | 0.00e+0(0.00e+0) |
| | 20 | 0.00e+0(0.00e+0)≈ | 0.00e+0(0.00e+0)≈ | 0.00e+0(0.00e+0)≈ | 0.00e+0(0.00e+0)≈ | 0.00e+0(0.00e+0)≈ | 0.00e+0(0.00e+0)≈ | 0.00e+0(0.00e+0) |
| DTLZ2 | 5 | 2.15e-1(1.98e-2)+ | 4.00e-1(2.88e-3)+ | 4.26e-1(1.70e-3)− | 3.39e-1(1.61e-2)+ | 3.78e-1(1.08e-2)+ | 3.83e-1(1.22e-2)+ | 4.21e-1(4.35e-3) |
| | 10 | 4.53e-2(2.22e-2)+ | 2.61e-1(4.46e-2)+ | 3.87e-1(6.59e-3)− | 1.55e-1(3.85e-2)+ | 2.49e-1(3.32e-2)+ | 3.49e-1(1.33e-2)+ | 3.77e-1(6.75e-3) |
| | 20 | 1.02e-3(3.44e-3)+ | 7.41e-5(3.74e-4)+ | 8.31e-2(4.46e-2)+ | 5.91e-3(9.22e-3)+ | 3.81e-2(2.47e-2)+ | 2.38e-1(2.81e-2)+ | 3.01e-1(2.25e-2) |
| DTLZ3 | 5 | 0.00e+0(0.00e+0)≈ | 0.00e+0(0.00e+0)≈ | 0.00e+0(0.00e+0)≈ | 0.00e+0(0.00e+0)≈ | 0.00e+0(0.00e+0)≈ | 0.00e+0(0.00e+0)≈ | 0.00e+0(0.00e+0) |
| | 10 | 0.00e+0(0.00e+0)≈ | 0.00e+0(0.00e+0)≈ | 0.00e+0(0.00e+0)≈ | 0.00e+0(0.00e+0)≈ | 0.00e+0(0.00e+0)≈ | 0.00e+0(0.00e+0)≈ | 0.00e+0(0.00e+0) |
| | 20 | 0.00e+0(0.00e+0)≈ | 0.00e+0(0.00e+0)≈ | 0.00e+0(0.00e+0)≈ | 0.00e+0(0.00e+0)≈ | 0.00e+0(0.00e+0)≈ | 0.00e+0(0.00e+0)≈ | 0.00e+0(0.00e+0) |
| DTLZ4 | 5 | 2.28e-2(2.65e-2)+ | 2.93e-1(7.80e-2)+ | 3.02e-1(8.32e-2)+ | 1.87e-1(5.36e-2)+ | 3.07e-1(5.76e-2)+ | 2.65e-1(1.11e-1)≈ | 2.49e-1(1.66e-1) |
| | 10 | 4.20e-4(2.03e-3)+ | 6.42e-2(5.54e-2)+ | 8.85e-2(7.53e-2)+ | 6.53e-2(3.42e-2)+ | 1.99e-1(6.05e-2)+ | 2.52e-1(6.75e-2)+ | 3.24e-1(9.98e-2) |
| | 20 | 0.00e+0(0.00e+0)+ | 0.00e+0(0.00e+0)+ | 8.09e-4(2.67e-3)+ | 1.20e-3(5.76e-3)+ | 6.38e-3(8.46e-3)+ | 8.86e-2(6.97e-2)+ | 1.97e-1(1.08e-1) |
| DTLZ5 | 5 | 7.09e-2(2.85e-3)+ | 7.93e-2(2.59e-3)+ | 9.36e-2(1.60e-4)− | 8.00e-2(2.29e-3)+ | 8.58e-2(2.49e-3)+ | 9.14e-2(6.46e-4)+ | 9.27e-2(5.11e-4) |
| | 10 | 7.49e-3(1.04e-2)+ | 2.60e-2(1.04e-2)+ | 8.60e-2(1.99e-3)+ | 2.54e-2(9.46e-3)+ | 4.66e-2(1.02e-2)+ | 8.48e-2(1.78e-3)+ | 8.53e-2(2.03e-3) |
| | 20 | 4.12e-5(2.22e-4)+ | 0.00e+0(0.00e+0)+ | 1.00e-2(1.02e-2)+ | 0.00e+0(0.00e+0)+ | 9.09e-4(2.11e-3)+ | 5.09e-2(7.32e-3)+ | 6.15e-2(7.35e-3) |
| DTLZ6 | 5 | 6.52e-2(7.55e-3)+ | 6.06e-3(1.28e-2)+ | 3.10e-2(1.98e-2)+ | 3.56e-3(1.03e-2)+ | 1.93e-2(2.10e-2)+ | 8.70e-2(8.64e-3)− | 7.68e-2(1.94e-2) |
| | 10 | 3.91e-3(7.22e-3)+ | 0.00e+0(0.00e+0)+ | 0.00e+0(0.00e+0)+ | 0.00e+0(0.00e+0)+ | 0.00e+0(0.00e+0)+ | 3.52e-2(2.51e-2)+ | 4.91e-2(2.38e-2) |
| | 20 | 0.00e+0(0.00e+0)≈ | 0.00e+0(0.00e+0)≈ | 0.00e+0(0.00e+0)≈ | 0.00e+0(0.00e+0)≈ | 0.00e+0(0.00e+0)≈ | 0.00e+0(0.00e+0)≈ | 2.06e-3(7.33e-3) |
| DTLZ7 | 5 | 2.29e-1(2.23e-2)+ | 2.82e-1(5.98e-3)+ | 3.08e-1(7.28e-3)− | 1.90e-1(3.80e-2)+ | 2.24e-1(2.41e-2)+ | 2.49e-1(4.23e-2)+ | 2.84e-1(3.96e-2) |
| | 10 | 1.81e-1(4.40e-2)+ | 2.53e-1(9.02e-3)≈ | 2.81e-1(3.28e-2)− | 1.44e-2(2.31e-2)+ | 2.11e-2(2.95e-2)+ | 2.23e-1(3.95e-2)+ | 2.47e-1(3.63e-2) |
| | 20 | 1.59e-1(4.85e-2)+ | 1.56e-1(4.53e-2)+ | 2.21e-1(3.02e-2)≈ | 0.00e+0(0.00e+0)+ | 1.56e-6(8.40e-6)+ | 1.15e-1(4.03e-2)+ | 2.03e-1(4.17e-2) |
| $+/≈/−$ | | 14/7/0 | 12/9/0 | 6/10/5 | 14/7/0 | 13/8/0 | 11/9/1 | |



- $p$: the population size for a generation.

The model used in LORA-MaOO is Gaussian Process, the training time complexity is analyzed as follows:

- Time complexity of covariance matrix computation is $O(T_n^2)$
- Time complexity of Cholesky decomposition and computation of likelihood: $O(T_n^3)$

The prediction time complexity is analyzed as follows:

- Time complexity of computing the covariance between test sample and training samples: $O(T_n * T_N)$
- Time complexity of predicting the mean: $O(T_n * T_N)$
- Time complexity of predicting the variance: $O(T_n^2 * T_N)$



In summary, the overall training complexity is $O(T_n^3)$, and the overall prediction complexity is $O(T_n^2 * T_N)$.

Now we analyze the time complexity of model-based optimization algorithms, for each iteration, the number of test samples is $p * g$, so the total number of test samples is approximately $T_N = T_n * p * g$.

For LORA-MaOO with a $M$-objective optimization problem:

- Time complexity of calculating ordinal values is $O(T_n * T_{nd})$, where $T_{nd}$ is the number of non-dominated solutions in the archive. When calculating artificial ordinal relations, an additional time complexity for KNN clustering is $O(T_n)$. As $T_n > T_{nd}$, we have overall time complexity $O(T_n^2)$.
- Time complexity of training an ordinal model and $M - 1$ angular models is $O(T_n^3 * M)$.
- Time complexity of prediction in the ordinal model is $O(T_n^3 * g * p)$.
- The time complexity of prediction in $M - 1$ angular models: $O(T_n^3 * p * (M - 1))$.
- The overtime time complexity in models for LORA-MaOO:
  $O(T_n^3 * (M + g * p + p * M - p) + T_n^2) \approx O(T_n^3 * (g * p + p * M)) = O(T_n^3 * p * (g + M))$.

It can be observed that the time complexity of calculating ordinal value is trivial.

In comparison, for other optimization algorithms with $M$ surrogate models:

- The time complexity of training models: $O(T_n^3 * M)$.
- The time complexity of prediction: $O(T_n^3 * g * p * M)$.
- Time overtime time complexity in models: $O(T_n^3 * M * (1 + g * p)) \approx O(T_n^3 * p * g * M)$.

For other optimization algorithms with only one surrogate model:

- Time overtime time complexity: $O(T_n^3 * (1 + g * p)) \approx O(T_n^3 * p * g)$.

Therefore, increasing the number of objectives $M$ has limited impact on the time cost of LORA-MaOO ($O(T_n^3 * p * (g + M))$), but for the comparison algorithms with $M$ surrogate models, their time cost would increase rapidly ($O(T_n^3 * p * g * M)$).

Although LORA-MOO has $M$ surrogate models in total, its time complexity does not significantly larger than the time complexity of optimization algorithms with only one surrogate model ($O(T_n^3 * p * g)$).

