# OpenReview forum: "LORA-MaOO: Learning Ordinal Relations and Angles for Expensive Many-Objective Optimization"
_ICLR.cc/2025/Conference — Submitted to ICLR 2025_

### Official Review · Reviewer_9iD6 · 2024-10-18

**Soundness:** 3
**Presentation:** 2
**Contribution:** 2
**Rating:** 6
**Confidence:** 4

**Summary:**

This paper proposes an evolutionary algorithm assisted by ordinal surrogate and angular surrogate for solving expensive many-objective optimization problems. The authors introduce two strategies, ordinal surrogate and angular surrogate, to maintain the convergence and diversity. Additionally, the authors propose a improved method to quantify dominance-based ordinal relations. Extensive experiments on benchmark and neural architecture search problems are conducted to demonstrate the effectiveness of the proposed method.

**Strengths:**

1. This paper proposes a  novel ordinal-regression-based surrogate model that is used to learn ordinal landscapes of expensive many-objective optimization problems. This is a very interesting topic.
2. Extensive experiments on benchmarks and a practical case are employed to verify the effectiveness of the proposed method.
3. This paper clearly introduces the proposed method. The paper will provide a link which claims the code availability after the paper is accepted.

**Weaknesses:**

1. The clarity of the motivation requires substantial improvement. In the abstract, the authors point out that the challenges of many-objective optimization algorithms include the cost of surrogate models and maintaining solution diversity. "Learning surrogates from spherical coordinates" is intended to overcome the aforementioned challenges. Firstly, "spherical coordinates" are not clearly explained in the abstract. Secondly, the term "M − 1 regression-based surrogates" is not sufficiently clear in context. Furthermore, the advantages of ordinal surrogate and angular surrogate compared to existing methods are not adequately explained. The simple description, "which lowers the cost of using surrogates and thus enhances the optimization efficiency," is not convincing.
2. In the motivation of the Introduction, what are the advantages of ordinal surrogate compared to existing methods? Additionally, what are the advantages of angular surrogate compared to methods introducing reference vectors? In fact, reference vectors can be generated from a probability distribution, which is not a complex method. The motivation behind the "non-parametric diversity maintenance strategy" mentioned in the text is unconvincing.
3. The statement, "This framework provides flexibility to use only a subset of surrogates during the optimization and thus reduce the cost of using surrogates," should be supported by evidence. Compared to what other methods does it reduce the cost of using surrogates?
4. In Section 3.3.1, the initial population is comprised of two components (as stated in Lines 1-9 of Algorithm 3). What is the rationale behind this design? "The remaining initial solutions are mutants of current reference points." The motivation behind this design should be clearly articulated. Furthermore, its effectiveness should be demonstrated through ablation experiments.
5. The statement, "Some HV-based MOBO methods are not compared as they failed to solve many objectives," is somewhat absolute. Furthermore, reference vector-based methods should be included as baselines. The work titled “Pareto Set Learning for Expensive Multi-Objective Optimization” should be included in the comparison.
6. The effectiveness of ordinal surrogate and angular surrogate should be further analyzed. Specifically, the accuracy of the regression model should be clearly demonstrated.
7. How can the superiority of angular surrogate in maintaining diversity, as compared to decomposition-based methods, be experimentally or theoretically demonstrated?
8. The time complexity should be briefly analyzed, especially for calculating ordinal values (Alg. 2).

Minor issues:
1. There is a missing space after "It is a challenge to reach diverse optimal solutions with a relatively low cost of using surrogates for MaOO problems."
2. In the introduction, the description of "Pareto front" is inaccurate. "A set of non-dominated solutions that represent different optimal balance between conflicting objectives." should refer to the Pareto set, not the Pareto front.
3. In the "Connection to SAEAs" section, the descriptions of "SAEA" and "MOBO" are not rigorous. The former refers to surrogate-assisted evolutionary algorithms, while the latter refers to multi-objective Bayesian optimization. Firstly, the former can be applied to both multi-objective and single-objective problems. Secondly, the main difference between the two lies in the method of solution generation. Thirdly, it is not rigorous to distinguish between them based on whether a probability model is used.
4. In Figures 1 and 2, certain content is not visible due to obscuration. The size of the images should be adjusted accordingly.

I think the work would have a good impact after substantial update. If the authors can address the aforementioned issues, I am prepared to adjust my rating upwards.

**Questions:**

See above.

---

> ### Comment · Reviewer_9iD6 · 2024-11-27
>
> Dear AC, the author has not provided a response. I will keep my score.

---

> > ### Author Response · Authors · 2024-11-30
> >
> > Thank you for your thorough review and valuable feedback on our work.
> > $\textbf{The revised paper is attached and revisions are marked in red color.}$
> > > Weakness1
> >
> > We have revised our abstract to make our explanation more straightforward.
> > - Spherical coordinates are coordinates defined in a $m$-dimensional Euclidean space which is analogous to the spherical coordinate system defined for 3-dimensional Euclidean space. The coordinates consist of a radial coordinate and $m$-1 angular coordinates.
> > - $M$ is the number of objectives in the optimization problem. LORA-MaOO fits $M$ surrogates, one surrogate is ordinal-regression-based surrogate and the remaining surrogates are regression-based. The total number of these regression-based surrogates is $M$-1.
> > - Our descriptions before the statement "which lowers the cost of using surrogates and thus enhances the optimization efficiency," explained why our method can reduce the cost of using surrogates.
> > We use only one surrogate to complete most optimization search workload and reduce the frequency of using remaining surrogates.
> >
> > > Weakness2
> >
> > - Ordinal surrogate learns ordinal relations between solutions directly, indicating that we can estimate the ordinal relations between candidates with a single surrogate and guide the model-based optimization. In comparison, for regression-based surrogates, we need fit a separate surrogate for each objective. To sort candidates and estimate their ordinal relations, all regression-based surrogates would be used to predict objective values. In addition, for classification-based surrogates, although it can predict good or bad candidates and guide the model-based optimization with a single surrogate, the candidates in good or bad categories are not comparable.
> > - Generating reference vectors is not complex but it is hard to identify the optimal hyperparameters for reference vectors. For a given MaOOP, it is hard to know how many reference vectors should we generate. In contrast, using angular surrogates to maintain diversity is a non-parametric strategy, we do not need to tune hyperparameters for reference vectors.
> >
> > > Weakness3
> >
> > The results reported in Fig. 4 support our statement. In this runtime comparison, KRVEA, KTA2, and LORA-MaOO maintain $M$ surrogates. As the number of objectives $M$ increases, the runtime of LORA-MaOO increases at a slower rate compared to KRVEA and KTA2. Therefore, compared with other algorithms maintaining $M$ surrogates, LORA-MaOO reduces the computational cost of using surrogates.
> > 	In addition, we have analyzed the time complexity of comparison algorithms in Appendix I, and the difference in their time complexities further support our statement. LORA-MaOO's runtime is proportional to $O(g + M)$, while KRVEA and KTA2 have a time complexity proportional to $O(g \times M)$.
> >
> > > Weakness4
> >
> > The current reference points represent the optimal non-dominated solutions in the archive. Initializing the population using mutants of these reference points helps enhance convergence during optimization. However, initializing the entire population solely with mutants of optimal solutions significantly increases the risk of being trapped in a local optimum. To improve population diversity and mitigate this risk, we introduce random solutions during initialization.
> >
> > Mixing optimal and random solutions for population initialization is a common strategy that has been demonstrated to be effective in the literature. For example, ParEGO initialize its population with 5 mutants of optimal solutions and 15 random solutions.
> > 	We have revised Appendix. C.2 to clarify the motivation of this initialization method.
> >
> > Considering this initialization method widely exists in the literature and is not our novelty, we have not conducted ablation studies on this. However, we purposed a clustering-based initialization strategy for these mutants of optimal solutions, so we conducted ablation studies on the parameter $n_c$ in this strategy. The result reported in Table 6 shows that our strategy is effective since the results obtained when $n_c > 1$ are better than the results of $n_c = 1$, while $n_c = 1$ denotes the situation without our initialization strategy.

---

> > > ### Author Response · Authors · 2024-11-30
> > >
> > > > Weakness 5
> > >
> > > - We made this statement because we attempted to compare several HV-based MOBO methods in our experiments. Unfortunately, when the number of objective $M$ = 8 or 10, the computation of HV values becomes extremely time-consuming. A single HV value computation can take more than one day, indicating that a single run of these MOBO methods on a benchmark problem could take more than one month, which is not affordable. Therefore, we add this clarification in our work. We have revised it to "Some HV-based MOBO methods are not compared as they failed to solve many objectives due to their unaffordable time cost for the computation of HV value in high-dimensional objective space."
> > >
> > > - We have included some reference vector-based methods, such as K-RVEA, as our baselines.
> > >
> > > - Regarding the work mentioned by the reviewer, it proposed an effective MOBO known as PSL. However, we previously reviewed this work and found that it focused solely on multi-objective optimization, not many-objective optimization. From Appendix E of this PSL work, we can observe that all problems considered in this work has only 2 or 3 objectives. Notably, this PSL work computes the HV improvement (HVI) value during optimization. Therefore, we think it is the time cost of HVI computation prevents the PSL authors from investigating MaOPs.
> > >
> > > > Weakness 6
> > >
> > > - For the ordinal surrogate, the regression result is a quantified value of ordinal relations, which fluctuates based on the ordinal relations between all solutions in the archive. Therefore, we alternatively analyzed the component contributions of our ordinal surrogate in Section 4.2 and Appendix F. Specifically, the effectiveness of ordinal levels, $\lambda$-dominance, and artificial ordinal relations are demonstrated in Tables 3, 4, and 5, respectively.
> > > - Regarding angular surrogates, they are GPs that have been widely used in the literature, the accuracy of GP regression has been demonstrated in existing studies. However, to address the reviewer's concern, we have added a new comparison to demonstrate the effectiveness of angular surrogates: We generate a variant of LORA-MaOO by removing its angular surrogates and diversity maintenance method, denoted as LOR-MaOO, and compare its performance with LORA-MaOO. The comparison is conducted on several 10-objective problems, with IGD as the performance indicator.
> > > 	```
> > > 							DTLZ1			DTLZ2			WFG4
> > > 	LORA-MaOO	2.31e-1(3.89e-2)	4.54e-1(1.41e-2)	5.47e+0(4.14e-1)
> > > 	LOR-MaOO	3.09e-1(8.54e-2)	5.07e-1(2.37e-2)	6.11e+0(5.03e-1)
> > > 	```
> > > It can be observed that LORA-MaOO outperforms LOR-MaOO, demonstrating the effectiveness of our angular surrogate based diversity maintenance method.
> > >
> > > > Weakness 7
> > >
> > > Our angular surrogate method and decomposition-based methods are not experimentally comparable due to their use of different coordinate systems for training surrogates. Specifically, our angular surrogate method is based on spherical coordinates, whereas the decomposition-based method relies on Cartesian coordinates. Therefore, we cannot simply replace the angular surrogate method in LORA-MaOO with a decomposition-based method and then conduct several optimization experiments to compare the optimization performance differences caused by changing diversity maintenance methods.
> > >
> > > In addition, Diversity maintenance methods are components of model-based optimization and must work with their underlying surrogates. Consequently, theoretical analysis of their performance is impractical as the interference from their  underlying surrogates cannot be excluded.
> > >
> > > Therefore, we can only analyze the superiority of our diversity maintenance method from the perspective of methodology. For an $M$-objective optimization problem:
> > > 1. Angular surrogate method needs $M$-1 surrogates to maintain diversity, while decomposition-based method requires $M$ surrogates.
> > > 2. Angular surrogate method is parametric-free but decomposition-based method require parameters to determine the number of subproblems.
> > > 3. Specific objective landscapes, such as degenerated Pareto fronts, have no adverse influence on angular surrogate method. In contrast, some decomposition-based methods, such as MOEA/D-EGO, use static reference vectors to define subproblems, making some subproblems less effective if their vectors do not intersect with the degenerated Pareto front.
> > >
> > >
> > > > Weakness 8
> > >
> > > We have added a time complexity analysis to Appendix I.

---

> > > > ### Author Response · Authors · 2024-11-30
> > > >
> > > > > Minor Weakness1
> > > >
> > > > Thanks for your comment.
> > > > We have added a space after this sentence.
> > > >
> > > > > Minor Weakness2
> > > >
> > > > Thank you for your comment regarding our vague description.
> > > > Pareto set denotes the solutions in the decision space which are unable to represent the balance between objectives in the objective space. Therefore, to clarify our description, we have revised it as follows:
> > > >
> > > > "A set of non-dominated solutions in the objective space that represent different optimal balance between conflicting objectives."
> > > >
> > > >
> > > > > Minor Weakness3
> > > >
> > > >  Thank you for your comment regarding our vague description.
> > > > 	Since the topic of our work is MaOO, the SAEAs referenced in our work, by default, are multi-objective and many-objective SAEAs. To clarify this, we have added the prefix `multi-/many-objective' to SAEAs.
> > > > Additionally, using a probabilistic model is associated with an acquisition function that considers model uncertainty, so we simplified our description. We have clarified this as:
> > > >
> > > > ``A SAEA is also a MOBO if it uses probabilistic models as surrogates and employs an acquisition function for candidate selection''.
> > > >
> > > > > Minor Weakness4
> > > >
> > > > Thanks for your comment. We have adjusted the sizes of these figures.

---

### Official Review · Reviewer_kgDs · 2024-10-31

**Soundness:** 2
**Presentation:** 2
**Contribution:** 2
**Rating:** 3
**Confidence:** 4

**Summary:**

The paper proposes a surrogate assisted strategy to solve (expensive, black-box) many objective optimization (MaOO) problems.
LORA-MaOO is based on the idea of making use of ordinal relations and angels to build surrogate models during optimziation which can be used to identify promising candidate points to evaluate next.
Compared to existing approaches, LORA-MaOO uses spherical coordinates to separate the concepts of convergence (to the true Pareto front) and diversity (of solutions) into distinct surrogate models.
LORA-MaOO learns one ordinal-regression based surrogate (on radial coordinates) that tries to reflect quality w.r.t convergence, i.e., to the true Pareto front, and M-1 regression surrogate models (on angular coordinates) that try to reflect diversity of solutions.
To improve the training of the ordinal surrogate they propose to create artificial ordinal levels via a clustering approach.
During the sequential optimization procedure, LORA-MaOO identifies one candidate point for next evaluation via Expected Improvement on ordinal values, and one candidate point for next evaluation based on the angles between candidate solutions and the reference points taking the point with the largest minimum angle with any point in the reference set.
In a benchmark study, LORA-MaOO is compared to 6 other baselines and competitors on 5 benchmarks of the DTLZ family (with 3, 6, 10 objectives each), 9 benchmarks of the WFG family (with 3, 6, 10 objectives each), and NAS-BENCH 201 with 5 or 8 objectives.
In general, LORA-MaOO performs well and outperforms most competitors on some benchmarks with respect to quality of solutions and diversity and with respect to runtime being slightly better compared to other surrogate assisted approaches that leverage multiple surrogates.
The paper also presents results of an ablation study on the hyperparameters of LORA-MaOO (i.e, the minimal number of ordinal levels; dominance coefficient; ratio threshold of reference points and the clustering number for reproduction).

**Strengths:**

The introduction of MaOOP and their characteristics and challenges is done well and LORA-MaOO is inspired by these challenges in a reasonable way, although motivated heuristically.
The general idea of multiple surrogate models based on spherical coordinates is apparently novel to some extent and interesting and so is the separation of quality vs. diversity when proposing candidate points.
In general, the paper has a clear defined structured and is written reasonably well.
Multi Objective Opimization problems and also MaOOP are a relevant field that is also connected to the ML and DL community, as the authors try to demonstrate based on the NAS-Bench-201 benchmarks.

**Weaknesses:**

While the idea of LORA-MaOO of using multiple surrogates based on spherical coordinates and having one ordinal model that captures quality whereas M-1 models capture diversity is somewhat reasonable, it is mostly inspired in a heuristic way, as is the definition of the criteria used to select each of the two candidate points from the two different modeling approaches during a proposal step.

LORA-MaOO does perform well on the benchmarks presented in the paper, however, improvements do depend on benchmarks and number of objectives at hand and especially anytime performance can vary.
Overall, improvements appear to be rather mild on average.
Gains in efficiency mostly come from leveraging the different modeling approaches during the candidate prediction step (i.e. one candidate point based on the ordinal regression model, on radial coordinates, and one candidate point based on the M-1 regression models, on angular coordinates) which results in speed up during inference, however, LORA-MaOO still requires the training of M independent models.
Further, it is not discussed if and how multi-task modeling approaches could be useful here, especially for the M-1 regression models on angular coordinates.

The actual proposal step of LORA-MaOO is a two-step procedure: One candidate is proposed based on quality criteria of the ordinal regression model on radial coordinates, One candidate is proposed based on diversity criteria of the M-1 regression models on angular coordinates.
While this seems to be reasonable it is a heuristic approach that tries to capture this trade-off in improving quality vs. increasing / maintaining diversity.
Maybe designing an adequate acquisition function jointly on both surrogates could add in adding some technical depth to the proposed method.

One strong downside is that the paper does not have a strong direct connection to the ML and DL community but would be better suited within the black-box optimization community.
While the authors try to motivate relevance by benchmarking also on NASBench201, these benchmarks feel very much constructed and in my opinion are not suited as many objective problems:
For example, the 5 objectives for the first NASBench201 benchmark on CIFAR-10 presented in the paper are accuracy, number of flops, number of parameters, latency and energy cost. However, these objectives are usually strongly correlated, especially number of flops and number of parameters as well as latency and energy cost, because they are mostly tied to the complexity of the neural architecture.
In essence there is only one trade-off in this benchmark (accuracy vs. proxy measures of model complexity).
Generally, performance of LORA-MaOO compared to the limited number of competitors run on this benchmark is not convincing.
Moreover, applicability of LORA-MaOO to real-world NAS problems is strongly limited because it operates in the full-fidelity black-box setting and the authors do not discuss extensions to grey-box multi-fidelity variants, i.e., speeding up the costly evaluation of architectures by terminating the evaluation of poor-performing architectures after a low number of epochs trained which is the de-facto standard nowadays when trying to make NAS more sample- and cost-efficient.

Overall, due to limited direct relevance to the ML and DL community which is the main target audience of ICLR and limited improvements over baselines and competitors as well as mostly heuristic motivation of the algorithm and its component, I have a hard time recommending acceptance.

**Questions:**

What I felt was missing in the introduction is some background on expensiveness and black-box nature of the problems. The paper surely feels to be positioned in this expensive black-box setting but this is not made clear in the abstract or introduction. Could the authors maybe improve that?

2.1 Why is ParEGO listed under 2.1 and not 2.2? As far as I know, it can be considered a standard example for a scalarized MO BO approaches?

2.2 The connection between MOBO and SEAEs is not well established, can the authors maybe revisit that?

EQ (3) Can you elaborate on the $\max(f_\{in}(x))$ part when defining $g_{\lambda,i}$ - is this actually added to all $g_{\lambda,i}$ and increases them by the same amount, i.e., a flat penalty? Because the $\max$ is taken over all $j = 1, ..., M$ objectives?

EQ (2) Is there any particular reason to denote the normalized objective as $f_{in}(x)$?
It is not initially clear what the $n$ would stand for especially when looking at $f_{jn}$ in EQ(3)?

EQ (4) Within the extension coefficient: The assignment formula $v_{i} = 1 - (i-1/N_{o}-1)$ assumes each ordinal level to represent an equal increment of quality from the reference set. Can you discuss this assumption? E.g. what about solutions near the reference boundary vs. solutions further away?

Can you provide some details with respect to estimated runtime overhead of the clustering approach to introduce artificial ordinal levels?

Overall LORA-MaOO still trains M surrogate models in total; can the authors elaborate on the claim that LORA-MaOO is more efficient with respect to surrogate modeling compared to other approaches?
In Line 287 this is discussed to some extent, i.e, the ordinal surrogate being used for one candidate whereas the regression surrogates are used for the other candidate, however, this still requires to train M models and only inference is sped up?
Also based on Figure 4, efficiency gains with respect to runtime are not substantial and competitors such as PAREGO, CSEA and REMO naturally are more efficient than LORA-MAOO due to their different modeling approaches.

Was there any specific reason to present DTLZ in the main paper but not WFG (except for limited space)?

As you are comparing LORA-MaOO to 6 competitors on many benchmarks of the DTLZ and WFG family via Wilcoxon signed-rank tests: Have you corrected for multiple comparisons? Based on the table caption in Table 1 you state that you test at an $\alpha$ level of $0.05$. However, as you are repeatedly comparing the same methods on similar benchmarks there is some multiple comparison testing issue that should be considered when performing the Wilcoxon signed-rank tests. Otherwise you risk false positive results.

Was there a specific reason to exclude ParEGO as an MO BO baseline in the NASBench201 benchmarks?

Some further minor comments/questions:

L107: I would rather write "scalarized" instead of "aggregate" objective

L109: In ParEGO the Tchebycheff scalarization is newly performed within each step of BO and not pre-defined and constant. Maybe the authors can revisit that section and correct this?

L214: "is" appears to be missing ("that used for training")

Eq(3) There appears to be some space missing between $\lambda$ and $\max$

LORA-MaOO is likely not a good idea name for the method due to the very popular LoRA method in the context of LLMs so people might draw wrong conclusions about the paper based on the title and name of the method.

Writing quality is somewhat decreasing in the experimental setup section, i.e., missing verbs in L346 and L353.
Can the authors maybe revisit that section?

In 4.2 results of the ablation study are presented first before presenting the main results in 4.3.
Maybe reversing the order would be better for the flow of the paper.
Also, the ablation study is in essence meta-configuration of LORA-MaOO (i.e., Hyperparameter optimization) but on (partially) the same benchmarks, LORA-MaOO is then compared to competitors; this may result in overfitting to the benchmarks. Can the authors comment on that?

**Details Of Ethics Concerns:**

There are no ethical issues.

---

> ### Comment · Reviewer_kgDs · 2024-11-26
>
> As the authors have not provided a rebuttal, I will keep my score.

---

> ### Author Response · Authors · 2024-12-03
>
> Thank you for your thorough review and valuable feedback on our work.
>
> > Weakness 1.
>
> We would like to clarify two points:
> 1. Our selection criteria are not heuristic. For example, our convergence criterion is Expected Improvement (EI).
> 2. Heuristic components should not be considered as a weakness. Many existing Bayesian Optimization (BO) methods and gradient-free optimization methods use heuristic methods to generate candidates, such as ParEGO, MOEA/D-EGO. If containing heuristic components can be a weakness for rejection, then many existing publications on BO and gradient-free optimization would be rejected.
>
> > Weakness 2.
>
> - $\textbf{Performance}$: For comparison algorithms with $M$ surrogates, LORA-MaOO outperforms them in both optimization results and runtime. For comparison algorithms with only one surrogate, LOEA-MaOO significantly outperforms them in optimization results. Our statistical test results show significant performance improvement.
>
> - $\textbf{Time complexity}$: We have added a time complexity analysis in Appendix I, although LORA-MaOO trains $M$ surrogates, it remains more efficient than other model-based approaches with $M$ surrogates.
>
> - $\textbf{Multi-task modeling}$: It is unclear why a discussion about multi-task modeling approaches is required. Multi-task modeling is irrelevant to our work. Could the review provide further explanations about this?
>
> > Weakness 3.
>
> One of our main contributions is the proposal of a new framework to model surrogates from spherical coordinates, allowing LORA-MaOO to address convergence and diversity separately using independent surrogates. Specifically, convergence is handled by our ordinal surrogate without interference from angular surrogates, and diversity is handled by our angular surrogates without interference from the ordinal surrogate.
>
> This framework plays a crucial role in improving computational efficiency of LORA-MaOO.
> Designing an acquisition function jointly on both surrogates opposes our contribution, which will make our work similar to the existing studies in the literature.
>
> > Weakness 4.
>
> - $\textbf{Connection to ML}$
>
> Model-based optimization for expensive optimization problems, such as Bayesian optimization, is an important topic in the optimization field and it has a strong connection to the ML community. Many studies on this topic have been published in ICLR [1] and other ML conferences [2], where novel surrogate modeling methods are proposed for optimization purposes.
>
> ```
> [1] Zhao, Yiyang, et al. "Multi-objective Optimization by Learning Space Partitions." International Conference on Learning Representations (ICLR'22). 2022.
> [2] Lin, Xi, et al. "Pareto Set Learning for Expensive Multi-objective Optimization." Advances in Neural Information Processing Systems 35 (NeurIPs'22). 2022.
> ```
>
> Our work has similar connection to the ML community as the above works: We proposes a new strategy to learn surrogates for expensive optimization. Specifically, we discuss the representation of ordinal relations in the high dimensional objective space and develop novel surrogate modeling methods to learn ordinal relations and angles for optimization problems.
>
> - $\textbf{NAS experiments}$
>
> We have two NAS problems in our experiments.
> 	The correlation matrix for 5-objective NAS problem is:
> ```
> 	[1.0000  -0.6045  -0.6037  -0.2864  -0.3086]
>  	[-0.6045  1.0000  1.0000  0.4458  0.4966]
> 	[-0.6037  1.0000  1.0000  0.4446  0.4955]
> 	[-0.2864  0.4458  0.4446  1.0000   0.9899]
> 	[-0.3086  0.4966  0.4955  0.9899  1.0000]
> ```
> Although (obj2, obj3) and (obj4, obj5) are strong correlated, this problem can still be a 3-objective problem.
>
> The correlation matrix for 8-objective NAS problem is:
> ```
> 	[ 1.0000  -0.1283  -0.1277  -0.5987  -0.5883  -0.1778  -0.1670  0.0783]
>  	[-0.1283   1.0000   1.0000   0.4437   0.5189   0.8387   0.8666   0.7155]
>  	[-0.1277   1.0000   1.0000   0.4425   0.5178   0.8390   0.8670   0.7156]
>  	[-0.5987   0.4437   0.4425   1.0000   0.9940   0.4993   0.4830   0.0795]
>  	[-0.5883   0.5189   0.5178   0.9940   1.0000   0.5651   0.5521   0.1392]
>  	[-0.1778   0.8387   0.8390   0.4993   0.5651   1.0000   0.9975   0.2859]
>  	[-0.1670   0.8666   0.8670   0.4830   0.5521   0.9975   1.0000   0.3397]
>  	[ 0.0783   0.7155   0.7156   0.0795   0.1392   0.2859   0.3397   1.0000]
> ```
> Although (obj2, obj3), (obj4, obj5), and (obj6, obj7) are strong correlated, this problem can still be a 5-objective problem, which is a many-objective problem.

---

> ### Author Response · Authors · 2024-12-03
>
> > Weakness 4. (continue)
>
> - $\textbf{Applicability of LORA-MaOO}$:
>
> We would like to clarify that we evaluate the performance of LORA-MaOO on NAS problems in the black-box setting since NAS problems are widely used as benchmarks for expensive multi-/many-objective optimization algorithms in the literature [1, 3].
> ```
> [1] Zhao, Yiyang, et al. ”Multi-objective Optimization by Learning Space Partitions.” International Conference on Learning Representations (ICLR’22). 2022.
> [3] Shuhei Watanabe, et al. "Speeding up multi-objective hyperparameter optimization by task similarity-based meta-learning for the tree-structured parzen estimator." International Joint Conference on Artificial Intelligence (IJCAI'23), 2023.
> ```
>
> To ensure a fair comparison, in our experiments, we follow the conventional NAS problem setting used in the literature. We did not discuss extensions to the grey-box setting because our work is a generalized algorithm to solve expensive MaOOPs, not one specifically developed for NAS problems. Therefore, discussing specific settings for NAS problems is unnecessary. Similarly, existing studies in the literature did not discuss such extensions for the same reason.
>
> Additionally, we cannot see any limitations that prevents LORA-MaOO from being adapted to solve grey-box NAS problems. A specific variant of LORA-MaOO for grey-box NAS problems could be a future work.
>
> > Weakness Summary
>
> We have provided evidence showing the connection between our topic and the ML community, many studies on the same topic have been published in major ML conferences such as ICLR and NeurIPs.
>
> Statistical test results demonstrate that our optimization performance improvement over the baselines is significant, especially for optimization algorithms with only one surrogate (e.g. ParEGO, CSEA, OREA, REMO). In addition, cost of using surrogates is reduced when compared to optimization algorithms maintaining $M$ surrogates (e.g. K-RVEA, KTA2).
>
> In addition, the only heuristic component in our LORA-MaOO is the use of a PSO to generate candidates, while other components such as the selection criteria, are not heuristic.
>
> > Question 1.
>
> The following explanations would be added to Section I:
> However, in many real-world MOOPs and MaOOPs, the evaluation of solution performance could be costly in terms of computational time or financial budget. Therefore, the evaluation budget allows only a limited number of solutions to be evaluated on the expensive objective functions, making these problems expensive MOOPs or expensive MaOOPs.
>
> > Question 2.
>
> ParEGO can also be considered as a standard SAEA. It uses GPs as surrogates and employs an EA to generate candidates. At the end of Section 2.2, we described the connection between SAEAs and MOBO and explained that ParEGO can be considered either an SAEA or a MOBO.
>
> > Question 3.
>
> We have revised it to clarify our explanation, the acquisition functions in BO are highlighted.
>
> > Question 4.
>
> Yes. All $g_{\lambda,i}$ are increased by the same amount $\max(f_{in}(x))$.
>
> > Question 5.
>
> The word 'normalization' begins with the character 'n'. EQ (3) is followed by its notation explanations.
> We have detailed our notation explanations.
>
> > Question 6.
>
> In the last paragraph of Section 3.2.1, we explained that LORA-MaOO does not use extension coefficients directly. To generate a stable ordinal surrogate, extension coefficients are divided into several ordinal levels. In this way, a solution's ordinal relation value will not fluctuate frequently over optimization iterations. In addition, the ordinal surrogate predicts the ordinal relation between solutions instead of accurate objectives, so the key point is that there is an increment between two successive ordinal values, it does not matter if the increments are equal.
>
>
> > Question 7.
>
> In the context of expensive optimization, the number of evaluated solutions in the archive is small. So the runtime of this clustering operation is trivial and negligible to LORA-MaOO.
>
> > Question 8.
>
> In our abstract, we stated that $M$-1 angular surrogates were only used to select diverse optimal candidate solutions. In Section 3.2, we explained that $M$-1 angular surrogate were used in the selection procedure for diversity but are idle in the search procedure. Therefore, although LORA-MaOO maintains $M$ surrogates, it is more efficient than other algorithms that maintain $M$ surrogates.
>
> We have added a time complexity analysis in Appendix I to support this claim.
>
> In Section 4.5, lines 516 to 521, we explained that ParEGO, CSEA, and REMO do not have $M$ surrogates, as a result, they are more efficient but less effective than LORA-MaOO.
>
> > Question 9.
>
> No. It does not matter which results are presented in the main paper.
>
> WFG has 9 problems, while DTLZ has 7 problems. Presenting a table of DTLZ results instead of WFG results would save 10 columns of space (2 problems on 5 scales) in the main file.

---

> ### Author Response · Authors · 2024-12-03
>
> > Question 10.
>
> We would like to clarify that the statistical tests used in our experiments are $\textbf{Wilcoxon rank sum}$ tests (see the caption of Table 1), not Wilcoxon signed-rank tests. Therefore, the multiple comparison testing issue raised by the reviewer does not apply to our work.
>
> > Question 11.
>
> In Section 4.4, we explained that the 3 algorithms with the best performance in previous benchmark experiments were selected.
>
> > Question 12.
>
> - We have revised the word 'scalarized'.
> - The EQ(1) in ParEGO paper pre-defined a set of weight vectors $\lambda$. In each iteration of the algorithm, a weight vector $\lambda$ is drawn uniformly at random from this constant set.
> - We have added 'is'.
>
> > Question 13.
>
> We have added a space between them.
>
> > Question 14.
>
> LORA-MaOO is an abbreviation that highlights the algorithm's features: learning ordinal relations and learning angles. Although the LoRA method is well-known in the context of LLMs, this work is unrelated to LLMs and focuses on optimization. Additionally, we have double checked that the term 'LLM' does not appear in our abstract or manuscript. Therefore, readers will not be misled.
>
> > Question 15.
>
> We have revised our manuscript to correct them.
>
> > Question 16.
>
> - $\textbf{Order of Ablation Studies}$:
>
> We are unsure why this review suggests that the ablation study should not be presented before the main results. In fact, if ablation studies are not conducted to demonstrate the effectiveness of algorithm components, the main experiments would be less convincing.
>
> Many publications present their ablation studies before the main results. For example, we noticed that the 10-score submission:
> ```
> Scaling In-the-Wild Training for Diffusion-based Illumination Harmonization and Editing by Imposing Consistent Light Transport
> ```
> from ICLR'25 presents its ablation studies (in Section 4.2) before its main results (Sections 4.3 and 4.4).
>
> - $\textbf{Overfitting}$:
>
> We have evaluated our algorithm's performance on 16 different benchmark problems, each of them has unique features and properties. It seems unreasonable to say that a parameter setting overfits 16 different problems with 5 scales.
>
> A different parameter setting for LORA-MaOO may achieve better optimization results on some benchmark problems but worse on other benchmark problems, demonstrating that it is impossible to achieve optimal optimization performance across all benchmark problems with a single parameter setting. In total, 80 benchmark instances across 16 problems and 5 different scales cannot be overfitted by a single parameter setting.

---

### Official Review · Reviewer_epn3 · 2024-11-04

**Soundness:** 2
**Presentation:** 2
**Contribution:** 2
**Rating:** 5
**Confidence:** 4

**Summary:**

This work proposes a learning ordinal relations and angles (LORA) model-based method for expensive many-objective optimization (MaOO). The key contributions are 1) separate surrogate model buildings, which include one ordinal regression-based model for convergence and a set of surrogate models based on angular coordinates for maintaining diversity, and 2) a non-parametric approach to select diverse solutions for expensive evaluations. Experimental results show that the proposed method (LORA-MaOO) can outperform other surrogate-assisted evolutionary algorithms on different test problems.

**Strengths:**

+ Expensive many-objective optimization is important for real-world applications.

+ The proposed LORA-MAOO method can outperforms other surrogate-assisted methods on different test problems.

**Weaknesses:**

**1. Motivation on the Cost of Surrogate Model**

The key motivation of this work is to reduce the cost of using surrogate models for expensive optimization. However, in the expensive optimization setting, the cost of building and using surrogate models (e.g., a few minutes) could usually be trivial compared with the truly expensive solution evaluation cost (e.g., days or weeks, or huge financial cost). A detailed and convincing discussion is needed to support the claim that reducing the cost of using surrogate models is important for expensive optimization.

**2. Solution Selection Method**

The non-parametric solution selection approach is another main contribution of this work. However, its advantages over existing methods are not clearly discussed. For example, can we use the crowding distance in NSGA-II to replace the proposed angle-based diversity criterion to select candidate solutions? Many other diversity-based indicators have been proposed in the multi-objective optimization community but have not been discussed or compared in this work.

In addition to the computational complexity, what other advantages does the proposed method have compared with hypervolume?

No theoretical analysis is provided for the proposed solution selection method.

**3. Hypervolume-based Method**

In Section 2.2, this work claims that "the time complexity of computing HV increases exponentially with the number of objectives, which may limit the application of MOBO methods on MaOOPs". However, many efficient hypervolume-based methods have already been proposed for many-objective optimization (e.g., [1,2]). All these methods are not discussed and compared with in this work.

[1] An Expensive Many-Objective Optimization Algorithm Based on Efficient Expected Hypervolume Improvement, IEEE Transactions on Evolutionary Computation 2023.

[2] Surrogate-Assisted Environmental Selection for Fast Hypervolume-Based Many-Objective Optimization, IEEE Transactions on Evolutionary Computation 2024.

**4. Pareto Front Approximation for Expensive Many-Objective Optimization**

The proposed method aims to find a set of solutions to approximate the Pareto front of the many-objective optimization problem. However, in the expensive optimization setting, a small set of evaluated solutions is clearly inadequate to approximate the Pareto front well for problems with many optimization objectives (e.g., 10). A more reasonable choice is to find specific trade-off solution(s) as in [3].

A more detailed discussion on how a small set of evaluated solutions could be useful for many-objective optimization in practice will be very helpful in motivating this work.

[3] The Kalai-Smorodinsky Solution for Many-Objective Bayesian Optimization, JMLR 2020.

**5. Experiments**

- The comparison with some expensive many-objective optimization methods such as [1,2] is missing.

- To better understand the proposed method's performance, it could be very helpful to compare it with MOBO methods on multi-objective optimization problems (2 and 3 objectives) with the found approximate Pareto front.

- According to the results in Table 4, the proposed method does not show a clear advantage in the run time. Is it contradictory to the motivation for reducing the cost of using surrogate models for expensive optimization?

**Questions:**

See weaknesses.

---

> ### Comment · Reviewer_epn3 · 2024-11-27
>
> Since no response is provided, I will keep my original score for this paper.

---

> ### Author Response · Authors · 2024-12-01
>
> Thank you for your thorough review and valuable feedback on our work.
> > Weakness 1: Motivation on the Cost of Surrogate Model
>
> We will discuss our motivation from the following perspectives:
>
> - $\textbf{Real-World Industrial Applications}$: Many real-world industrial applications, e.g., aircraft design and engine calibration, need to pre-calibrate their designs using simulations before calibrating with expensive test facilities. During this pre-calibration process, the computational cost of MaOO methods is not trivial.
>
> - $\textbf{Algorithm Efficiency}$: We aim to reach diverse optimal solutions with a relatively low cost of using surrogates, which indicates our priority is the quality of optimization results. It would be meaningless for expensive MaOO algorithms to save time cost if they fail to acheive desirable optimization performance.
> 	However, when two algorithms have similar effectiveness, the more efficient algorithm is definitely preferable.
>
> - $\textbf{Time cost for expensive MaOOPs}$. For expensive MaOOPs, the cost of using surrogates increases with the number of objectives $M$ and the number of solutions for modeling $N$. Consequently, the overall cost of using surrogates is not trivial when both $M$ and $N$ are large. For instance, in our experiments, using 10 surrogates to predict objective values and then compute Hypervolume values for an archive of 300 non-dominated solutions is very time-consuming, making the HV-based optimization algorithm fails to generate a new candidate within one day. As a result, many HV-based MOBO in the literature consider only MOOPs with an evaluation budget of 100. However, many real-world applications have many objectives and have evaluation budgets exceeding 300. On the other hand, if the computational cost of optimization algorithms is completely trivial to expensive evaluation cost, research studies such as [1] would be meaningless.
>
> Therefore, an effective and more efficient method is actually meaningful to our field. The situation mentioned in the comment (cost is trivial) happens only when $M$ and $N$ are small.
>
> [1] An Expensive Many-Objective Optimization Algorithm Based on Efficient Expected Hypervolume Improvement, IEEE Transactions on Evolutionary Computation 2023.
>
> > Weakness 2: Solution Selection Method
> 2.1.
>
> We would like to clarify that our main contribution is the framework of modeling from spherical coordinates, as the first point listed in our contributions at the end of Section I. Our two following contributions (points 2 and 3), the ordinal-regression-based surrogate for many-objective optimization and the angular surrogate based non-parametric diversity maintenance method, are closely linked to this primary contribution.
> 	Our surrogate modeling framework makes it possible to consider convergence and diversity separately. As a result, the key points in our diversity maintenance method are:
> 1. We use $M$ - 1 angular surrogates to maintenance diversity;
> 2. The method is parameter-free.
>
> In comparison, most existing diversity-based indicators, such as the crowding distance in NSGA-II, are based on Cartesian coordinates and rely on $M$ surrogates, thus they cannot replace our angle-based method.
> 	Additionally, some existing methods use a set of reference vectors to maintain diversity, such as ParEGO and K-RVEA. However, it is difficult to identify the optimal density of reference vectors before optimization begins.
>
> > 2.2.
>
> Computational complexity is the main limitation of Hypervolume method compared to other diversity maintenance methods. As a result, Hypervolume is mainly used in multi-objective optimization and is considered when the evaluation budget is very small (A larger evaluation budget indicates more evaluated non-dominated solutions, making the computation of Hypervolume more time-consuming). Due to the significance of Hypervolume time complexity, many studies are motivated to improve Hypervolume method and reduce its computation time [1].
> 	From this perspective, our method obtained competitive results (as reported in our response to Weakness 3) and overcomes the drawback of Hypervolume method. We think this advantage is significant enough when compared to existing HV-based MOBO methods.
>
> In addition, our surrogate modeling strategy in the spherical coordinate system is completely different from existing studies, it develops a new approach to solving expensive MaOOPs rather than extending any existing methods, which can offer new insights to the expensive optimization community.
>
> [1] An Expensive Many-Objective Optimization Algorithm Based on Efficient Expected Hypervolume Improvement, IEEE Transactions on Evolutionary Computation 2023.
>
> > 2.3.
>
> This is an experimental study rather than a theoretical one. Therefore, the theoretical analysis is limited. However, we included a time complexity discussion in Appendix I, which covers the time complexity of angular surrogates used for angle prediction.

---

> > ### Author Response · Authors · 2024-12-01
> >
> > > Weakness 3
> >
> > We have reviewed the references mentioned by this comment. [1] proposed an HV-based MOBO approach, NSGAIII-EHVI, to address expensive MaOOPs. [2] introduced an efficient surrogate-assisted greedy inclusion algorithm, SAGIF, to estimate HV values, however, SAGIF is not an independent MaOO algorithm. The experiments in [2] were conducted on several datasets rather than MaOOPs.
> >
> > As for NSGAIII-EHVI, since its paper did not provide access to the source code, we reproduced the experiments reported in [1] to compare it with our LORA-MaOO.
> > 	The experimental setup is exactly the same as described in [1], where the number of initial sample sets evaluated with expensive functions is set to 20, and the maximum number of evaluations with expensive functions is set to 100. Experimental results are reported as follows:
> > ```
> > 		M			LORA-MaOO			NSGAIII-EHVI
> > DTLZ1 	8			8.77e+0(4.76e+0)		1.15e+1(5.84e+0)
> > 		10			5.39e-1(3.26e-1)		5.92e-1(3.52e-1)
> > DTLZ2	8			6.28e-1(5.98e-2)		5.54e-1(2.18e-2)
> > 		10			6.89e-1(4.43e-2)		5.56e-1(1.85e-2)
> > DTLZ3	8			3.52e+1(1.42e+1)		3.66e+1(1.73e+1)
> > 		10			1.59e+0(7.59e-1)		2.81e+0(2.13e-1)
> > DTLZ4	8			8.98e-1(1.16e-1)		6.57e-1(5.40e-2)
> > 		10			8.21e-1(6.72e-2)		6.85e-1(3.19e-2)
> > DTLZ5	8			9.11e-2(2.68e-2)		9.53e-2(1.88e-2)
> > 		10			2.00e-2(9.10e-3)		3.91e-2(6.05e-3)
> > DTLZ6	8			1.56e+0(3.85e-1)		1.73e+0(3.98e-1)
> > 		10			2.29e-1(1.85e-1)		2.25e-1(1.36e-1)
> > DTLZ7	8			4.41e+0(3.64e+0)		3.03e+0(5.88e-1)
> > 		10			1.74e+0(2.77e-1)		3.05e+0(8.14e-1)
> > ```
> > It shows that LORA-MaOO outperforms NSGAIII-EHVI on DTLZ1, DTLZ3, and DTLZ5, reaches comparable results on DTLZ6 and DTLZ 7, and is inferior to NSGAIII-EHVI on DTLZ2 and DTLZ4.
> > 	The overall performance of LORA-MaOO is better than that of NSGAIII-EHVI.
> >
> > > Weakness 4
> >
> > We have reviewed [3] and noticed that this study is based on an important assumption: all objectives have a priori equal importance. The approach to identifying specific trade-off solutions in [3] is not applicable to many real-world applications since such an assumption is not realistic for these applications.
> >
> > In practice, although a small set of solutions is inadequate to approximate the Pareto front, a set of non-dominated solutions can be identified. MaOO algorithms aim to ensure the obtained non-dominated solutions are well-distributed and move toward the Pareto front [1]. Well-distributed solutions enable the identification of different trade-off solutions based on diverse demands, while moving toward the Pareto front indicates maximizing the benefit of each objective.
> >
> > [1] An Expensive Many-Objective Optimization Algorithm Based on Efficient Expected Hypervolume Improvement, IEEE Transactions on Evolutionary Computation 2023.
> > [3] The Kalai-Smorodinsky Solution for Many-Objective Bayesian Optimization, JMLR 2020.
> >
> > > Weakness 5
> > 5.1.
> >
> > We have added new comparison experiments with NSGAIII-EHVI and results are reported above.
> >
> > > 5.2
> >
> > We have compared [1] on problems with the number of objective $M$ = 3.
> > ```
> > 						LORA-MaOO			NSGAIII-EHVI
> > DTLZ1 			8.44e+1(3.09e+1)		9.31e+1(3.14e+1)
> > DTLZ2			2.90e-1(9.14e-2)		2.15e-1(3.69e-2)
> > DTLZ3			2.48e+2(6.53e+1)		2.83e+2(8.19e+1)
> > DTLZ4			8.85e-1(1.86e-1)		6.21e-1(2.07e-1)
> > DTLZ5			1.02e-1(6.47e-2)		1.21e-1(3.84e-2)
> > DTLZ6			5.05e+0(8.34e-1)		5.12e+0(5.89e-1)
> > DTLZ7			1.65e+0(1.28e+0)		4.88e-1(2.67e-1)
> > ```
> > It shows that LORA-MaOO is comparable to NSGAIII-EHVI when $M$ = 3.
> >
> > In addition, we are unclear the purpose of comparing with multi-objective MOBO in our experiments.
> > Our paper title is 'many-objective optimization' rather than 'multi-objective optimization', and some novelties in our work, such as the artificial ordinal relations, are specifically developed to address the difficulties unique to the scenario of many-objective optimization. In our humble opinion, comparing with optimization algorithms that are unable to handle many objectives cannot help us better understand the performance of LORA-MaOO, especially the contributions of components such as artificial ordinal relations.
> >
> > > 5.3.
> >
> > As we discussed in our response to weakness 1: We aim to reach diverse optimal solutions with a relatively low cost of using surrogates. So we need to discuss these comparison algorithms in two categories:
> > 1. When compared to the algorithms maintaining $M$ surrogates, such as KRVEA and KTA2, our advantage is the reduced runtime. Although LORA-MaOO also maintains $M$ surrogates, it shows a slower increase in runtime as $M$ increases. Our time complexity analysis in Appendix I demonstrates that LORA-MaOO's runtime is proportional to $O(g + M)$, while KRVEA and KTA2 have a time complexity proportional to $O(g \times M)$.
> > 2. In comparison to the algorithms that maintain only one surrogate, such as ParEGO, CSEA, and OREA, whose time complexity $O(g)$ is irrelevant to $M$, LORA-MaOO significantly outperforms them due to its effective diversity maintenance method.

---

> > > ### Comment · Reviewer_epn3 · 2024-12-03
> > >
> > > Thank you for your response, and I have also read other reviewers' comments.
> > >
> > > I will adjust my score after discussing with other reviewers in the reviewer's discussion phase.

---

### Official Review · Reviewer_EX8L · 2024-11-04

**Soundness:** 2
**Presentation:** 2
**Contribution:** 2
**Rating:** 5
**Confidence:** 4

**Summary:**

MaOO often faces the issue of higher costs with the number of objectives increases. This paper addresses this problem by proposing LORA-MaOO, a surrogate-assisted MaOO algorithm that learns surrogates from spherical coordinates. The LORA-MaOO attains competitive experimental results on the multi/many-objective benchmarks DTLZ, WFG5, and the neural architecture search benchmark NASbench201, while necessitating a comparatively low runtime. However, the experimental section has several issues, such as limited improvement in performance and a lack of Intuitive explanation.

**Strengths:**

1.	The paper is easy to read, with the background information and proposed framework explained clearly.
2.	Authors conducted extensive experiments to test their method.
3.	The problem of MaOO is clear.

**Weaknesses:**

1.	The presentation of the paper is bad. In the abstract, the authors provide too much detail about proposed method. However, more description about issue, behind reason, and necessary explanation are more important than details.
2.	The title of Section 3 is too long.
3.	The Figure 1 is not clear. Same questions can be found in, Figure 2, Figure 3, and Figure 4.
4.	Figure 2 is too big. This is illegal to rules of ICLR.
5.	This paper lacks a comprehensive comparison about costs. The main goal of the paper is to reduce costs. However, the authors did not provide any experiments about costs.
6.	Provide a more thorough discussion of the generalization ability, robustness, and potential applications of the proposed approach.
7.	The format of references is not correct.
8.	The main limitation of this paper is that proposed method lacks theriacal analysis, ablation study, and visualization.
9.	Exploring the reasons behind the success of these techniques and providing intuitive explanations would contribute to the overall scientific contribution of the work.

**Questions:**

see Weaknesses

---

> ### Comment · Reviewer_EX8L · 2024-11-26
>
> Dear AC, the author has not provided a response. I will keep my score.

---

> > ### Author Response · Authors · 2024-12-01
> >
> > Thank you for your thorough review and valuable feedback on our work. $\textbf{The revised paper is attached and revisions are marked in red color.}$
> >
> > > Weakness1
> >
> >  We have revised our abstract and added some descriptions about issues, we also added explanations about why our method can reduce the cost of using surrogates.
> >
> > > Weakness2
> >
> > We have revised our title of Section 3 to shorten it. New title is "LORA-MaOO: the Proposed Algorithm".
> >
> > > Weakness3
> >
> > We have revised our figures to make them large and clear.
> >
> > > Weakness4
> >
> > We have adjusted the size of Figure 2. Some subfigures are moved to Appendix to ensure each subfigure is large and clear, while the size of Figure 2 is legal.
> >
> > > Weakness5
> >
> > The runtime of model-based algorithms represents the cost of using surrogates. In Section 4.5, we compared the runtime of the comparison algorithms, and the following observations can be made:
> > 1. When compared to the algorithms maintaining $M$ surrogates, such as KRVEA and KTA2, although LORA-MaOO also maintains $M$ surrogates, it shows a slower increase in runtime as $M$ increases. Our time complexity analysis in Appendix I demonstrates that LORA-MaOO's runtime is proportional to $O(g + M)$, while KRVEA and KTA2 have a time complexity proportional to $O(g \times M)$.
> > 2. In comparison to the algorithms that maintain only one surrogate, such as ParEGO, CSEA, and OREA, whose time complexity $O(g)$ is irrelevant to $M$, LORA-MaOO significantly outperforms them due to its effective diversity maintenance method.
> >
> > > Weakness6
> >
> > - Our LORA-MaOO is designed to address a broad range of expensive multi-objective and many-objective optimization problems (MOOPs and MaOOPs), rather than being tailored to any specific optimization problem. Currently, LORA-MaOO is applicable to continuous optimization and can be adapted for combinatorial optimization by substituting individual operations (e.g., mutation) with those suited for combinatorial tasks.
> > 	Additional specialized techniques may be required to extend LORA-MaOO for solving expensive MaOOPs with additional optimization features, such as constrained optimization, large-scale optimization, and sparse optimization.
> >
> > - Expensive MOOPs and MaOOPs widely exist in various real-world industrial applications [1, 2] and LORA-MaOO can be employed to solve them.
> > 	[1] Koziel, Slawomir, and Adrian Bekasiewicz. "Multi-objective optimization of expensive electromagnetic simulation models." Applied Soft Computing 47 (2016): 332-342.
> > 	[2] Yu, Xunzhao, et al. "Internal combustion engine calibration using optimization algorithms." Applied Energy 305 (2022): 117894.
> >
> > > Weakness7
> >
> > Thanks for your comment. we would double check all references and ensure their formats are correct.
> >
> > > Weakness8
> >
> > - Many expensive optimization problems are black-box optimization problems with unknown properties, making theoretical analysis challenging unless several assumptions are made about the problem. In addition, common theoretical analysis in our field is runtime analysis [3, 4], although we did not include a formal runtime analysis in our work, we provided a runtime comparison experiment in Section 4.5 and a time complexity analysis in Appendix I.
> > [3] Weijie Zheng, Benjamin Doerr: Mathematical runtime analysis for the non-dominated sorting genetic algorithm II (NSGA-II). Artif. Intell. 325: 104016 (2023)
> > [4] Benjamin Doerr, Zhongdi Qu: A First Runtime Analysis of the NSGA-II on a Multimodal Problem. IEEE Trans. Evol. Comput. 27(5): 1288-1297 (2023)
> >
> > - We have reported ablation studies in Section 4.2, Figure 1 and Appendix F, where we investigated the contributions of our algorithm components: the minimal number of ordinal levels $n_o$, the dominance coefficient $\lambda$, the ratio threshold of reference points $rp\_{ratio}$, and the clustering number for reproduction $n_c$. These components play an important role in our ordinal-regression surrogate and population initialization process.
> >
> > Additionally, we have added a new ablation study to demonstrate the effectiveness of our diversity maintenance method as follows, the comparison is conducted on several 10-objective problems:
> > ```
> > 					DTLZ1			DTLZ2			WFG4
> > 	LORA-MaOO	2.31e-1(3.89e-2)	4.54e-1(1.41e-2)	5.47e+0(4.14e-1)
> > 	LOR-MaOO	3.09e-1(8.54e-2)	5.07e-1(2.37e-2)	6.11e+0(5.03e-1)
> > ```
> > LOR-MaOO is a variant of LORA-MaOO generated by removing its angular surrogates and diversity maintenance method. It can be observed that LORA-MaOO outperforms LOR-MaOO, demonstrating the effectiveness of our angular surrogate based diversity maintenance method.

---

> > > ### Author Response · Authors · 2024-12-01
> > >
> > > > Weakness8 (Continue)
> > >
> > > - Regarding the reviewer's request for virtualization, we would appreciate further clarification on the specific type of visualization needed. Currently, we use Figures. 1, 2, 3, 10, 11, and 12 to illustrate the convergence of comparison algorithms on different test problems. Figures 4 virtualizes the runtime of comparison algorithms. Figure 5 illustrates our artificial clustering-based ordinal relations. Furthermore, Figures 7, 8, and 9 virtualize the solution distribution of optimization results in the objective space.
> > >
> > > > Weakness9
> > >
> > > In Section 4.2 and Appendix F of our initial submission (ablation studies), we conducted a series of experiments to examine the impact of different components and their parameters. A brief discussion was available in Section 4.2 to provide intuitive explanations.
> > >
> > > In the revised version of our submission, we have removed this discussion from the main manuscript to save space for large and clear figures. We intend to extend these explanations as follows and include them in Appendix F (Appendix F.4 already has explanations):
> > > - Section F.1: When $M$ = 10, a large $n_o$ results in poor optimization performance. This is because the ratio of non-dominated solutions in the archive tends to be large when $M$ is large, hence, setting a large $n_o$ will lead to a lack of training samples in each dominated ordinal levels, which is detrimental to the performance of surrogate modeling.
> > > 	However, a small $n_o$ = 3 is unable to provide enough ordinal relations as the solutions in the same ordinal level are not comparable. Therefore, we set $n_o$ to 4.
> > > - Section F.2: When $\lambda = 0$, the algorithm component $\lambda$-dominance would be removed from LORA-MaOO and the resulting LORA-MaOO variant has the worst performance among all the variants. In comparison with other LORA-MaOO variants with $\lambda > 0$, it can be seen that using $\lambda$-dominance to sightly modify the original dominance relations is beneficial to the effectiveness of LORA-MaOO.
> > > 	Otherwise, some extreme solutions (e.g. solutions with one objective closes to the optimum while other objectives are far from desirable) may mislead the optimization.
> > > 	In addition, setting a large $\lambda$ could cause severe damage to the original dominance relations. Therefore, we set $\lambda$ to 0.2.
> > > - Section F.3:  When the ratio threshold of reference points $rp\_{ratio}$ is 1 and $M$ = 10, artificial ordinal relations are removed from LORA-MaOO, indicating no artificial ordinal relations are introduced to further divide ordinal levels for plenty of non-dominated solutions in the archive. Consequently, the imbalance of sample numbers in different ordinal levels leads to poor optimization performance. In contrast, other two variants with artificial ordinal relations achieve better optimization results.
> > > 	Notably, artificial ordinal relations are designed for MaOOPs with a large $M$. When $M$ = 3 and the size of ordinal levels are well balanced, dominance relations are preferable to artificial ordinal relations.
> > > 	Therefore, to ensure no artificial ordinal relations are introduced in this situation, the setting of $rp\_{ratio}$ should not smaller than the ratio of reference points in the archive.
> > > 	Hence, we set $rp\_{ratio}$ = 0.5.

---

### Author Response · Authors · 2024-12-03

Dear all reviewers:

Thank you for your thorough review and valuable feedback on our work.
We have addressed the comments point by point and made significant modifications to our manuscript.
New experiments and time complexity analysis are added and figures are revised.

Please let us know if we have addressed the concerns and if you have any additional questions.

Thanks!

The authors.

---

### Meta-Review · Area_Chair_SiAi · 2024-12-18

**Metareview:**

This paper proposes LORA-MaOO, a surrogate-assisted many-objective-optimization method. The reviews found the core idea of LORA-MaOO interesting, but raised concerns about the clarity, motivation, and effectiveness of the proposed method. Regarding the motivation, there is an unclear justification for cost reduction and non-parametric diversity maintenance. Furthermore, advantages over existing methods need stronger arguments. Regarding clarity, the abstract and the introduction lack precise explanations for key concepts. The effectiveness of the method is questioned by limited improvement over baselines in some cases. Experimental limitations include missing comparisons (some of them addressed in the discussion period) and unclear analysis of surrogate performance. I believe that this paper needs more work before it can be accepted. The authors should clearly explain how LORA-MaOO reduces surrogate cost and improves diversity compared to existing methods and quantify these benefits if possible. They should improve clarity and  address the limited performance gains over baselines. Conduct extra additional experiments with missing related methods.

**Additional Comments On Reviewer Discussion:**

The authors addressed some of the concerns of the reviewers during the discussion period, providing extra clarifications and experimental results. Some of the reviewers did increase their scores, but not enough to suggest accepting the paper.

---

### Decision · Program_Chairs · 2025-01-22

Reject